# Control of polymers' amorphous-crystalline transition enables miniaturization and multifunctional integration for hydrogel bioelectronics

Sizhe Huang[1,2], Xinyue Liu[3], Shaoting Lin [4], Christopher Glynn[2], Kayla Felix[2], Atharva Sahasrabudhe[5], Collin Maley[2], Jingyi Xu[2], Weixuan Chen [2], Eunji Hong[1,2], Alfred J. Crosby [6], Qianbin Wang [1,2] ✉ & Siyuan Rao [1,2] ✉

Soft bioelectronic devices exhibit motion-adaptive properties for neural interfaces to investigate complex neural circuits. Here, we develop a fabrication approach through the control of metamorphic polymers' amorphous-crystalline transition to miniaturize and integrate multiple components into hydrogel bioelectronics. We attain an about 80% diameter reduction in chemically cross-linked polyvinyl alcohol hydrogel fibers in a fully hydrated state. This strategy allows regulation of hydrogel properties, including refractive index (1.37-1.40 at 480 nm), light transmission (>96%), stretchability (139-169%), bending stiffness (4.6 ± 1.4 N/m), and elastic modulus (2.8-9.3 MPa). To exploit the applications, we apply step-index hydrogel optical probes in the mouse ventral tegmental area, coupled with fiber photometry recordings and social behavioral assays. Additionally, we fabricate carbon nanotubes-PVA hydrogel microelectrodes by incorporating conductive nanomaterials in hydrogel for spontaneous neural activities recording. We enable simultaneous optogenetic stimulation and electrophysiological recordings of light-triggered neural activities in Channelrhodopsin-2 transgenic mice.

Soft materials bioelectronics enable the interrogation of biological function from single-cell to organ-level resolution[1]. In dynamically moving in vivo environments, soft bio-interfaces can adapt to the persistent mechanical deformations of the living tissues, and consequently provide reliable access to biological systems. For the sophisticated yet delicate nervous system interfaces, elastic polymer materials, including polydimethylsiloxane (PDMS)[2], cyclic olefin copolymer elastomer (COCE)[3], polyurethane (PU)[4], and hydrogels[5,6] have been deployed as the suitably elastic substrate

for multifunctional devices that enable neural optogenetics stimulation[2,7–9], electrophysiological recording[10–12], drug infusion[13] and neurotransmitter detection[14]. However, fabricating dedicated microstructures in soft and elastic devices is limited to two-dimensional (2D) architectures and heavily relies on sophisticated manufacturing approaches such as lithography[15,16] and micro-printing[17].

The thermal pulling method yields multiple-step scaling-down feasibility for multifunctional polymer fibers[13,18], and this approach requires coherent parameters of the constituent materials, such as

[1]Department of Biomedical Engineering, Binghamton University, State University of New York, Binghamton, NY, USA. [2]Department of Biomedical Engineering, University of Massachusetts, Amherst, MA, USA. [3]Department of Chemical Engineering and Materials Science, Michigan State University, East Lansing, MI, USA. [4]Department of Mechanical Engineering, Michigan State University, East Lansing, MI, USA. [5]Research Laboratory of Electronics, Massachusetts Institute of Technology, Cambridge, MA, USA. [6]Department of Polymer Science and Engineering, University of Massachusetts, Amherst, MA, USA. ✉e-mail: qbwang@binghamton.edu; syrao@binghamton.edu

glass transition temperature ($T_g$), melting temperature ($T_m$) and thermal expansion coefficients ($\alpha$) to be drawn into an integrated fiber. Moreover, the high-temperature process narrows the selections of available polymers for high water content bioelectronics. Assisted with hydrogel cross-linking as a soft material matrix, hybrid multifunction fibers permit adaptive bending stiffness for long-term sensing and neural modulation[5,19].

Besides mechanical stiffness changes in the hydrated state and the desiccated state, hydrogel materials permit tunable volumetric control as the supporting scaffold. Hydrogel swelling behaviors in response to external stimuli have enabled drug release[20], ingestible devices[21], and expansion microscopy to enhance microimaging resolution[22]. Anti-swelling hydrogels are usually constructed by regulating the cross-linking, hydrophilicity/hydrophobicity balance, and nanocomposite for tissue engineering applications[23,24]. Meanwhile, hydrogel shrinking behaviors in a desiccated state have been applied to densify patterned materials in volumetric scaffold deposition and obtain nanoscale feature sizes in three dimensions[25,26]. However, the hydrogel swelling and shrinking behaviors in these techniques are based on reversible polymer-chain collapse in the desiccated state and expansion upon hydration. When applied to an aqueous in vivo environment, the shrunk hydrogels will expand and lose the miniaturized structures from the original manufacturing.

Inspired by the volumetric change resulting from polymer chains' folding and expansion, we propose a hypothesis centered on controlling the amorphous-crystalline transition in semi-crystalline hydrogels. By intervening in the polymer chain folding and crystallization process, we aim to limit the expansion of polymer chains from their nanocrystalline structure and consequently enable hydrogels to preserve their designed volumes under a solvated state. Utilizing the nanoscale structure change to regulate soft materials properties has been proven as an effective approach to biomedical applications[27-30]. Polyvinyl alcohol (PVA) hydrogels, one of the typical semi-crystalline polymers, have been extensively used in drug release[31,32], food packaging[33], and wound healing[34] owing to their high water content, transparency, and biocompatibility. In semi-crystalline polymer matrices, the swelling behavior involves water molecule diffusion, amorphous polymeric chain relaxation via hydration, and expansion of the cross-linked polymer network. To finely tune polymeric crystallization processes, we can apply engineering approaches that impact polymer chain interactions, solvent evaporation, and external stretching to facilitate molecular chain arrangement[35]. Moreover, the PVA polymer matrix can be incorporated with nanomaterials to enhance mechanical strength, conductivity, and biocompatibility[36,37]. For PVA hydrogel bioelectronics, controlling nanostructures through polymeric crystallization approaches can enhance the stability to maintain their designed architectures in biological environments.

Here, we developed a set of cross-linking chemistry and microfabrication processes to control polymeric crystalline domain growth with cross-linked PVA hydrogels. A stable and tunable volumetric decrease of hydrogels was consistently achieved in a hydrated state under physiological conditions (pH 6–8, 37 °C). Through acidification treatment that affects polymer chain interactions while introducing dual cross-linkers of the inorganic binder tetraethyl orthosilicate (TEOS) and the generic glutaraldehyde (GA), we minimized the polymetric crystalline scattering (crystal size around 3.5 nm) and increased the hydrogels' refractive indices (RI). Further nanocrystalline orientation induced by uniaxial deformation promoted the generation of nanoscale anisotropic architectures. This control of metamorphic polymers' amorphous-crystalline transition (COMPACT) strategy enabled a 79.7% diameter decrease of hydrogel fibers in the hydrated state while maintaining high stretchability (139.3-169.2%), relatively low elastic moduli (2.8–9.3 MPa), and low bending stiffness (4.6 ± 1.4 N/m). With the ability to control COMPACT hydrogel RI with a series of options, we developed core-cladding hydrogel fibers with distinct RI contrast ($n_{core}$ = 1.40, $n_{cladding}$ = 1.34). These core-cladding structured hydrogel fibers were applied for concurrent photometry recordings from mouse brain ventral tegmental area (VTA) in the context of social interactions. Taking advantage of these tunable hydrogel matrix scaffolds, we loaded conductive nanomaterials, carbon nanotubes (CNTs), into COMPACT hydrogels to fabricate soft microelectrodes, and tested its functionalities for electrophysiological recordings of spontaneous neural activity in the mouse brain. Integrating a hydrogel optical core and CNTs-PVA microelectrodes, we developed multifunctional hydrogel optoelectronic devices and demonstrated the simultaneous optogenetic stimulation and electrophysiological recording of optically triggered neural activities in transgenic *Thy1:: ChR2-EYFP* mice.

## Results and discussion
### COMPACT strategy for hydrogels controllable shrinking

Chemically cross-linked PVA hydrogels have been widely employed with superior optical properties[38], fatigue-resistance[39-41], and biocompatibility for bioelectronics applications[42,43]. To further explore PVA hydrogels' controllable miniaturization properties while preserving these advantageous features, we designed fabrication approaches by control of metamorphic polymers' amorphous-crystalline transition with the following aspects: (i) polymer chains folding and immobilization with multiple cross-linkers, (ii) intervention on intermolecular chain interactions in the hydrogel matrix, (iii) inducing the oriented growth of nanocrystalline domains. We implemented the COMPACT strategy following three major procedures to control individual polymer chain folding, polymer chain network interactions, and nanocrystalline growth. We first introduced the hydrolysis of TEOS in PVA solutions through homogenization (Fig. 1a and Supplementary Fig. 1), followed by the addition of a generic cross-linker (GA). A combination of two types of cross-linkers is chosen to allow the control of polymer chain mobility via covalent bonding and parallel tuning of hydrogels' refractive index. We then acidified the cross-linked hydrogels to promote intermolecular chain interactions. We prepared fiber-shaped hydrogels via molding and extrusion methods (Supplementary Fig. 2a). External mechanical stretching was applied to the fully acidified hydrogels and maintained during the desiccating process (Supplementary Fig. 2b). After the removal of water molecules from hydrogels, high-temperature (100 °C) annealing was employed to further promote the growth and orientation of the nanocrystalline domains. To test whether the COMPACT strategy can preserve hydrogels volumetric shrinking under a hydrated state, we next examined the dimensions and water fractions of cross-linked hydrogels under pristine, desiccated, and rehydrated states (Fig. 1b–e).

At the pristine (Fig. 1b) and desiccated states (Fig. 1c), the two hydrogel fibers with TEOS-GA cross-linking (COMPACT+) and GA cross-linking (COMPACT−) exhibited comparable geometries and water fractions (Fig. 1e); however, only the TEOS-GA cross-linked PVA hydrogel fiber with acidification and mechanical stretching maintained the reduced diameters in the rehydrated state (Fig. 1d, e).

After we confirmed that hydrogels retained shrinking behaviors in the rehydrated state with COMPACT treatment, we tested whether size reduction is dependent on the materials' geometries and external constraints. We prepared hydrogels with the shapes of thin film, fiber, and block, and examined the changes of COMPACT hydrogel film thickness (T, Fig. 1f), fiber diameter (D, Fig. 1g), and volume (V, Fig. 1h). TEOS-GA cross-linked PVA hydrogel thin films with acidification treatment exhibited a thickness reduction ratio of 93.4 ± 3.6% (pristine thickness: 501 ± 134 μm; rehydrated thickness: 33 ± 18 μm, Fig. 1f). TEOS-GA cross-linked PVA hydrogel fibers, with treatment of acidification and mechanical strain (200%), reached the maximum diameter shrinking ratio of 79.7 ± 2.3% (Fig. 1g). In three-dimensional (3D)

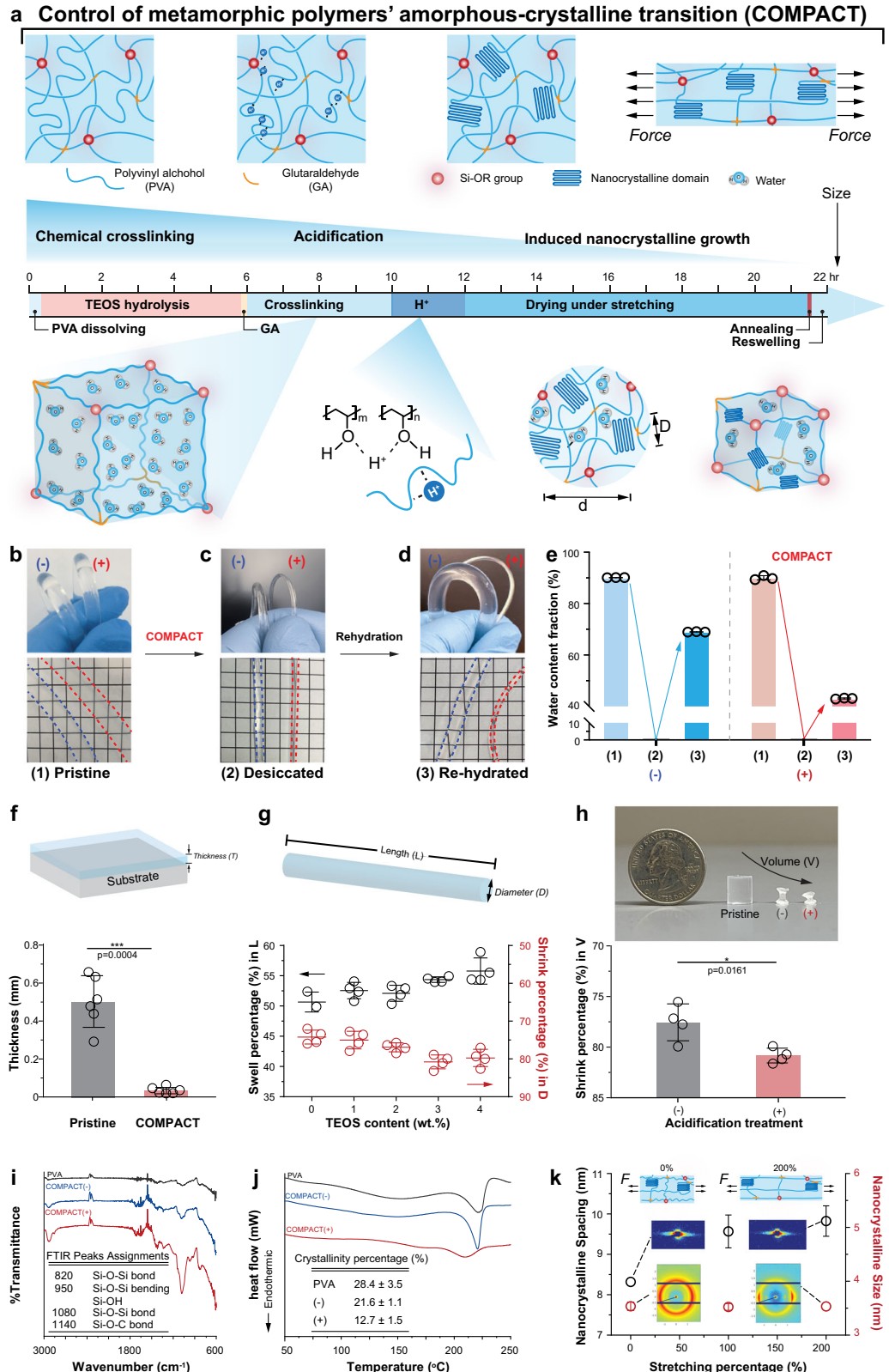

**a  Control of metamorphic polymers' amorphous-crystalline transition (COMPACT)**

free shrinking structures, we observed 80.9 ± 0.7% volumetric shrinking in acidified TEOS-GA cross-linked cylinders (Fig. 1h).

We then investigated the mechanisms of the sustained hydrogel volumetric decrease and the design of amorphous and crystalline architectures. Fourier transform infrared spectroscopy (FTIR) results indicated covalent bonds (Si-O-Si and Si-OH) generated in the COMPACT hydrogel network (Fig. 1i). The Si-O-Si (1080 cm⁻¹) and Si-OH

(950 cm⁻¹) bonds came from the hydrolyzed TEOS Si-OR groups' reactions with the hydroxyl groups on PVA chains. The generic cross-linker GA reactions were confirmed by the observation of C=O bond (1740 cm⁻¹) and the Si-O-C bond (1140 cm⁻¹) from the reaction with TEOS Si-OR groups. Besides confirming covalent bonds generated among hydrogel polymer chains, differential scanning calorimetry (DSC) results exhibited the change of polymer chain interactions and

**Fig. 1 | COMPACT strategy for hydrogel miniaturization. a** Schematic illustration of hydrogel network of metamorphic polymers' amorphous-crystal transition (COMPACT). COMPACT treatment includes cross-linking with both glutaraldehyde (GA) and tetraethyl orthosilicate (TEOS), acidification, and mechanical stretching. **b**–**d**, Representative photographs and water contents of TEOS-GA cross-linked polyvinyl alcohol (PVA) hydrogel with COMPACT treatment (+) and GA cross-linked hydrogel without acidification and stretching (−) at the pristine state (**b**), desiccated state (**c**) and rehydrated state (**d**). Grid size: 5 mm. **e** Water content of hydrogel fibers (non-COMPACT and COMPACT). Data presented as mean ± standard deviation (s.d., $n = 3$ independent hydrogel fibers). **f** Shrinking behaviors of TEOS-GA cross-linked PVA (4 wt. % TEOS) hydrogel film with acidification treatment. Film thickness is quantified as mean ± s.d. (Two-tailed paired student's $t$-test, $F = 56.78$, $t = 8.455$, df = 10, $\alpha = 0.05$, ***$p = 0.0004$, $n = 6$ independent hydrogel films). **g** Shrinking behaviors of COMPACT hydrogel fibers (1–4 wt. % TEOS and 200%

stretching). Hydrogel fibers' length and diameter are quantified as mean ± s.d. ($n = 4$ independent hydrogel fibers). **h** Shrinking behaviors of cross-linked hydrogel cylinders. The volume of TEOS-GA cross-linked hydrogel cylinders (4 wt. % TEOS) and with acidification treatment and GA cross-linked hydrogel cylinders without acidification treatment are compared with mean ± s.d. (Two-tailed unpaired student's $t$-test, $F = 6.084$, $t = 3.316$, df = 6, $\alpha = 0.05$, *$p = 0.0161$, $n = 4$ independent hydrogel cylinders). **i** Fourier transform infrared (FTIR) spectroscopy of COMPACT (−) and COMPACT (+) hydrogels. **j** Differential scanning calorimetry (DSC) profiles of COMPACT (−) and COMPACT (+) and their crystallinity percentages. **k** Small-angle X-ray (SAXS) and wide-angle X-ray (WAXS) results of hydrogel materials in the desiccated state (mean ± s.d., $n = 3$–6 independent hydrogel samples). Upper inset: schematic illustration of nanocrystalline distance change before and after stretching, middle inset: SAXS 2D patterns., lower inset: WAXS 2D patterns.

polymeric crystallinity after COMPACT treatment[44]. PVA powders showed 28.4 ± 3.5% crystallinity (Fig. 1j and Supplementary Table 1), similar to the reported crystallinity percentage of semi-crystalline PVA polymers[45]. GA-cross-linked PVA hydrogels exhibited 21.6 ± 1.1% crystallinity while the additional TEOS cross-linking and acidification suppressed the polymer chain folding to form crystalline domains (crystallinity: 12.7 ± 1.5%, Supplementary Figs. 3 and 4, and Supplementary Table 2). We further examined the nanocrystalline domains and orientation with X-ray scattering techniques (Supplementary Table 3). The Small-angle X-ray scattering (SAXS) results suggested that the size of PVA nanocrystals was measured as 3.5 ± 0.1 nm while the nanocrystalline spacing increased from 8.32 ± 0.08 nm to 9.83 ± 0.38 nm after 200% axial stretching (Fig. 1k and Supplementary Fig. 5–9). Wide-angle X-ray scattering (WAXS) 2D patterns suggested that the lamellae crystal domains were re-oriented along the axial stretching direction (Fig. 1k and Supplementary Figs. 3c, 10, 11a, b, 12, and 13). Similar anisotropic nanocrystalline structures have been reported in other PVA hydrogels[46].

In COMPACT hydrogels, chemical cross-linking and acidification treatment both contribute to the retained volumetric decrease upon rehydration while mechanical deformation induces the orientated nanocrystalline growth. An increased number of chemical cross-linkers, TEOS (0–4 wt. %, Fig. 1g), enhanced the anchoring of amorphous PVA chains through covalent cross-linking and prevented swelling in the hydrated state. Under the same cross-linking degree, acidification treatment granted polymer chains enhanced interactions and suppressed the folding of polymer chains to form crystalline (Fig. 1j and Supplementary Fig. 4). Nanocrystalline domains maintained the nanoscale size (~3.5 nm) without compromising the light transmittance in the visible range. Axial mechanical deformation introduced tensile stress to re-orient polymer chain alignment and created anisotropic nanostructures (Fig. 1k and Supplementary Figs. 6 and 7), which enabled hydrogel fibers' desired decrease in diameter while causing a minimal effect on crystallinity degree (Supplementary Fig. 3a, b) or nanocrystalline size (Fig. 1k and Supplementary Fig. 5c, d).

Controllable hydrogel shrinking through cross-linking and polymer chain crystallization process provides an effective methodology to miniaturize hydrogel bioelectronics, especially for the application in vivo. The approaches to regulating polymer chain folding and interactions can be extended to other semi-crystalline polymers. Without affecting the nanocrystalline size, the mechanical stretching method offers a straightforward way to create anisotropic orientations of polymeric nanostructures. The molding and extrusion approaches offer a series of precisely controlled hydrogel fiber diameters with structural homogeneity and low surface asperity to avoid diffuse reflection at the hydrogel interfaces. Although the mold sizes are commercially limited, COMPACT procedures, including regulating polymer and cross-linker constituent content and fiber extensions can expand the range of available fiber sizes.

## COMPACT hydrogel fibers' tunable properties

With COMPACT-enabled hydrated hydrogel size reduction, we expanded this methodology to develop a series of hydrogel fibers with controlled diameters and tunable optical and mechanical properties for biomedical use. We mapped a rational and comprehensive shrinking diagram by varying the content of inorganic cross-linker (TEOS), acidification, and external mechanical stretching (Fig. 2a). Generally, increasing cross-linking density with more cross-linkers yielded less ductile polymer chains with reduced dimensions upon hydration. Acidification treatment dramatically boosted shrinking percentages across different cross-linking densities while mechanical static stretching further decreased hydrogel fibers in diameters (79.7 ± 2.3%). To fit COMPACT into a practical molding-extrusion fabrication process (Supplementary Fig. 2a)[37,47], we examined a series of hydrogel fibers made with different sizes of silicone molds (Fig. 2b and Supplementary Fig. 14). Independent of the mold size, all COMPACT hydrogel fibers reached reduced diameters of more than 79%, which is consistent with the shrinking diagram (Fig. 2a). As an example, using 300 μm (inner diameter, ID) silicone molds, thin hydrogel fibers were fabricated with diameters of 80 ± 4 μm.

Considering the fiber optic in vivo applications[9,37,41], we examined the optical, mechanical, and cytotoxicity properties of COMPACT hydrogel fibers. To ensure efficient light transmission for optical stimulation and recordings, we considered two important parameters of the hydrogel fiber core: refractive index (RI) and light transmittance. We found that hydrogels' refractive indices can be tuned by TEOS contents. COMPACT hydrogels with 0 wt. % to 4 wt. % TEOS contents exhibited refractive indices ranging from 1.48 to 1.60 in the desiccated state (Fig. 2c) and 1.37 to 1.40 in the hydrated state (Supplementary Fig. 15a, b), which is comparable with the RI of other conventional polymers[48]. Although all the transmittance remained above 96%, increasing TEOS content also led to decreased transmittance (Fig. 2c and Supplementary Fig. 15c), and increased autofluorescence (17.8% increase of 4 wt. % TEOS hydrogels compared to 0 wt. % TEOS hydrogels, excitation wavelength: 485 nm, excitation peak: 520 nm, Supplementary Fig. 13d). The optimal TEOS content was chosen as 3 wt. %, which resulted in hydrogels with 1.54 ± 0.01 of refractive index (Fig. 2c), >96% of transmittance (Fig. 2c, for 0.15 ± 0.02 mm thick membranes), and 6.13 ± 0.16 relative fluorescent units (RFU)/mm of autofluorescence (for 0.15 ± 0.02 mm thick membranes. water: 3.70 RFU/mm, Supplementary Fig. 15d).

To mimic the in vivo working condition, we examined COMPACT hydrogels' mechanical properties in the hydrated state. The COMPACT hydrogel fibers exhibited relatively low elastic moduli while maintaining high stretchability (Fig. 2d and Supplementary Fig. 16). The optimized COMPACT hydrogel fiber (3 wt. % TEOS, 12 mM HCl acidification treatment and 200% stretching, diameter: 227 ± 18 μm) exhibited an elastic modulus of 4.8 ± 1.7 MPa and stretchability of 139.4 ± 26.0%. When fiber-shaped neural probes are inserted into brain tissues, their axial bending stiffness serves as an important mechanical

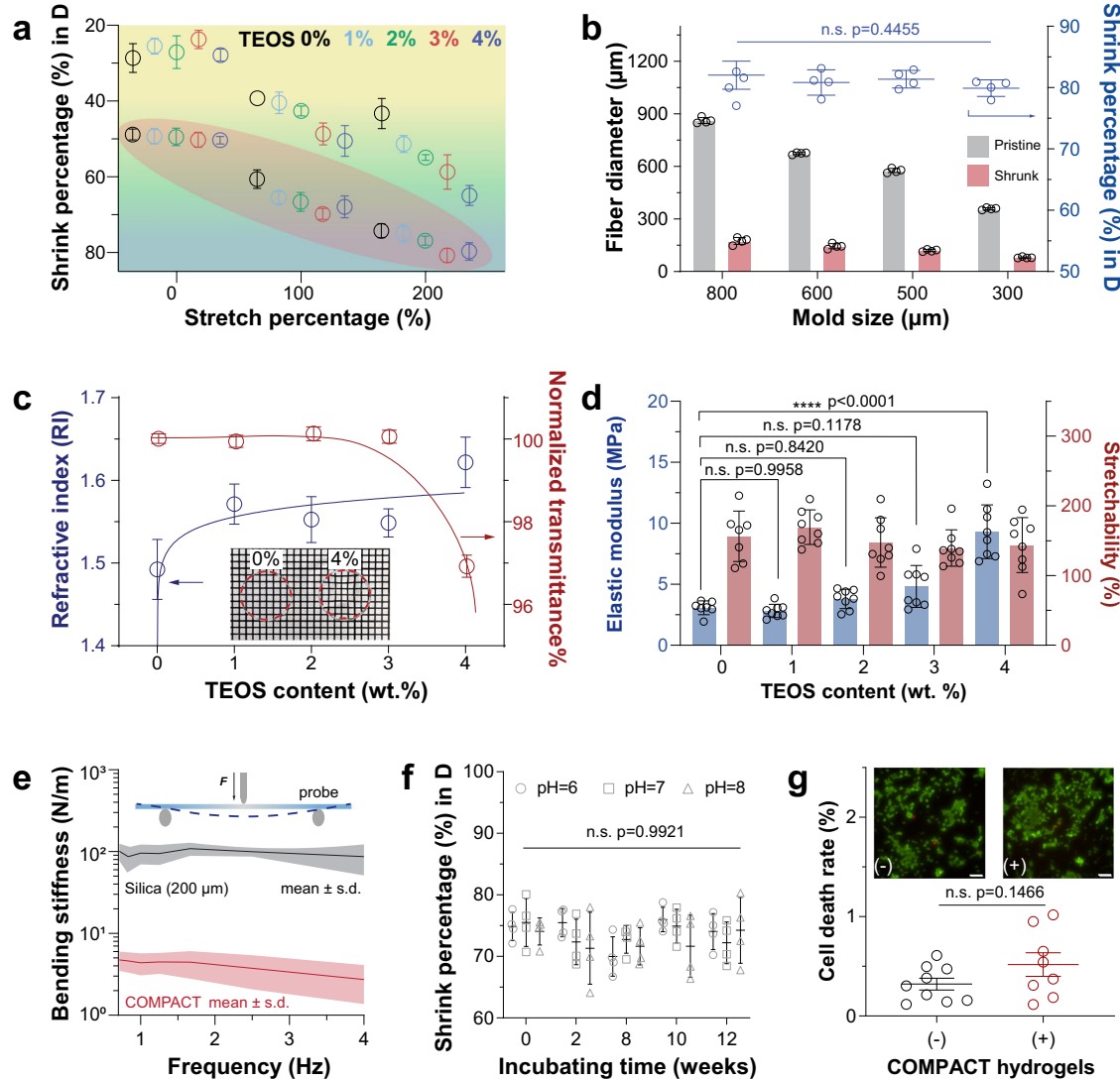

**Fig. 2 | Controllable hydrogel fiber fabrication and its properties. a** A shrinking diagram of COMPACT (+) hydrogel fibers. The samples shaded in red areas are treated with acidification. Mean ± s.d. $n = 3–4$ independent hydrogel fibers. **b** Shrinking behaviors of COMPACT hydrogel fibers (4 wt. % TEOS) prepared in different sizes of molds. One-way ANOVA and Tukey's multiple comparisons test ($F_{3,12} = 0.9543$, n.s. not significant, $p = 0.4455$). Mean ± s.d. $n = 4$ independent hydrogel fibers. **c** COMPACT hydrogel fibers' optical properties of refractive index and normalized light transmittance. Inset: representative photographs of 0 wt. % TEOS and 4 wt. % TEOS hydrogel membranes. Grid size: 1 mm. Mean ± s.d. $n = 5–9$ independent hydrogel samples. **d** COMPACT hydrogel fibers' elastic modulus and stretchability calculated from tensile tests in water. One-way ANOVA and Tukey's multiple comparisons test were used to determine the statistical significance of elastic modulus: ($F_{4,34} = 30.07$, ****$p < 0.0001$) and stretchability: ($F_{4,34} = 1.040$, n.s.,

$p = 0.4009$), respectively. Mean ± s.d., $n = 7–8$ independent hydrogel samples. **e** Bending stiffness of COMPACT (3 wt. % TEOS) hydrogel fiber ($n = 4$ independent hydrogel fibers) with identical cross-sections in comparison with silica fiber (diameter: 200 μm, mean ± s.d., $n = 3$ independent samples). **f** Stability assessment of diameter reduction of COMPACT hydrogel fibers (3 wt. % TEOS). Two-way ANOVA and Tukey's multiple comparisons tests ($F_{14, 72} = 0.3027$, n.s., $p = 0.9921$, mean ± s.d., $n = 4$ independent hydrogel fibers). **g** Cytotoxicity assessment of COMPACT (+) hydrogels. Hydrogel fibers were incubated with Human Embryonic Kidney (HEK) 293 cell cultures. Calcein-AM (green) was used to stain living cells and ethidium homodimer-1 (red) was used to stain dead cells. Cell death rates are presented as mean ± standard error (s.e.m.), Two-tailed unpaired student's $t$-test was used: $F = 3.570$, $t = 1.531$, df = 15, $\alpha = 0.05$, n.s. $p = 0.1466$, $n = 8–9$ microscopy measurements from 3 independent culture samples. Scale: 50 μm.

parameter under brain micromotions (Supplementary Fig. 17). Compared to silica fibers (~20 GPa elastic modulus)[49] and polymer fibers (~1 GPa elastic modulus)[3,5], COMPACT hydrogel fibers offer improved mechanical matching to the nervous tissues (1–4 kPa)[50,51] and much lower axial bending stiffness (Fig. 2e) to achieve less neural tissue damage from micro-motion involved in vivo studies[52,53]. Our initial evaluation focused on the immune response of brain tissues 14 days post-implantation. We observed a reduced presence of astrocytes, microglial accumulations, activated macrophages, and immunoglobulin G around the sites of hydrogel fiber implantation compared to the stiffer silica fibers (Supplementary Fig. 18). Subsequently, we assessed the chronic immune response after 30 days and noted a lower

incidence of astrocyte and microglial formation around the hydrogel fiber sites relative to those with silica fibers (Supplementary Fig. 19).

We then tested whether crystalline-enabled size reduction of COMPACT hydrogels can overcome the intrinsic hydrogel swelling exhibited upon hydration and maintain structural stability in vivo. We incubated COMPACT hydrogel fibers in ex vivo physiological conditions (pH: 6–8, 37 °C, saline solution) and monitored fibers' dimensions over time. We observed the shrinking percentage maintained above 74% over 3 months (Fig. 2f and Supplementary Fig. 20). Cytotoxicity tests with human embryonic kidney cells (HEK293) exhibited no significant cell death in the presence of COMPACT hydrogels (Fig. 2g and Supplementary Fig. 21).

Unlike other established approaches to shrink hydrogels via desiccation, where the collapse of the polymer chain during drying leads to reversible swelling upon hydration, COMPACT hydrogels' polymeric nanocrystalline and enhanced interpolymer chain interactions maintained stable folding in the hydrated state and therefore permit retained volumetric size reduction. Over 3 months of incubations under physiological temperature and osmolarity, the shrunk COMPACT hydrogel fibers maintained the designed diameters within less than 1% variance (Fig. 2f), which illustrates hydrogel bioelectronics' volumetric stability of their miniaturized size in vivo[37]. In contrast, COMPACT hydrogel fibers incubated at PVA dissolution temperature (100 °C) in water for several hours resumed their pristine swollen size; this volume reversion demonstrates the crystalline impact on size reduction through control of local free volume in hydrogel matrices. This crystalline-dominated hydrogel miniaturization phenomenon can be extended to other semi-crystalline polymers at different material interfaces, where volumetric stability is important, such as the proton-exchange membrane in packed fuel cells.

## Step-index hydrogel optical fibers

COMPACT hydrogels were first fabricated into step-index optical fibers (Supplementary Fig. 22). Increased RI contrast between optical core and cladding layers ensures light transmission and the consequent photodetection sensitivity (Fig. 3a). Based on tunable refractive indices of COMPACT hydrogels (Fig. 2c and Supplementary Fig. 15a and b), we designed step-index hydrogel fibers with high-RI core ($n_{core} = 1.40$) and low-RI cladding ($n_{cladding} = 1.34$). Hydrogel fibers were connected to a silica segment embedded in an optical ferrule, which provides a strong connection while preventing directly exposed hydrogel dehydration out of tissues and light loss[41] (Supplementary Fig. 22). We validated the function of RI-contrasting core-cladding structures by comparing the light transmission between bare core fibers, step-index fibers with plain cladding and those with light-protective cladding (Fig. 3b, c, and Supplementary Figs. 22 and 23). The bare core fibers (diameter of $329 \pm 17\,\mu m$) exhibited a relatively high attenuation ($1.87 \pm 0.53$ dB/cm) while introducing a thin low-RI cladding layer (thickness of $84 \pm 4\,\mu m$ on the surface of $372 \pm 10\,\mu m$ cores, $n_{cladding} = 1.34$) decreased the light transmission attenuation to $1.75 \pm 0.08$ dB/cm (Fig. 3d). A representative light-absorption nanomaterial[54,55], reduced graphene oxide (rGO) was loaded into low-RI cladding to further protect light leakage from fibers' lateral surface and consequently reduced the light attenuation to $0.94 \pm 0.25$ dB/cm (core $339 \pm 35\,\mu m$, cladding: $36 \pm 11\,\mu m$ of 5 wt. % PVA with 0.21 wt. % rGO) (Fig. 3d and Supplementary Fig. 24).

To validate their functionality for in vivo optical interrogation, we tested COMPACT hydrogel fibers with photometric recording in the context of mouse social behaviors. Activation of VTA region and its related circuits has been studied with various techniques[56–58], related to social behaviors in mice[59]. As a proof-of-concept application, we validated COMPACT hydrogel fibers to photometrically record mouse deep brain structure, VTA, with concurrent social behavior observation. We unilaterally implanted COMPACT optical fibers ($580 \pm 35\,\mu m$) in VTA after injecting adeno-associated virus (AAV) containing genetically encoded calcium indicator (hSyn::GCaMP6s) (Fig. 3e). The stiffness changes of the hydrogel fibers from a desiccated state (stiff) to a hydrated state (soft) facilitated the implantation with calibrated mouse brain coordinates (Supplementary Figs. 25 and 26). A fiber photometry system (wavelengths: $\lambda_{isosbestic\ point} = 405$ nm, $\lambda_{excitation} = 470$ nm, $\lambda_{emission} = 510$ nm) was used to collect GCaMP fluorescent changes as proxies to reflect the neural activity[60] (Fig. 3f, g and Supplementary Fig. 27). After an incubation period of 4 weeks for AAV expression, we subjected mice to social behavioral tests with concurrent photometric recordings. Mouse social interactions were analyzed with DeepLabCut (DLC)[61] markless pose estimation and a custom-developed MATLAB algorithm (Fig. 3f). We observed the increased fluorescent intensity of GCaMP was correlated with mouse social interaction epochs (Fig. 3h). Linking the neural activities at the cellular level to system neuroscience behavioral assessment provides important tools to discover the causal relationship of neural circuits and behaviors for neuroscience studies.

## Multifunctional hydrogel neural probes

Hydrogel matrix can incorporate various nanoscale materials to extend the functionalities while maintaining stretchability[12,62]. To enrich hydrogel neural probes' modality for electrical recordings, we introduced conductive CNTs (length to diameter ratio 2000–10,000:1) into PVA hydrogels during hydrogel cross-linking (Fig. 4a, b, and Supplementary Fig. 28). Through acidification to promote polymer chain interactions and mechanical stretching facilitated CNTs plaiting into polymer matrices and ensured entanglement with PVA chains and consequently augmented electrical conductivity as a percolated network[63,64]. After introducing CNTs (0.08–0.24 wt. %), we observed insignificantly changed nanocrystallinity (Supplementary Fig. 29 and Supplementary Table 4) but decreased nanocrystalline sizes (Supplementary Figs. 30 and 31). These results could be interpreted as the limited polymer chain folding under additional interactions between nanomaterials and polymer chains. When CNTs-PVA hydrogel fibers underwent mechanical stretching, they preserved anisotropic structures with similar sizes (Supplementary Figs. 11c, 32, and 33). With rigid carbon materials incorporated into the hydrogel matrix, the CNTs-PVA composite exhibited an increase in elastic modulus (0.16 wt. % CNTs-PVA hydrogels, $39.4 \pm 13.7$ MPa), and decreased stretchability ($47.9 \pm 12.2\%$, Supplementary Fig. 34). For the use as an electrode in vivo, we optimized the CNT concentration to balance the conductivity and mechanical properties. CNTs-PVA hydrogel electrodes ($86 \pm 5\,\mu m$ diameter, Fig. 4c), insulated with a viscoelastic coating of styrene-ethylene-butylene-styrene (SEBS) (Supplementary Figs. 28 and 35), exhibited impedances of $658 \pm 277$ kΩ at 1 kHz and impedance was tunable with designed mold sizes to control the electrode diameters and CNTs loadings (Fig. 4d, e). The stability of CNTs-PVA electrodes was evaluated through over 6 weeks of incubation in PBS (37 °C, Fig. 4f), over 2 weeks of incubation in artificial cerebrospinal fluid (aCSF), and implantation in mouse brain tissues (Supplementary Fig. 36). We observed stable impedance performance under all these conditions.

The functionality of CNTs-PVA hydrogel electrodes was validated for in vivo electromyographic (EMG) and extracellular electrophysiological recordings. We first deployed CNTs-PVA hydrogel electrodes for EMG recordings of mouse hindlimb muscles in response to the pulsed blue light illumination. CNTs-PVA hydrogel electrodes repeatedly detected hindlimb muscle electrical signals upon transdermal optical stimulation (wavelength $\lambda = 473$ nm, 200 mW/mm², 0.5 Hz, pulse width 50 ms) in *Thy1::ChR2-EYFP* mice, which express photo-excitatory opsin, Channelrhodopsin-2 (ChR2), in the nervous system (Supplementary Fig. 37). Instead of recording collective electrical response from muscles, we then implanted CNTs-PVA hydrogel electrodes in mouse VTA to record neuron spontaneous spiking activity in anesthetized wild-type mice under continuous isoflurane (Fig. 4g–i). A bandpass filter of 300–3000 Hz was applied to detect spiking activity, and we found one distinct cluster of spikes of principal component analysis (PCA). The signal-to-noise ratio (SNR) of these spiking activities was approximately 3.73 with repeatable waveforms.

When extending hydrogel miniaturization from bulk materials to interfaces, the COMPACT strategy offers an opportunity for multiple components integration. Since RI-distinct core-cladding structures ensure light transmission in optical cores, we introduced two CNTs-PVA electrodes into the cladding layers with a COMPACT hydrogel core (Fig. 4j). A hydrogel optoelectrical device termed optrode (Fig. 4k and Supplementary Fig. 38), is designed to enable optical modulation with simultaneous electrophysiological recording. In *Thy1::ChR2-EYFP*

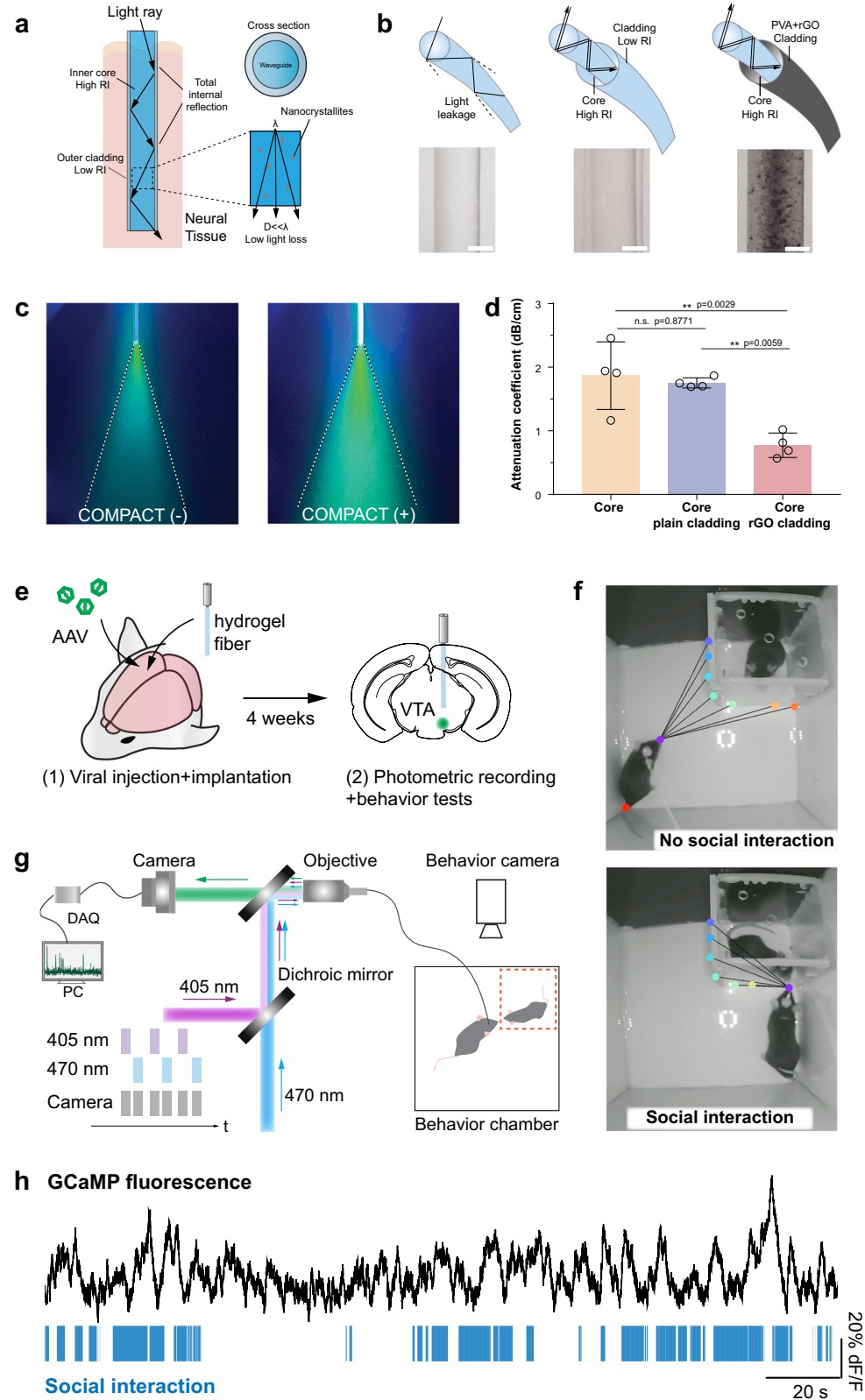

**h** GCaMP fluorescence

Social interaction

20% dF/F

20 s

mice, blue light pulses (λ = 473 nm, 0.5 Hz, pulse width 50 ms, 10 mW/mm²), delivered through the hydrogel optical core, consistently activated ChR2-expressing neurons in VTA while the neural electrical signals were collected through CNTs-PVA electrodes (Fig. 4l-m). The optical evoked potentials were repeatedly captured with correlation with the onset of light stimulation over 10 weeks post-implantation (Fig. 4n, o).

Simultaneous bi-directional stimulation and recording of neurons with optical and electrical modalities offer comprehensive approaches to studying brain function. The COMPACT strategy offers the convenience of integrating multiple functional components into a single miniaturized device. Successive rounds of molding with strong polymer chain infiltration at the interfaces enable the design of multimodal microstructures. Currently, the number of integrated components,

**Fig. 3 | Hydrogel optical neural probes for photometric recording with behavioral assessment. a** A schematic illustration of light transmission in a step-index hydrogel fiber. **b** Schematic illustrations and representative photographs of a COMPACT core hydrogel fiber, a COMPACT core-plain-cladding hydrogel fiber, and a COMPACT core-reduced reduced graphene oxide (rGO, 0.21 wt. %)-cladding fiber. Scale: 200 μm. **c** Representative photographs of blue light (480 nm) transmission from a COMPACT (−) core hydrogel fiber and a COMPACT (+) core hydrogel fiber into solutions containing Calcein fluorescent dye. **d** Light attenuation coefficients of COMPACT core hydrogel fibers, COMPACT core-plain-cladding hydrogel fibers, and COMPACT core-rGO-cladding fibers (mean ± s.d., One-way ANOVA and Tukey's

multiple comparisons test, $F_{2,9} = 13.3$, \*\**p* = 0.0021, *n* = 4 independent optical probes). **e** Experimental scheme for the viral injection, optical fiber implantation, fiber photometry recording, and mouse social behavior tests. **f** Representative images of tracking mouse social interactions. **g** A schematic illustration of fiber photometry recording setup with concurrent mouse social behavior tests. **h** Normalized fluorescence intensity change ($\Delta F/F_0$) of GCaMP6s from mouse VTA during social behavior tests. Blue bars indicate social interaction time analyzed by DeepLabCut (DLC). Fiber photometry recordings were performed with 5 mice implanted with COMPACT hydrogel optical probes.

such as electrodes and microfluidic channels, is limited by the coaxial alignment in the secondary molding step; the accessibility and throughput of multimodal fabrication can be further improved with guiding devices to facilitate integration and alignment, or alternative coating approaches.

In this study, we developed a set of hydrogel cross-linking chemistry and fiber-shaped device microfabrication approaches through a bottom-up strategy of tuning polymers' amorphous-crystalline transition for hydrogel bioelectronics miniaturization and integration. The COMPACT strategy provides an accessible, scalable, and controllable fabrication method for micro-structured hydrogel fibers with consistently low asperity. These hydrogels provide a platform for functionally augmented interfaces through loadings of additional nanomaterials. COMPACT hydrogels can be further designed into step-index optical probes and optoelectronic devices which are well-suited for neural modulation and recordings concurrent with behavioral assays in mice.

The COMPACT strategy is generalizable for soft and stretchable bioelectronics. Polymer matrices provide sufficient free volume for water access as well as nanomaterials' incorporation. High aspect-ratio nanomaterials can be effectively entangled with polymer chains through cross-linking and condensation during acidification and stretching. This procedure augments electrical conductivity while maintaining viscoelasticity. Compared to other soft bioelectronics fabrication approaches, such as lithography and micro-printing, COMPACT technique offers scalable and efficient multimodal hydrogel fibers manufacturing without the need for expensive and sophisticated facilities. COMPACT multifunctional hydrogel neural probes have been employed for bi-directional optical interrogation concomitant with mouse social behaviors and electrical recordings of light-triggered neural activity in mice. Extended functionalities, such as drug or viral vector delivery, can be further achieved by integrating additional microfluidic channels in the cladding layer and retaining light transmission efficiency in the optical core. COMPACT multifunctional neural probes involve independent component alignment and miniaturization steps, which potentiates the integration of multiple components with various lengths to target multiple depths of tissue within single-step implantation. This adaptability will increase the density of functional interfaces and overcome the traditional limitation of fiber-shaped neural probes with single-target interfaces at the tip.

Control over semi-crystalline polymers' amorphous-crystalline transition creates a direct fabrication methodology for elastic soft materials. Extending it to the manufacture of sophisticated optoelectronic devices, the COMPACT strategy imparts a generalizable and modular platform for hydrogel bioelectronics' miniaturization and integration, which consequently enables multimodal interrogation of complex biological systems.

## Methods
### Ethical statement
All experiments on mice were reviewed and approved by the Institutional Animal Care and Use Committee at Binghamton University (Protocol number: 897-23) and University of Massachusetts Amherst

(Protocol number: 2520). Wild-type (C57BL/6 J, https://www.jax.org/strain/000664) mice and *Thy1::ChR2-EYFP* transgenic mice (https://www.jax.org/strain/007612) were purchased from the Jackson Laboratory. Mice were given ad libitum access to food and water and were housed at $24 \pm 1\,°C$, with 50% relative humidity, and on a 12-h light/12-h dark cycle. All experiments were conducted during the light cycle. The species, strain, sex, number, and age of mice used in every experiment are included in Supplementary Table 5.

### Hydrogel synthesis
The chemicals used in this study included tetraethyl orthosilicate (TEOS, Sigma-Aldrich 86578, 99%), hydrochloric acid (HCl, Sigma-Aldrich, 258148, 37%), glutaraldehyde solution (GA, Sigma-Aldrich G6257, 25% in water), and polyvinyl alcohol (PVA) with an average molecular weight of 146,000 to 186,000 Da and 99 + % hydrolyzed (Sigma-Aldrich, 363065). Materials were used as received. MilliQ water with a resistivity of 18 MΩ·cm at 25 °C was used throughout the experiments. To prepare the PVA (10 wt. %) solution, PVA was dissolved in MilliQ water and stirred in a water bath at 100 °C for at least 4 h until a clear and transparent solution was obtained. The hydrolysis of TEOS was carried out using HCl as a catalyst in PVA solutions with a molar ratio of TEOS: HCl: $H_2O$ = x: 4: y, where x was between 1 and 4, and y started from 4 to 16. TEOS solutions with concentrations ranging from 2 wt. % to 8 wt. % were added to the PVA solutions, which were then homogenized at two different levels. A mixture of HCl and MilliQ water in a molar ratio of 4: y, where y was in the range of 4 to 16, was added dropwise to the PVA-TEOS emulsion while homogenizing at 12,000 rpm using a portable homogenizer until a stable emulsion was formed. The resulting emulsion was further homogenized using a high-speed homogenizer (FSH2A lab). The mixed solutions were stirred in a water bath at 100 °C for 1 h until transparent solutions were obtained, followed by an additional 12 h of stirring at 60 °C. The composition of all solutions used in this study is provided in Table 1.

### Optical hydrogel probe fabrication
A step-index multimode silica fiber (core diameter 400 μm, NA 0.5, Thorlabs FP400URT) was prepared by removing the protective coating using a fiber stripping tool (Micro-strip, Micro Electronics, Inc). The stripped fiber was then divided into 13-mm segments using a diamond cutter. These fiber segments were inserted and extruded from one end of an optical ferrule (bore diameter 400 μm, Thorlabs CFX440-10) with a length of 2.5 mm and secured with EccoBond F adhesive (Loctite). Both ends of the silica fibers in the ferrules were polished using a polish kit (Thorlabs D50-F, NRS913A, and CTG913). The light transmission of all silica fibers and ferrules was tested by coupling with a 470 nm blue light-emitting diode (LED) (Thorlabs M470F3) after polishing. To remove the plastic coatings on the extruded silica fibers, they were treated with 2 M sodium hydroxide solution (Sigma-Aldrich, 1064980500) for 2 h followed by an additional treatment with chloroform (Sigma-Aldrich, 472476) for 30 min. A thin layer of 10 wt. % PVA was then coated on the extruded silica fibers via dip coating, and the PVA-coated silica fibers were air-dried at room temperature for 12 h and annealed at 100 °C for 2 h. A vacuum planetary mixer (Musashi

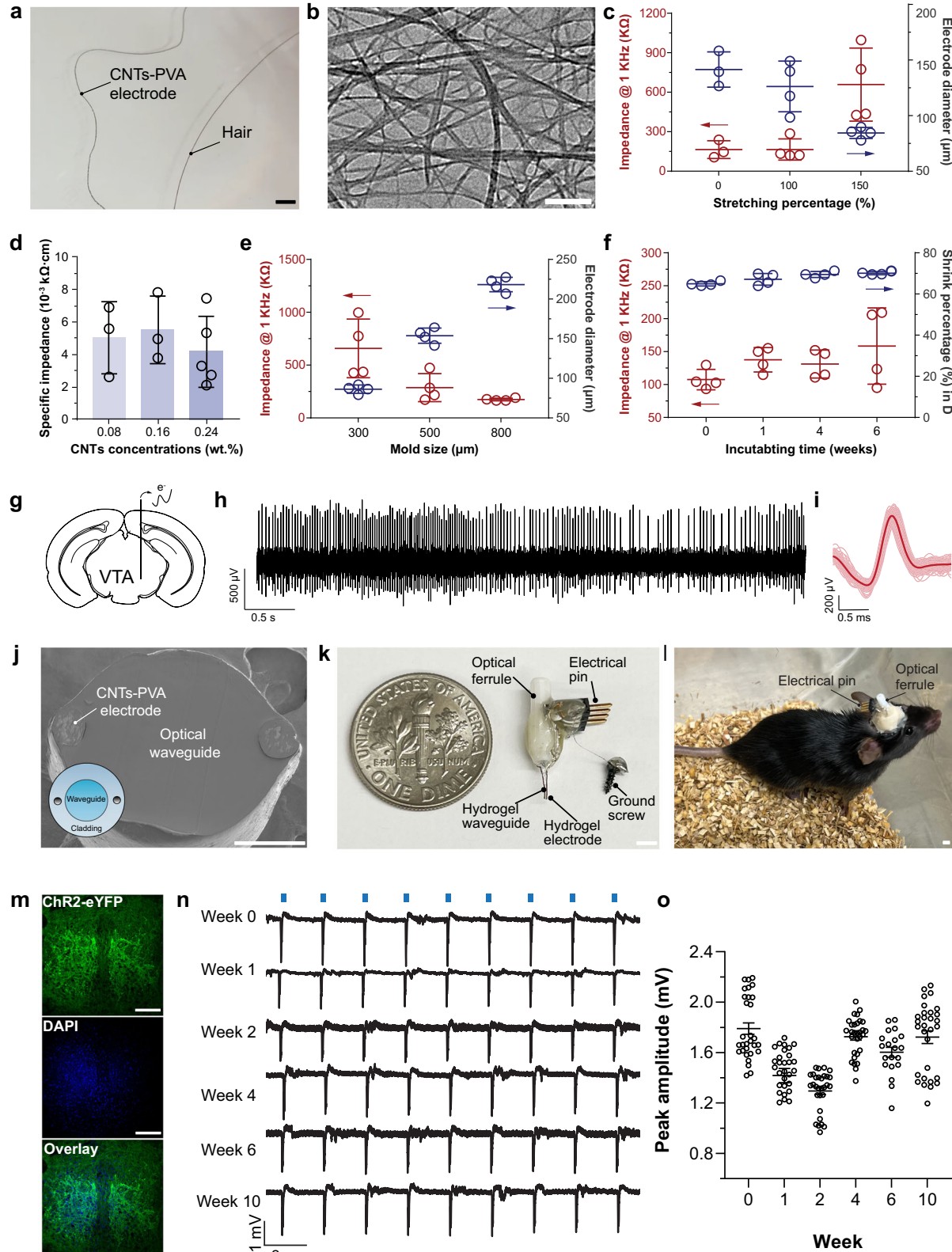

ARV-310, 2000 rpm, and 16 kPa vacuum) was utilized for the mixing and degassing of all solutions. For degassing and mixing, 100 µL of GA was added to 10 g of 10 wt. % PVA pre-solution and agitated for 1 min. 10 g of pre-made PVA-TEOS solution was also degassed and mixed for 1 min. Subsequently, the above two solutions were combined (weight ratio of 1:1) and mixed for another minute. The resulting PVA-TEOS-GA solution was infused into silicone tubes (McMaster-Carr 5236k204,

80 mm in length), and the optic ferrules were inserted into the silicone tubes, with the silica fiber end connected to the PVA mixture. After curing at room temperature for 4 h, the PVA-TEOS-GA fibers were demolded using dichloromethane (DCM, Sigma-Aldrich, 270997, 99.8%) and washed with a large amount of water to remove residual chemicals for two days. Ferrule-connected fibers were air-dried at room temperature for 12 h and annealed at 100 °C for 20 min. Finally,

**Fig. 4 | Integrated multifunctional hydrogel neural probes. a** A representative photograph of a carbon nanotubes (CNTs)-PVA hydrogel electrode as compared with a piece of human hair. Scale: 300 μm. **b** A transmission electron microscopy (TEM) image of CNTs (repeated 10 times with similar results). Scale: 200 nm. **c** Impedance at 1 kHz and diameters of the electrodes fabricated with different stretching percentages (mean ± s.d., $n$ = 3–4 independent hydrogel electrodes). **d** Impedance at 1 kHz of electrodes fabricated with different CNTs concentrations (mean ± s.d., $n$ = 3-5 independent hydrogel electrodes). **e** Impedance at 1 kHz of electrodes and diameters of the electrode fabricated with different sizes of molds (mean ± s.d., $n$ = 4 independent hydrogel electrodes). **f** Stability assessment on impedance and diameters of hydrogel electrodes incubated in PBS at 37 °C (mean ± s.d., $n$ = 4 independent hydrogel electrodes). **g** A schematic illustration of electrical recordings from mouse VTA with a CNTs-PVA electrode bundle (5 electrodes). **h** Representative electrophysiology recording signals from mouse VTA

with CNTs-PVA hydrogel electrodes. **i** A representative sorted neural spiking signal. **j** A representative scanning electron microscopy (SEM) image at the cross-section of an integrated multifunctional neural probe containing a hydrogel optical core and two CNTs-PVA hydrogel electrodes (repeated 10 times with similar results). Scale: 100 μm. **k, l,** Photographs of a hydrogel optoelectronic device (optrode) before implantation and after implantation in a *Thy1::ChR2-EYFP* mouse brain. Scale: 2 mm. **m** Confocal images of the expression of ChR2-EYFP in the VTA region of the mouse. Scale: 50 μm. **n** Representative in vivo electrical signals recorded with optrodes upon optical stimulation (blue bars, λ = 473 nm, 0.5 Hz, pulse width 50 ms, 10 mW/mm²). **o** Amplitudes of electrical signals recorded with optical stimulation over 10 weeks post-implantation (λ = 473 nm, 0.5 Hz, pulse width 50 ms, 10 mW/mm², mean ± s.e.m., $n$ = 20–30 individual optically evoked peak measurements from 3 mice).

the hydrogel fibers were rehydrated with MilliQ water before use. The compositions of all fabricated fibers are listed in Table 2.

## Core-cladding optical probe fabrication

A vacuum planetary mixer (Musashi ARV-310, 2000 rpm, and 16 kPa vacuum) was employed for mixing and degassing of all solutions. The optical fiber probes were first dried and then re-inserted into silicone tubing (McMaster-Carr 51845K66) and reswelled in water. For the preparation of the core-cladding optical fiber probes, 100 μL of GA was added to 10 g of 5 wt. % PVA pre-solution, which was then degassed and mixed for 1 min. Additionally, 150 μL of HCl was added to 10 g of 5 wt. % PVA pre-solution, which was also degassed and mixed for 1 min. The two solutions were combined (weight ratio of 1:1) and mixed for 1 min. The resulting mixed solution was infused into the silicone tubing and allowed to cross-link for 4 h at room temperature. The core-cladding optical fiber probes were extruded by immersing them in DCM and stored in MilliQ water until further use.

## XRD characterization of hydrogel materials

X-ray scattering measurements were conducted using the SAXSLAB GANESHA 300XL instrument, equipped with a Dectris Pilatus 300K 2D CMOS photon counting detector (size: 83.8 × 106.5 mm²). A small-angle 2 mm beamstop was utilized for SAXS measurements, while a wide-angle 2 mm beamstop was employed for WAXS measurements. The exposure time was set at 600 s. The average size of the nanocrystalline domain was determined using Scherrer's equation, which is expressed as $D = \frac{k\lambda}{\beta \cos \theta}$, where $k$ is a dimensionless shape factor that

varies based on the actual shape of the nanocrystalline domain ($k$ = 1, approximating the spherical shape of the nanocrystalline domains), $\lambda$ is the wavelength of X-ray diffraction ($\lambda$ = 1.54 Å), θ is the peak of the Bragg angle, and $\beta$ is the full width at half maximum (FWHM) of the WAXS peaks. The d-spacing between nanocrystalline domains was calculated using $d = \frac{2\pi}{q_{max}}$, where $q_{max}$ is the q value at its maximum intensity from SAXS patterns. The FWHM ($\beta$) and $q_{max}$ were obtained by curve fitting of the WAXS and SAXS patterns, respectively, in Origin software (OriginLab Corporation). For representative WAXS and SAXS 2D patterns, each hydrogel fiber was repeated three times with similar results.

## DSC characterization of hydrogel materials

The degree of crystallinity of hydrogel fibers and materials was assessed using a DSC instrument (2920 TA instrument). The PVA hydrogels were analyzed in the desiccated state. A small quantity of sample (1–15 mg) was loaded into a crucible (TA instrument T81006) and inserted into a temperature-controlled DSC cell. A blank crucible served as a reference. The sample was heated from 30 °C to 300 °C in air, with a heating rate of 20 °C/min. The differential heat flow to the sample and reference was recorded by the instrument. To determine the melting fusion enthalpy of endothermic peaks, heat flow (mW) over sample weight (mg) was plotted against time (s). The areas of melting endothermic peaks were integrated using TA analysis software (TA Universal Analysis). The degree of crystallinity $\alpha$ was estimated using the equation: $\alpha = \frac{\triangle H_m}{\triangle H_m^*} \cdot 100\%$, where $\triangle H_m$ (J/g) was calculated from the integration of melting endothermic peaks and $\triangle H_m^*$ (150 J/g) was the enthalpy of melting 100% of PVA crystallites. For representative DSC data, each sample was repeated three times with similar results. The crystallinity outcomes of PVA samples are presented in Supplementary Table 1.

## Hydrogel refractive index measurement

A series of hydrogel membranes were prepared via spin-coating using a spin-coating instrument (SETCAS, KW-4A) on silicon (Si) substrates (University Wafer, Inc., Model 447). The Si substrates were cut into square wafers (13.5 mm × 17.5 mm) using a diamond cutter and then subjected to a rigorous cleaning process. The cleaning process involved washing and ultrasonication in Acetone (Sigma-Aldrich 179124, 99.5%) for 3 min, followed by rinsing with MilliQ water. The Si wafers were then washed and ultrasonicated in 30 wt. % $H_2SO_4$ solution (Fisher Chemical 210524, 95.0%) for 3 min, followed by rinsing with MilliQ water. Finally, the Si wafers were washed and ultrasonicated in 10 wt. % of $H_2O_2$ solution (Sigma-Aldrich 216763, 30 wt. % in water) for 3 min, followed by rinsing with 95% ethanol (Fisher Chemical A962P4, 95.0%). The Si wafers were mounted on the spin coater and coated with 10P-GA, 10P-1T-GA, 10P-2T-GA, 10P-3T-GA, and 10P-4T-GA membranes ($n$ = 4 for each group) at 1000 rpm for 10 s, and at 5000 rpm 50 s. PVA solutions used for the membranes were prepared using the same method as discussed

### Table.1 | TEOS and PVA concentrations of PVA-TEOS solutions

| TEOS: HCl: H₂O (molar ratio) | TEOS wt. % in PVA pre-solutions | HCl wt. % in solutions | PVA wt. % in solutions |
|---|---|---|---|
| 1: 4: 4 | 2 | 0.014 | 10 |
| 2: 4: 8 | 4 | 0.014 | 10 |
| 3: 4: 12 | 6 | 0.014 | 10 |
| 4: 4: 16 | 8 | 0.014 | 10 |

### Table 2 | TEOS and PVA concentrations in PVA-TEOS-GA fibers

| Nomenclature | TEOS: HCl: H₂O (molar ratio) | TEOS wt. % in fibers | HCl wt. % in fibers | GA wt. % in fibers | PVA wt. % in fibers |
|---|---|---|---|---|---|
| 10P–1T-GA | 1: 4: 4 | 1 | 0.007 | 0.005 | 10 |
| 10P-2T-GA | 2: 4: 8 | 2 | 0.007 | 0.005 | 10 |
| 10P-3T-GA | 3: 4: 12 | 3 | 0.007 | 0.005 | 10 |
| 10P–1T-GA | 4: 4: 16 | 4 | 0.007 | 0.005 | 10 |

above. After spin-coating, the PVA membrane-coated Si wafers were allowed to cross-link and dry in the air for at least 12 h and then annealed at 100 °C for 20 min. The refractive index (RI) of the PVA membrane-coated Si wafers was measured using an ellipsometer (J.A. Woollam RC2) in the range of 400–700 nm. The measurements were carried out on the membranes in their desiccated states. A series of COMPACT hydrogel membranes (0–4 wt. % TEOS) were prepared using a similar procedure as described above but using a rectangular mold (21.5 × 21.5 × 1 mm). The membranes were demolded after cross-linking, dried at room temperature for 12 h, and cut into small sheets (2 × 2 mm). The sheets were then annealed at 100 °C for 20 min and reswelled in MilliQ water for 1 h. The RI of the membranes in their hydrated states was measured using a refractometer (Sper Scientific 300034) with water used for calibration.

## Hydrogel absorbance and fluorescence measurement

A set of hydrogel membranes (designated as 10P-GA, 10P-1T-GA, 10P-2T-GA, 10P-3T-GA, and 10P-4T-GA, comprising 4 replicates for each group) were synthesized and cross-linked in a 96-well plate using established techniques. Subsequently, 1 mL of PVA solution was added to each well and allowed to cross-link and air dry for at least 12 h, followed by annealing at 100 °C for 20 min. Rehydration of the membranes was achieved by the addition of 100 μL of MilliQ water to each well. To obtain transmittance spectra in the range of 400–700 nm, the 96-well plate was subjected to analysis using a plate reader (Biotek Synergy 2). Autofluorescence measurements were acquired using excitation/emission wavelengths of 470 nm/510 nm and 485 nm/520 nm, respectively. Membrane thickness was determined by caliper measurements and recorded three times to normalize the transmittance spectra and autofluorescence readings with respect to thickness. A blank control consisting of 200 μL of MilliQ water was included for comparison purposes.

## Mechanical characterization of hydrogel fibers

To ensure consistency, all hydrogel fibers were hydrated prior to the extension test. Tensile tests were conducted using a tensile instrument equipped with a 50 N load cell (Stable Micro System TA, XT plusC). The fibers were stretched at a constant rate of 1 mm/s. The nominal stress was calculated from the formula $\sigma = \frac{F}{A}$, where $F$ represents the force measured by the instrument, and $A$ represents the cross-sectional area of the fibers in their hydrated state. The strain was calculated using $\varepsilon = \frac{\Delta L}{L}$, where $\Delta L$ represents the displacement and $L$ represents the initial gauge length. Two marks were labeled on the fibers using a sharpie pen to determine the initial gauge length $L$ prior to the tensile test. A high-resolution camera was used to capture the entire tensile process and track displacement. The stress-strain curve was generated based on the calculated nominal stress and strain. The elastic moduli (E) were determined by calculating the average slope of the stress-strain relationship in the first 10% of applied strain. The average slope was determined by linear regression analysis (OriginLab Corporation). The stretchability of the fibers was reported as a percentage of the strain at the fracture point obtained from the stress-strain curves. The bending stiffness of hydrogel and silica fibers was measured using a mechanical tester (CellScale, Univert) equipped with a three-point beam bending setup. A deflection amplitude of 100 μm within the frequency range of 0.5–4 Hz (heartbeat frequency range) was employed[65,66].

## Light attenuation of hydrogel fibers

The light transmission loss of hydrogel fibers was tested by the cutback method. Ferrule-connected hydrogel fibers were inserted into a plastic tube (5 cm in length and 3 mm in diameter) and injected with 1 wt. % agar gel to maintain their hydrated state. The ferrule was connected to a 470 nm LED light (Thorlabs M470F3) via an adapter (Thorlabs SM1FCM). The power (in dB) of transmitted light through the hydrogel fiber was measured using a power meter (Thorlabs, PM16-122). The original power reading was recorded, and a 5 mm interval of cutting was adapted. Starting from the far end of the ferrule, the output power was measured after each cut using a cutter. The attenuation coefficient ($\alpha$) was calculated using the formula $\alpha = \left(\frac{10^4}{L_1 - L_2}\right) \cdot \log\left(\frac{P_1}{P_2}\right)$, where $L_1$ and $L_2$ represent the original and cut lengths of the fiber in meters, respectively. $P_1$ and $P_2$ are the transmitted power readings before and after the cut, respectively.

## Dimension measurements of hydrogel fibers

Microscopic images of hydrogel fibers were captured using a bright field mode microscope (AmScope) in MilliQ water. Three distinct regions of each fiber, namely two ends and the middle part, were imaged. The diameter of each fiber was measured using ImageJ software, with nine measurements taken for each fiber. The length of the fibers was measured using a caliper, with three measurements taken for each fiber.

## SEM imaging

SEM was performed on dried samples using an FEI Magellan 400 XHR instrument. To analyze the cross-sectional morphology of the integrated hydrogel optrode probe, the probe was sectioned into thin pillars (0.1 mm in height) and subsequently mounted on carbon tape for imaging. For representative SEM images, each sample was repeated ten times with similar results.

## TEM imaging

The TME images were acquired under a transmission electron microscope (FEI Tecnai 12). The carbon nanotubes were diluted (1:10) in MilliQ water and deposited on a copper grid (Sigma-Aldrich, FCF200-Cu) for imaging. For representative TEM images, each sample was repeated ten times with similar results.

## Stability tests of hydrogel fibers

The fabricated COMPACT hydrogel fibers (3 wt. % TEOS) were incubated at 37 °C under physiological-like solutions (saline, ionic strength 305-310 mOsm, pH from 6.0 to 8.0) over 3 months to validate the stability of hydrogel materials. The dimensions of fiber were measured before and after the incubation and statistical analysis was performed on the dimensions between pre-incubation and post-incubation each week.

## Cell culture and biocompatibility tests

The HEK 293T cell line was a gift from F. Zhang (MIT) and P. Anikeeva (MIT). Detailed information of the cell line can be found here from the American Type Culture Collection (item number CRL-3216). HEK 293T cells were authenticated before receiving. HEK 293T cells were maintained in DMEM (with GlutaMax, Sigma-Aldrich, D5796) + 10% fetal bovine serum and seeded in a 24-well plate. COMPACT hydrogel fibers (3 wt. % TEOS) were incubated in DMEM for 24 h at 37 °C. Hydrogel-incubated DMEM was then added to the well plate and incubated for 24 h. Calcein-AM (green, 2 μL of 1 mg/mL per well, Sigma-Aldrich 17783) was added to indicate living cells, and ethidium homodimer-1 (red, 2 μL of 1 mg/mL per well, Sigma-Aldrich 46043) was added to indicate dead cells. A fluorescent microscope (Nikon TiU with SOLA Light Engine Gen III illumination hardware and PCO panda sCMOS camera) was used to take images of cells with and without hydrogel incubation. ImageJ was utilized to count living cells and dead cells. Cell death rate (%) was calculated by using the formula: $death\ rate\ (\%) = \frac{dead\ cell\ numbers}{total\ cell\ numbers} \cdot 100\%$.

## Electrochemical impedance spectroscopy (EIS) of COMPACT hydrogel electrodes

The impedance of COMPACT hydrogel electrodes was assessed using an Electrochemical working station (Princeton Applied Research,

PARSTAT 2273) by applying a sinusoidal driving voltage (10 mV, 10 Hz~1 MHz). Impedance spectra of COMPACT hydrogel electrodes were acquired in PBS solutions.

## Virus package

pAAV-hSyn-GCaMP6s-WPRE-SV40 was a gift from The Genetically Encoded Neuronal Indicator and Effector Project (GENIE) and D. Kim (Addgene viral preparation no. 100843-AAV9). AAV9-hSyn-GCaMP6s were prepared in Rao Lab at UMass Amherst with Beckman Coulter Ultracentrifuge Optima XL70 with VTi 50.1 rotor. Before use, the viral vector was diluted to a titer of $10^{12}$ transducing units per milliliter.

## In vivo hydrogel optical probe implantation into the mouse brain

Mice were anesthetized using 1.0% isoflurane administered in a chamber and subsequently secured onto a stereotactic frame (RWD Life Science) with a heating pad to maintain their body temperature. All surgical procedures were conducted in sterile conditions with 1% isoflurane used to maintain anesthesia. The Allen Brain Atlas was used to align the skull and determine the coordinates for viral injection and fiber implantation, specifically targeting the ventral tegmental area (VTA) at coordinates AP: −2.95 mm, ML: ±0.50 mm, DV: −4.80 mm. An opening was made in the skull using a micro drill (RWD Life Science) at the designated coordinates. A total of 600 nL of adeno-associated virus (AAV) carrying hSyn::GCaMP6s was injected into the target region via a micro syringe and pump (World Precision Instruments, Micro 4). The viral injection device was held in place in the VTA region for 15 min to facilitate virus diffusion. Following fiber probe insertion, the probes were lifted by 0.1 mm to accommodate the viral volume. Finally, the fiber probes were secured to the skull using an adhesive (Parkell, C&B METABOND) and reinforced using dental cement (Jet Set-4). The mice were monitored on the heating pad following the removal of isoflurane until they were fully awake. C57BL/6J mice ($n = 8$) were implanted with a hydrogel optical probe, detailed information is listed in Supplementary Table 5.

## In vivo optrode device implantation into the mouse brain

Mice were anesthetized with 1.0% isoflurane and placed on a stereotactic frame (RWD Life Science) equipped with a heating pad to maintain body temperature. Surgery was conducted under sterile conditions, and 1% isoflurane was continuously administered to maintain anesthesia. Allen Brain Atlas was utilized to align the skull and establish optrode device coordinates (VTA, AP: −3.00 mm, ML: + (or −) 0.45 mm, DV: −4.80 mm) based on the mouse brain atlas. Prior to optrode implantation, a ground screw was implanted (AP: −3.50 mm, ML: − (or +) 1.50 mm, DV: −0.20 mm) and cerebrospinal fluid was contacted with the screw. The optrode devices were fixed on the skull with adhesive (Parkell, C&B METABOND) and reinforced with dental cement (Jet Set-4). Following the removal of isoflurane, the mice were monitored on the heating pad until fully awakened. *Thy1::ChR2-EYFP* mice ($n = 5$) were implanted with optrode devices, detailed information is listed in Supplementary Table 5.

## Fiber photometry recording

Following a four-week recovery period, hSyn::GCaMP6s injected mice were tethered to a fiber photometry (FIP) system (RWD and Thorlabs. Inc) using silica fiber (with a core diameter of 400 μm and a numerical aperture of 0.5, Thorlabs FP400URT). The silica fiber was connected to the FIP system using an adapter (Thorlabs SM1SMA), and a ferrule (Thorlabs CF440) was fixed to the other end of the fiber. The ferrule was coupled to the implanted fiber probe using a connecting sleeve (Thorlabs ADAF1). The mice were placed in a custom-made chamber (20 × 20 × 20 cm) for social preference tests, and fluorescent signals were computed using custom-written Python code. To excite the fluorescent signal, a custom setup consisting of a 470 nm LED

(Thorlabs M470F3), a 405 nm LED (Thorlabs M405F3), and dichroic mirrors (Thorlabs DMLP425R) were used. Illumination periods were determined by detecting synchronization ON/OFF pulses for each LED, with each illumination containing pulses at 10 Hz. To eliminate moving artifacts, the fitted 470 nm signals were subtracted from the fitted 405 nm signals.

## Social behavioral assay

For all behavioral experiments, adult C57BL/6 mice implanted with optical fiber probes were utilized during the dark phase of the light/dark cycle and were given at least 30 min of acclimatization in the behavior chamber before testing. Adult male C57BL/6 mice aged 5−6 weeks were used as strangers, and tests were performed in a dark environment. A chamber box (20 × 20 × 20 cm) containing a social cage was utilized for social interactions. Subsequently, a strange mouse was introduced to the social zone, and the test mouse was exposed to the strange mouse and allowed to interact freely. Concurrently, GCaMP fluorescence changes were recorded during social tests. A dark-vision camera was installed above the social chamber to record video footage during the social tests. The time spent interacting and the distance of social interaction were analyzed using customized algorithms for social interaction assessment with DeepLabCut. The analyzed social interaction epochs were then correlated with GCaMP signals.

## Immunohistology

The mice were euthanized using the fatal plus (Vortech Pharmaceuticals, LTD) and transcardiac perfusion was carried out using 20 mL of PBS (Sigma-Aldrich P3813) solution followed by 20 mL of 4% paraformaldehyde (PFA, Sigma-Aldrich 8187151000) solution. The brains were then dissected from the bodies and fixed in 4% PFA solution at 4 °C overnight. After fixation, the brain tissues were treated with 30% sucrose in PBS for 2 days and subsequently frozen at −20 °C in an O.C.T. cube (21.5 × 21.5 × 22 mm) and sectioned on a cryostat (Leica CM1900) with a thickness of 20 μm. The sectioned tissues were then permeabilized with PBST (0.3% Triton-X-100 in PBS, Sigma-Aldrich 93443) for 15 min at room temperature and blocked with 1% bovine serum albumin in PBS (Sigma-Aldrich A9647) for 30 min prior to staining. Primary antibody solutions (Table 3) were applied to stain the tissues and incubated overnight at room temperature. After washing the tissues with PBS three times, secondary antibody solutions (Table 3) were applied and incubated at room temperature for 2 h. The tissues were then washed with PBS three times and mounted on glass slides. DAPI mounting medium (Southernbiotech, Fluoromount-G, Cat. No. 0100-01) was used to mount the cover glass on top of the glass slide with the sections. The slides were left to dry in the air at room temperature overnight before images were acquired using a confocal microscope (Leica SP2 and ZEISS LSM 880, confocal fluorescent imaging was repeated 10−30 times with similar results). The quantitative analysis (fluorescence intensity, total fluorescence area, and total cell number counts) was performed with ImageJ. Area analysis of antibody-labeled cells was performed by creating binary layers of the implantation sites using the threshold tool and quantified using the measurement tool in ImageJ[67].

## Electromyography

EMG signals were recorded from the gastrocnemius muscle with one reference needle electrode, one hydrogel working electrode (287 ± 14 μm), and one ground electrode. A 473 nm laser (200 mW/mm$^2$, 0.5 Hz, pulse width 50 ms) was used for transdermal optical stimulation. EMG data triggered by optogenetic activation were amplified (1000×, DAM50, World Precision Instruments), filtered (1–1000 Hz, DAM50, World Precision Instruments), and digitized at 10 kHz (DI-1100, DATAQ Instruments).

**Table 3 | Antibodies and dilutions for immunohistology**

| Primary antibody (dilution) | Secondary antibody (dilution) |
|---|---|
| Tissue assessment 14 days post-implantation | |
| GFAP (Mouse, Southern Biotech 12075-01, 1:1000) | Goat anti-Mouse (Alexa Fluor 555 Invitrogen A-21422, 1:500) |
| IBA1 (Rabbit, Invitrogen PA5–119231, 1:300) | Chicken anti-Rabbit (Alexa Fluor 488 Invitrogen A-21441, 1:200) |
| CD68 (Rat, Invitrogen 14-0681-82, 1:500) | Goat anti-Rat (Alexa Fluor 555 Invitrogen A-21434, 1:1000) |
| CD16/32 (Rat, Invitrogen 14-0161-82, 1:200) | Goat anti-Rat (Alexa Fluor 555 Invitrogen A-21434, 1:1000) |
| NeuN (Rabbit, Invitrogen PA5-78499, 1:200) | Chicken anti-Rabbit (Alexa Fluor 488 Invitrogen A-21441, 1: 200) |
| Tissue assessment 1 month post-implantation | |
| GFAP (Rabbit, Agilent Dako Z0334, 1:400) | Donkey anti-Rabbit (Alexa Fluor 488 Invitrogen A-21206, 1: 200) |
| IBA1 (Rabbit, Invitrogen PA5–119231, 1:400) | Chicken anti-Rabbit (Alexa Fluor 594 Invitrogen A-21442, 1:200) |

## In vivo electrophysiology

Electrophysiological recordings were performed by connecting the pin connectors of optrode devices to an extracellular amplifier (DAM50, World Precision Instruments). Endogenous electrophysiological activities were filtered within the frequency range of 300–3000 Hz and digitized at a sampling frequency of 40 kHz (PowerLab 4/20T, ADInstruments). Subsequent signal processing and analysis were conducted using ROSS Offline Spike Sorter[68]. Optical illumination was carried out using a 473 nm laser connected to the implanted optrode devices via a ferrule-sleeve-ferrule connecting system. The laser (10 mW/mm$^2$) was pulsed at a frequency of 0.5 Hz with a pulse width of 50 ms during optical stimulation. Signals were digitized at 10 kHz (DI-1100, DATAQ Instruments) and filtered between 1 and 1000 Hz. The amplitude and noise level of evoked potentials were assessed utilizing a MATLAB algorithm incorporating a bandpass filter ranging from 0.1 to 300 Hz.

## Statistical analysis

We conducted statistical analyses using GraphPad Prism version 10 and R Studio. Sample sizes were based on pilot and previous (similar types of) experiments and were not statistically predetermined. Initially, we applied the Shapiro–Wilk test to assess data normality. The data in all main figures met the normality criteria, enabling us to use parametric statistical tests. For comparing three or more independent groups, we employed a One-way Analysis of Variance (ANOVA) with Tukey's Honest Significant Difference (HSD) test for post-hoc multiple comparisons. To analyze unpaired data across two groups, we used the unpaired student's t-test. Similarly, we applied the paired student's $t$-test for paired datasets. We considered $P$-values less than 0.05 as significant. The specific significance levels are as follows: * for $0.01 \leq P < 0.05$, ** for $0.001 \leq P < 0.01$, *** for $0.0001 \leq P < 0.001$, and **** for $P < 0.0001$.

## Reporting summary

Further information on research design is available in the Nature Portfolio Reporting Summary linked to this article.

## Data availability

The data relevant to this study including the material characterizations, biocompatibility tests, fiber photometry recording results, and electrophysiological recording results are comprehensively detailed within the article and its Supplementary Information. Additionally, the original datasets have been made publicly accessible via the public repository figshare (https://figshare.com) and are available here: https://doi.org/10.6084/m9.figshare.25521286. Source data are provided with this paper.

## Code availability

The customized code for fiber photometry recordings and social interaction analysis is available at https://github.com/neurobiologylab.

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

 **14**

## Acknowledgements

We thank D. Kim and P. Anikeeva for the generous gifts of the plasmids and cell lines, Y. Liu for his assistance on electrochemical characterization, H. Kim for her assistance on mechanical property characterization, R. Chen for his thoughtful comments on our manuscript, and T. Liu, A. Koh, and K. Ye for their support on hydrogel synthesis and characterization. This work was funded in part by the National Institutes of Health (R00MH120279), National Science Foundation Faculty Early Career Development Program (CAREER, 2239030), Brain&Behavior Research Foundation Young Investigator Grant (29878), Faculty Startup Funds from Binghamton University and UMass Amherst, UMass Amherst Faculty Research Grant (P1FRG0000000295), UMass OTCV Technology Development Fund and Binghamton University S3IP Small Grant award (ADLG267). This work made use of the Binghamton University and UMass Amherst core facilities of Electron Microscopy, Light Microscopy, Raman, IR and XRF Spectroscopy, Roll-to-Roll Fabrication and Processing, and X-Ray Scattering, Analytical & Diagnostics Laboratory, Laboratory Animal Resources and Animal Care Service.

## Author contributions

S.H., S.R., and Q.W. initiated the concept, designed the overall experiments, and conducted the in vivo photometry and electrophysiological recordings. S.H., X.L., Q.W., and S.R. were responsible for designing the materials and fabrication methods. S.H., S.L., and S.R. focused on the X-ray characterization. S.H. carried out the characterization of materials and devices and analyzed the data. A.S. and S.R. conceptualized the design for the carbon nanotubes hydrogel fibers. C.G. and C.M. developed the Python code and LabVIEW software for the customized fiber photometry system. S.H. and W.C. performed the in vivo experiments. K.F. and J.X. conducted the DeepLabCut analysis on social interaction behaviors. S.H., S.R., and A.C. were in charge of designing the experiments for characterizing mechanical properties. E.H. executed the tissue sample processing, immunostaining, and confocal imaging. S.H., S.R., and Q.W. prepared the figures and wrote the manuscript with contributions from all authors.

## Competing interests

The authors declare no competing interests.
