## [Peer Review File · Nature Communications]

Control of Polymers' Amorphous-crystalline Transition Enables Miniaturization and Multifunctional Integration for Hydrogel BioelectronicsEditorial Note: Parts of this Peer Review File have been redacted as indicated to remove third-party material where no permission to publish could be obtained.

REVIEWER COMMENTS

Reviewer #1 (Remarks to the Author):

Huang et al. introduce a fabrication strategy to miniaturize the diameters of hydrogel fibers and demonstrate their application as neural probes. There are several major issues that need to be addressed, and it's not clear whether these issues can be resolved at a level need for publication in a high impact journals such as nature communications.

1. The target application is bioelectronics, but all the characterizations are done with pure hydrogel fibers. The authors should carry out the characterizations in Figure 1 and 2 with conductive hydrogel fibers by introducing CNTs.

2. When CNTs are introduced in the fabrication of neural probes, will CNTs affect the crystallinity and the oriented crystalline growth by stretching? The interaction between CNTs and PVA chains can be essential for controlled crystallization. Besides, will CNTs aggregate above the percolation threshold in the hydrogel fibers?

3. The explanation for the "crystalline-dominated hydrogel miniaturization" is unclear and lacks experimental results to support.

Specifically, what is the relation between TEOS, crystallinity, and reduced dimension? The authors should provide the crystalline sizes related to different amounts of TEOS.

Similarly, how can the acidification procedure suppress crystallinity? A quantitative comparison between hydrogels with and without acidification needs to be provided.

The authors claim that stretching can induce the formation of anisotropic nanostructures, but how would these anisotropic nanocrystals benefit the shrinking procedure? And what does the increase in nanocrystalline spacing in Figure 1k indicate?

4. COMPACT hydrogels are described to exhibit tissue-like elasticity, but their modulus is 34 MPa, which is much larger than the modulus of nerve tissues. The elastic modulus of these hydrogels should be compared with other available neural probes in the literature, and demonstrate that the modulus in the MPa range is low enough to minimize damage to neural tissues.

5. The manuscript will be more readable if the discussion section at the end can be integrated with previous sections.

Reviewer #2 (Remarks to the Author):

Huang et al. proposed hydrogel-based fibers for optically stimulating and electrically recording neural signals. The authors presented a method to control the properties of hydrogels, including their optical and mechanical characteristics. While previous hydrogel-based optical fibers expanded when swelling, the proposed process allows for maintaining the volume of hydrogels. Nevertheless, the reviewer has

major concerns about the hydrogel fiber developed in the manuscript.

1. The hydrogel structure proposed does not appear to offer advantages over previous implantable devices based on hydrogels. Its mechanical properties, such as a higher elastic modulus than normal hydrogels, present a severe disadvantage for an implantable device. Therefore, the immune response shown in Supplementary Figure 14 does not exhibit significant differences between the hydrogel fiber and silica fiber. Additionally, the immune response should be quantitatively compared.
2. The hydrogel fiber used for neural signal recording does not demonstrate convincing results. The recorded signals lack spontaneous activity, and the noise level appears to be very high. Moreover, there are fluctuations in the noise level and recorded signals over the implanted time, indicating a potential decrease in electrode performance over time.
3. The dimensions of the implanted fibers seem considerably larger compared to previously reported hydrogel fiber electrodes. Although the fiber can maintain its volume and be manufactured in a small form factor, its size and elastic modulus are quite large, which pose critical disadvantages for an implanted device.
4. The authors measured the GCaMP signal from socially interacting mice. However, no significant correlation was found between the GCaMP signal and the duration of social interaction.
5. In the past, several hydrogel-based optical fibers and electrodes have been developed. However, the reviewer was unable to identify any advantages of the proposed device as an implantable device.

Reviewer #3 (Remarks to the Author):

The manuscript presents an innovative fabrication strategy for hydrogel bioelectronics using metamorphic polymers' amorphous-crystalline transition (COMPACT). The approach enables the miniaturization of hydrogel fibers with multifunctional interrogation of neural circuits. The work also focuses on the fabrication of step-index hydrogel optical probes and CNTs-PVA hydrogel microelectrodes for optogenetics and photometric recordings in the mouse brain region of the ventral tegmental area (VTA) and hindlimb muscle electromyographic and brain electrophysiological recordings of light-triggered neural activities in transgenic mice expressing Channelrhodopsin-2 (ChR2). Overall, the manuscript was well prepared. The results were presented in a logical and easy-to-follow manner, with appropriate figures and tables to support the findings. The study is of high significance, as it presents a novel approach for constructing hydrogel bioelectronics with miniaturized fiber shape and multifunctional interrogation of neural circuits, which has potential applications in brain-machine interfaces and investigating complex neural circuits. Therefore, the manuscript is recommended for publication in Nature Communications, subject to minor revisions detailed as follows:

1. The manuscript could benefit from a more detailed discussion of the potential limitations and

challenges of the proposed approach. For instance, the study only demonstrates the fabrication of hydrogel fibers with a diameter decrease of about 80%. It would be helpful to discuss the potential challenges of achieving further miniaturization and the impact of the hydrogel properties on the device's performance.

2. The explanation of the social behavioral assessment and the results obtained from the optogenetics and photometric recordings in the mouse brain region of the ventral tegmental area (VTA) should be further elaborated, along with more detailed discussion of the significance of the findings and their potential implications for future research.

3. This work is related to the hydrophilic materials, polyvinyl alcohol (PVA) in particular. In fact, PVA is a versatile polymer that has been studied extensively for various applications. There might be a need to address the structure and properties of PVA when it is combined with other materials like biopolymer in terms of different applications (see <https://doi.org/10.1016/j.jobab.2022.01.002>; <https://doi.org/10.1016/j.jobab.2021.11.004>; and <https://www.nature.com/articles/s41467-022-32987-6>). A concise review is suggested to be added to the introduction section. More importantly, PVA hydrogels are amorphous polymers that can undergo an amorphous-crystalline transition under certain conditions. The as-proposed COMPACT strategy for controlling this transition involves the introduction of multiple cross-linkers and acidification treatment to facilitate oriented polymeric crystalline growth under mechanical stretching.

4. The mechanisms involved in oriented polymeric crystalline growth under mechanical stretching may need to be further depicted, and videos or cartoons may be added to facilitate the readers' understanding. When a polymer is mechanically stretched, the degree of crystallinity and the orientation of polymer chains depend on the stretching conditions such as the rate and magnitude of the applied force, temperature, and the molecular structure of the polymer. The oriented crystalline structures contribute to the mechanical properties of the polymer, such as strength and stiffness, and can be utilized in various applications such as fibers, films, and composites.

5. The abstract is well-written and effectively communicates the research findings. In the second sentence (Line 4), "microfabrication" should be hyphenated as "micro-fabrication." The "polymetric" in line 12 should be corrected to "polymeric."

6. The novelty of this work lies in the development of a new fabrication strategy, called COMPACT, for hydrogel bioelectronics with miniaturized fiber shape and multifunctional interrogation of neural circuits. To highlight the novelty, the authors could emphasize the use of metamorphic polymers (e.g., in the abstract), the achievement of miniaturized fiber shapes, and the multifunctional interrogation of neural circuits using the newly developed hydrogel bioelectronics. Additionally, the use of these materials for optogenetics and photometric recordings in the mouse brain and hindlimb muscle electromyographic and brain electrophysiological recordings of light-triggered neural activities in transgenic mice expressing Channelrhodopsin-2 (ChR2) could be emphasized.

Response to Referees

We are appreciative of the feedback from our peer reviewers. Their constructive comments have significantly contributed to enhancing our manuscript. In response to their concerns, we have expanded our research with additional experiments, analyses, and discussions. These improvements are thoroughly integrated into the revised manuscript.

To facilitate a clear understanding of our revisions, we have organized our responses as follows:

- **Reviewer Comments and Our Responses:** Each comment from the reviewers (presented in blue text) is followed by our corresponding response (in black text).
- **Referencing Figures and Tables:** To support our responses, we have used figures and tables, which are referenced as follows:
 - Response-Related Figures and Tables: Figures and tables accompanying our responses to the reviewers are labeled as Fig. R# or Table R#.
 - Revised Manuscript Figures: Figures mentioned within the main text of the revised manuscript are numbered as Fig. #.
 - Supplementary Information Figures: Figures included in the revised Supplementary Information are denoted as Supplementary Fig. S#.
- **Highlighted Text in the Manuscript:** For ease of identification, all text modifications in the revised manuscript are highlighted in red.

Responses to Reviewer# 1 comments

General comments:

Huang et al. introduce a fabrication strategy to miniaturize the diameters of hydrogel fibers and demonstrate their application as neural probes. There are several major issues that need to be addressed, and it's not clear whether these issues can be resolved at a level need for publication in a high impact journal such as nature communications.

Response to general comment:

We appreciate the reviewer's expectation of our work. In the revised manuscript, we have extensively performed additional experiments, analysis, and discussion, aiming to comprehensively present our control of metamorphic polymers' amorphous-crystalline transition (COMPACT) strategy for hydrogel bioelectronics miniaturization and integration. The key revisions of our manuscript are summarized below.

- **Analysis of CNTs-PVA Hydrogels:** We have conducted a thorough characterization of carbon nanotubes (CNTs)-integrated cross-linked polyvinyl alcohol (PVA) hydrogels, used as microelectrode materials. This includes studying the effects of various CNTs concentrations (0.08-0.24 wt. %) on aspects such as impedance at 1kHz, nanocrystalline sizes, elastic modulus, stretchability, and the maintenance of miniaturized dimensions under electrophysiology conditions (37°C, PBS). These additional results are now included in the Supplementary Information (**Supplementary Fig. S8-11 and 23-28**).
- **Crystallinity and Orientation Assessment:** We have comprehensively examined the changes in crystallinity percentage and crystalline orientation upon introducing CNTs into PVA hydrogels. This comparison spans pure COMPACT hydrogels and CNTs-PVA

hydrogels with varying CNTs concentrations (0.08 wt. %, 0.16 wt. %, and 0.24 wt. %). These additional results are now included in the Supplementary Information (**Supplementary Fig. S10, 11, and 26**).

- We have conducted new experiments to investigate the CNTs percolation effect on CNTs-PVA hydrogel impedance changes (**Fig. R6**).
- We conducted additional experiments to investigate the effect of TEOS (0-4 wt. %) concentration on the size and orientation of nanocrystalline structures (**Supplementary Fig. S2, 6, and 7**).
- Acidification Treatment Impact: During the revision, the effects of acidification treatment on the size and orientation of nanocrystalline in different hydrogels were investigated. A detailed comparison of hydrogels with and without acidification treatment covers aspects such as fiber diameter shrinkage ratio, elastic modulus, stretchability, crystallinity, and nanocrystalline size (**Supplementary Fig. S8-11 and Table R1**).
- Anisotropic Nanostructures Formation: New X-ray measurement experiments, including WAXS 2D pattern and SAXS, were employed to characterize the size and spacing of nanocrystalline structures resulting from stretching with CNTs-PVA hydrogel fibers. We compared these effects in various hydrogels, with and without acidification, and at different stretching levels (0%, 100%, and 200%). These results are now added in the Supplementary Information (**Supplementary Fig. S25 and 27**).
- Mechanical Properties of COMPACT Hydrogel Fibers: To better understand the behavior of fiber-shaped neural probes in dynamic brain environments, we performed additional 3-point beam bending tests. This allowed us to measure the axial bending stiffness of COMPACT hydrogel fibers and compare them with other neural probes in the literature, focusing on properties like Young's modulus, stretchability, and bending stiffness. We have updated the new results in the Supplementary Information (**Fig. 2e and Supplementary Fig. S15**).
- As suggested, we have merged the discussion section with the experimental results to streamline the manuscript and enhance its readability.

Comment 1:

The target application is bioelectronics, but all the characterizations are done with pure hydrogel fibers. The authors should carry out the characterizations in Figure 1 and 2 with conductive hydrogel fibers by introducing CNTs.

Response 1:

We are grateful for the reviewer's insightful feedback. In response, we have conducted a thorough re-evaluation of our conductive hydrogel fibers, now incorporating carbon nanotubes (CNTs), to align more closely with the intended bioelectronics application. The key aspects of our revised characterization are as follows:

1. Impedance Analysis with CNTs-PVA Hydrogel Fibers: We assessed the impedances of CNTs-PVA hydrogel fibers (**Fig. R1**), specifically looking at 10% PVA cross-linked with varying CNTs concentrations, under conditions of acidification and 200% axial stretch. These measurements were made at 1 kHz and normalized to the fiber cross-section. We observed a notable decrease in specific impedance as the CNTs concentration increased within the hydrogel matrix (One-way ANOVA, $F_{3,16}=34.75$, $***p<0.0001$). These results are now reflected in the main text (**Fig. 4d**).

Figure R1. Initial normalized impedance of CNTs-PVA hydrogels with different CNTs concentrations (One-way ANOVA, $F_{3,16}=34.75$, $***p<0.0001$). Each dot represents one individual hydrogel fiber sample, mean \pm s.d. (standard deviation).

2. Nanocrystalline Analysis: We used Differential Scanning Calorimetry (DSC) to study the nanocrystalline structures in CNTs-PVA hydrogel fibers (**Fig. R2**). Results indicated a decrease in polymeric crystallinity with the addition of CNTs, shifting from $16.1 \pm 1.0\%$ to $12.3 \pm 0.6\%$. These findings are now part of the Supplementary Information (**Supplementary Fig. S23**). Wide-Angle X-ray Scattering (WAXS) was employed to analyze the nanocrystal size in these hydrogels (**Fig. R3**). The nanocrystal size in PVA hydrogel reduced from 3.22 ± 0.08 to 2.75 ± 0.06 nm upon CNTs integration. These results are now added to the Supplementary Information (**Supplementary Fig. S24**).

Figure R2. a, Representative DSC profiles of CNTs-PVA hydrogel fibers with CNTs content varying from 0 to 0.24 wt.%. **b,** Crystallinity CNTs-PVA hydrogel fibers with CNTs content varying from 0 to 0.24 wt.% calculated from **a** (One-way ANOVA, n.s.: non-significant). Each dot presents one individual fiber sample, mean \pm s.d..

Figure R3. **a**, Representative WAXS spectra of CNTs-PVA hydrogel fibers with CNTs content varying from 0 to 0.24 wt.%. **b**, Nanocrystal size of CNTs-PVA hydrogel fibers with CNTs content varying from 0 to 0.24 wt.% calculated from **b** (One-way ANOVA, **** $p < 0.0001$). Each dot presents one individual fiber sample, mean \pm s.d..

- Mechanical Properties Assessment:** We examined how the addition of CNTs (0.16 wt%) altered the mechanical properties of COMPACT PVA hydrogels (**Fig. R4a**). The CNTs increased the elastic modulus to 39.4 ± 13.7 MPa, with a corresponding decrease in stretchability to $47.9 \pm 12.2\%$ (**Fig. R4b**), due to the inhibited nanocrystallization and CNTs network percolation. These findings are now updated to Supplementary Information (**Supplementary Fig. S28**).

Figure R4. **a**, Representative stress-strain curves of CNTs-PVA hydrogel fibers (0.00 wt.% CNTs and 0.16 wt.% CNTs). **b-c**, CNTs-PVA hydrogel fibers' (0.00 wt.% CNTs and 0.16 wt.% CNTs) mechanical properties of elastic modulus (red) and stretchability percentage (blue). Each dot represents one independent fiber sample. Unpaired student t-tests were used to determine the statistical significance of elastic modulus: ($F_{5,5}=4.460$, * $p=0.0431$) and stretchability: ($F_{5,5}=4.467$, *** $p=0.0004$), respectively.

- Long-term Stability in Physiological Environment:** We tested the stability of CNTs (0.16%)-PVA hydrogel fibers in PBS at 37°C over six weeks, monitoring impedance and diameter shrinkage (**Fig. R5**, also reflected as **Fig. 4f** in the main text). The hydrogel microelectrodes maintained stable impedance (133.7 ± 15.6 k Ω) and a consistent diameter shrinkage ratio ($67.9 \pm 1.1\%$).

Figure R5. Stability assessment on impedance (red dots) and diameters (blue dots) of hydrogel electrodes (mean \pm s.d.). Each dot represents one independent hydrogel electrode.

5. Conductivity under Mechanical Stress: We evaluated whether the CNTs-PVA hydrogels will maintain conductivity under complex mechanical conditions. The stability of CNTs (0.16%)-PVA hydrogel microelectrodes was tested under cyclic stretching in water (**Fig. R6**), with strains of 5%, 10%, and 20% over 3000 cycles. We observed that the impedance improved post cyclic stretching, indicating fatigue resistance and suitability for in vivo applications under complex mechanical conditions. These results indicate that CNTs-PVA hydrogel fibers exhibited fatigue-resistance properties, aligning with our previous report about chemically crosslinked semi-crystalline PVA hydrogels (*Liu, et.al, Nat. Methods, 2023*). It suggests the potential of CNTs-PVA hydrogels as microelectrodes for in vivo electrophysiological recordings in complex mechanical working conditions.

Figure R6. a, Plot of stress vs. cycle number of the CNTs-PVA fiber under cyclic tensile test. The applied strain amplitude in the cyclic tensile test is 10% and the total cycle number is 3000. **b**, Plot of stress vs. cycle number of the CNTs-PVA fiber under cyclic tensile test from cycle 1500 to 1504. Strain amplitude: 10%. **c**, Normalized impedance of CNTs-PVA fibers pre and post cyclic tensile tests under different strain amplitudes (5%, 10%, and 20%). Paired t-tests were used to determine statistical significance (5%: * $p=0.0338$, 10%: ** $p=0.0051$, and 20%: ** $p=0.0026$).

These comprehensive characterizations underscore the potential of CNTs-PVA hydrogel fibers for bioelectronics, particularly in terms of electrical and mechanical properties suitable for dynamic in vivo environments.

To emphasize this point, we added experimental results and discussions in the revised main text Page 14, line 19 to page 15 line 5:

“After introducing CNTs (0.08-0.24 wt. %), we observed decreased insignificantly changed nanocrystallinity (Supplementary Fig. 23 and Table. S3), nanocrystalline sizes (Supplementary Fig. 24 and 25), preserved anisotropic structures (Supplementary Fig. 9c, 26 and 27), increased elastic modulus (0.16 wt. % CNTs-PVA hydrogels, 39.4 ± 13.7 MPa, and decreased stretchability ($47.9 \pm 12.2\%$, Supplementary Fig. 28). CNTs-PVA hydrogel electrodes (86 ± 5 μm diameter, Fig. 4c), insulated with a viscoelastic coating of styrene-ethylene-butylene-styrene (SEBS) (Supplementary Note 5 and Supplementary Fig. 29), exhibited impedances of 658 ± 277 k Ω at 1kHz and impedance was tunable with designed mold sizes and CNTs loadings (Fig. 4d and e). Their impedance kept stable over 6 weeks of incubation in PBS (37 °C, Fig.4f).”

Comment 2:

When CNTs are introduced in the fabrication of neural probes, will CNTs affect the crystallinity and the oriented crystalline growth by stretching? The interaction between CNTs and PVA chains can be essential for controlled crystallization. Besides, will CNTs aggregate above the percolation threshold in the hydrogel fibers?

Response 2:

We thank the reviewer for the constructive feedback. We agree that the interaction between CNTs and PVA chains can affect polymeric crystallization. To investigate the impact of CNTs on the crystallinity and oriented crystalline growth in PVA hydrogels, we have carried out a series of experiments:

Using DSC, we systematically analyzed the change in crystallinity (%) upon adding CNTs into the PVA hydrogel matrix. Our findings show a decrease in crystallinity from $16.1 \pm 1.0\%$ to $12.3 \pm 0.6\%$ as CNTs concentration increased (**Fig. R2**). Additionally, WAXS was employed to assess the nanocrystal size in PVA hydrogels. We observed a reduction in nanocrystal size from 3.22 ± 0.08 to 2.75 ± 0.06 nm with the introduction of CNTs (**Fig. R3**). Furthermore, WAXS 2D patterns indicated oriented crystalline growth in CNTs-PVA hydrogels along the axial stretching direction (**Fig. R7**), similar as the COMPACT hydrogels without adding CNTs. However, as the CNTs concentration rose, we noted a decrease in crystallinity percentage and a diminished effect of crystalline orientation. These results have been added into the revised manuscript Supplementary Information (**Supplementary Fig. S23, 24 and 26**).

Regarding the percolation threshold, we understand from previous studies (*Grieder, et. al, Percolation in Carbon Nanotube Networks, 2020; Bauhofer, et. al., A review and analysis of electrical percolation in carbon nanotube polymer composites, Composites science and technology 2009*) and our experimental results that the conductivity of CNTs-PVA hydrogels arises from the percolation effect of CNT fillers. Our experiments confirmed that the CNT concentration of 0.16 wt. % is above the percolation threshold. We further subjected CNTs-PVA hydrogel fibers to cyclic stretching tests (5%, 10%, and 20% strain over 3000 cycles, **Fig. R8**) and analyzed the impedance before and after stretching. The impedance of CNTs-PVA hydrogel microelectrodes was observed to decrease from 4.6 ± 1.0 to 4.0 ± 0.9 k $\Omega \cdot \text{mm}$. This finding supports the notion that CNTs within the PVA hydrogel matrix are above the percolation threshold, enabling effective conductivity.

Figure R7. Representative 2-dimensional (2D) WAXS spectra (vertical and horizontal measurement) collected from CNTs-PVA hydrogel fibers with 0.00 wt.%, 0.08 wt.%, 0.16 wt.%, and 0.24 wt.%, respectively.

Figure R8. **a**, Plot of stress vs. cycle number of the CNTs-PVA fiber under cyclic tensile test. The applied strain amplitude in the cyclic tensile test is 10% and the total cycle number is 5000. **b**, Plot of stress vs. cycle number of the CNTs-PVA fiber under cyclic tensile test from cycle 1500 to 1504. Strain amplitude: 10%. **c**, Normalized impedance of CNTs-PVA fibers pre and post cyclic tensile tests under different strain amplitudes (5%, 10%, and 20%). Paired t-tests were used to determine statistical significance (5%: * $p=0.0338$, 10%: ** $p=0.0051$, and 20%: ** $p=0.0026$).

To show the detailed results, we included experimental results and discussions in the revised main text Page 14, line 19 to page 14 line 23:

“After introducing CNTs (0.08-0.24 wt. %), we observed decreased insignificantly changed nanocrystallinity (Supplementary Fig. 23 and Table. S3), nanocrystalline sizes (Supplementary Fig. 24 and 25), preserved anisotropic structures (Supplementary Fig. 9c, 26 and 27), increased elastic modulus (0.16 wt. % CNTs-PVA hydrogels, 39.4 ± 13.7 MPa, and decreased stretchability ($47.9 \pm 12.2\%$, Supplementary Fig. 28).”

Comment 3:

The explanation for the “crystalline-dominated hydrogel miniaturization” is unclear and lacks experimental results to support. Specifically, what is the relation between TEOS, crystallinity, and reduced dimension? The authors should provide the crystalline sizes related to different amounts

of TEOS. Similarly, how can the acidification procedure suppress crystallinity? A quantitative comparison between hydrogels with and without acidification needs to be provided. The authors claim that stretching can induce the formation of anisotropic nanostructures, but how would these anisotropic nanocrystals benefit the shrinking procedure? And what does the increase in nanocrystalline spacing in Figure 1k indicate?

Response 3:

We appreciate the reviewer's constructive critique and have undertaken additional experiments to clarify and support our explanation for the "crystalline-dominated hydrogel miniaturization" process. Our investigations focused on understanding the relationship between TEOS concentration, crystallinity, and dimensional reduction.

TEOS Concentration and Crystallinity: In response to the reviewer's suggestion, we assessed the effect of varying TEOS concentrations (0, 3 wt. %, and 4 wt. %) on nanocrystal size and crystallinity in COMPACT hydrogels. Using WAXS and DSC, we found that increasing TEOS concentration led to a reduction in nanocrystal size (from 3.53 ± 0.02 to 3.11 ± 0.09 nm, **Fig. R9**) and suppressed crystallinity (from $21.6 \pm 1.1\%$ to $12.7 \pm 1.5\%$, **Fig. R10**). This effect can be attributed to TEOS acting as an additional crosslinker, limiting the mobility of PVA polymer chains and thereby hindering their folding into crystalline structures. These results have been added into the revised manuscript Supplementary Information (**Supplementary Fig. S2 and 6**).

Figure R9 . a, Representative WAXS curves of COMPACT hydrogel fibers including different TEOS contents (0, 3% and 4%, with acidification and 200% stretching). **b**, Nanocrystal size of COMPACT hydrogel fibers calculated from a (One-way ANOVA and Tukey's multiple comparisons tests, ***p=0.0009). Each dot represents one independent fiber sample, mean ± s.d..

Figure R10. a, DSC curves of COMPACT hydrogel fibers with different TEOS contents (0%, 3% and 4%). d, crystallinity (%) of COMPACT hydrogel fibers with different TEOS contents (0%, 3% and 4%) calculated from b (One-way ANOVA and Tukey’s multiple comparisons tests, ***p=0.0003). Each dot represents one independent fiber sample, mean \pm s.d..

These results have been updated to our revised main text Page 8 line 21 to Page 9 line 4:

GA-cross-linked PVA hydrogels exhibited $21.6 \pm 1.1\%$ crystallinity while the additional TEOS cross-linking and acidification suppressed the polymer chain folding to form crystalline domains (crystallinity: $12.7 \pm 1.5\%$, Supplementary Fig. 1, 2, and Table. S2). We further examined the nanocrystalline domains and orientation with X-ray scattering techniques. The size of PVA nanocrystals was measured as 3.5 ± 0.1 nm while the nanocrystalline spacing increased from 8.32 ± 0.08 nm to 9.83 ± 0.38 nm after 200% axial stretching (Fig. 1k and Supplementary Fig. 3-7).

Impact of Acidification on Crystallinity: We evaluated the effect of acidification on crystallinity in COMPACT hydrogels. Our comparative analysis (**Table R1**) between hydrogels with and without acidification revealed that acidification plays a key role in dimension reduction. The hydrogels maintained similar nanocrystal sizes but exhibited higher elastic modulus and decreased stretchability upon acidification. This trend was consistent across different hydrogel types, including GA cross-linked PVA hydrogels, TEOS-GA cross-linked PVA hydrogels, and CNTs-PVA hydrogels. Acidification generally suppressed nanocrystalline size, further reduced by the addition of TEOS and CNTs (**Fig. R11**). Despite these changes, the anisotropic orientation of nanocrystalline structures was preserved (**Fig. R12-13**). We have updated these results in the Supplementary Information (**Supplementary Fig. S8, 10 and 11**).

Table R1. Quantitative comparison between hydrogels with and without acidification		
	PVA hydrogel (3% TEOS, with acidification)	PVA hydrogel (3% TEOS, without acidification)
Diameter shrinking ratio (%)	80.78 ± 1.84	28.63 ± 3.81
Elastic modulus (MPa)	34.03 ± 7.38	3.78 ± 1.47
Stretchability (%)	94.46 ± 18.22	126.23 ± 29.21
Crystallinity (%)	16.09 ± 1.00	20.32 ± 1.79

Nanocrystal size (nm)	3.22 ± 0.08	3.28 ± 0.10
-----------------	-----------------

Figure R11. a, WAXS curves of different sample groups with or without acidification treatment. Group 1 (black): TEOS (-) and CNTs (-) with and without acidification; group 2 (red), TEOS (+) and CNTs (-) with and without acidification; group 3 (blue), TEOS (+) and CNTs (+) with and without acidification. **b**, Calculated nanocrystal size from a (Unpaired student's t-test, group 1(black): *p=0.0260; group 2 (red): n.s., group 3 (blue): *p=0.0174). mean ± s.d., each dot presents one individual fiber sample.

Figure R12. a, Representative 2D WAXS spectra (vertical measurement) collected from PVA hydrogel fibers (groups of TEOS (-) and CNTs (-), TEOS (+) and CNTs (-), and TEOS (+) and CNTs (+)) with and without acidification. **b**, Representative 2D WAXS spectra (horizontal measurement) collected from PVA hydrogel fibers (groups of TEOS (-) and CNTs (-), TEOS (+) and CNTs (-), and TEOS (+) and CNTs (+)) with and without acidification.

Figure R13. **a**, Representative 2D SAXS spectra (vertical measurement) collected from PVA hydrogel fibers (groups of TEOS (-) and CNTs (-), TEOS (+) and CNTs (-), and TEOS (+) and CNTs (+)) with and without acidification. **b**, Representative 2D SAXS spectra (horizontal measurement) collected from PVA hydrogel fibers (groups of TEOS (-) and CNTs (-), TEOS (+) and CNTs (-), and TEOS (+) and CNTs (+)) with and without acidification.

We have added a description in the main text Page 9 line 12 to page 9 line 14:

“Under the same cross-linking degree, acidification treatment granted polymer chains enhanced interactions and suppressed crystallinity (Fig. 1j and Supplementary Fig. 1c, 8, 9a-b, 10, and 11).”

Anisotropic Nanocrystals and Shrinking Procedure: To elucidate how anisotropic nanocrystals facilitate the shrinking process, we analyzed the orientation of nanocrystal domains under various strain conditions during hydrogel fiber preparation, using SAXS and WAXS 2D patterns (**Figs. R14-16**). Consistent with prior studies (*Lin, et.al, PNAS, 2019*), external stretching was found to increase the spacing between nanocrystalline structures, aligning them along the stretch direction. This anisotropic orientation proved beneficial in maintaining reduced hydrogel fiber diameters (**Fig. R17** and **Table R2**), which is important for minimizing axial bending stiffness and tissue damage in fiber-shaped bioelectronics. These results have been included to the revised manuscript Supplementary Information (**Supplementary Fig. S3a, 4, and 5** and **Fig. 1j**).

Figure R14. Representative SAXS profiles of 0% stretch GA cross-linked fibers, 200% stretch GA cross-linked fibers, and 200% stretched and acidified GA cross-linked fibers. d : spacing between nanocrystalline domains.

Figure R15. a, Representative 2D SAXS spectra (vertical measurement) collected from 0% stretch, 100% stretch, 200% stretch COMPACT (-), and COMPACT (+) hydrogels, respectively. **b,** Representative 2-dimensional SAXS spectra (horizontal measurement) collected from 0% stretch, 100% stretch, 200% stretch COMPACT (-), and COMPACT (+) hydrogels, respectively.

Figure R16. **a**, Representative 2-dimensional WAXS spectra (vertical measurement) collected from 0% stretch, 100% stretch, 200% stretch COMPACT (-), and COMPACT (+) hydrogels, respectively. **b**, Representative 2-dimensional WAXS spectra (horizontal measurement) collected from 0% stretch, 100% stretch, 200% stretch COMPACT (-), and COMPACT (+) hydrogels, respectively.

Figure R17. Shrinking behaviors of COMPACT hydrogel fibers (1-4 wt.% TEOS and 200% stretching). Hydrogel fibers' length (black) and diameter (red) are quantified as mean \pm s.d. Each dot represents one independent fiber.

Table R2. Shrinking profiles across different stretching strains.

TEOS content (wt. %)	0% stretch	100% stretch	200% stretch
0	48.83 \pm 1.43	60.58 \pm 2.43	74.22 \pm 1.84
1	49.36 \pm 2.04	65.53 \pm 1.66	75.02 \pm 2.33
2	49.53 \pm 2.30	66.67 \pm 2.49	76.93 \pm 1.19
3	50.29 \pm 1.96	69.83 \pm 1.46	80.78 \pm 1.84
4	50.37 \pm 1.00	67.95 \pm 2.87	79.74 \pm 2.30

We have detailed these results in the revised main text Page 9 line 2 to page 9 line 7:

“The size of PVA nanocrystals was measured as 3.5 ± 0.1 nm while the nanocrystalline spacing increased from 8.32 ± 0.08 nm to 9.83 ± 0.38 nm after 200% axial stretching (Fig. 1k and Supplementary Fig. 3-7). Wide-angle X-ray scattering (WAXS) 2D patterns suggested that the lamellae crystal domains were re-oriented along the axial stretching direction (Fig. 1k and Supplementary Fig. 5). Similar anisotropic nanocrystalline structures have been reported in other PVA hydrogels.”

Comment 4:

COMPACT hydrogels are described to exhibit tissue-like elasticity, but their modulus is 34 MPa, which is much larger than the modulus of nerve tissues. The elastic modulus of these hydrogels should be compared with other available neural probes in the literature, and demonstrate that the modulus in the MPa range is low enough to minimize damage to neural tissues.

Response 4:

We thank the reviewer’s comments and suggestions. We have conducted a detailed analysis to contextualize the mechanical properties of COMPACT hydrogels in relation to biological tissues and other neural probes.

We have compiled data from literature and our experiments on the Young’s modulus and water content of various soft materials, comparing them to biological tissues (**Fig. R18**). While the modulus of COMPACT hydrogels is higher than that of brain tissues, it falls within a range that is compatible with biological tissues. For biological applications especially in freely moving animals, the consideration of comprehensive mechanical properties besides Young’s modulus is needed. For instance, the stretchability and fatigue resistance of materials are important for adapting to tissue movements. COMPACT hydrogel fibers exhibited 95%-111% stretchability. In our and others’ previous work (*Liu, et. al, Nat. Methods, 2023; Lin, et.al, PNAS, 2019; Liu, et.al, Nat. Commun. 2020*), the chemically crosslinked PVA hydrogel fibers exhibited fatigue resistance, providing advantageous features for in vivo studies involved with freely moving experimental subjects.

Figure R18. Elastic modulus versus water contents for the all-hydrogel bioelectronic interface poly(3,4-ethylenedioxythiophene) and polystyrene sulfonate (PEDOT:PSS)¹, Ionic hydrogel², Peripheral nerve³, Heart³, Skin³, Tendon^{3, 4}, Polydimethylsiloxane (PDMS)^{5, 6, 7, 8, 9}, Styrene-Ethylene-Butylene-Styrene (SEBS)^{10, 11}, Poly(1,8-octanediol-co-citric acid) (POC)¹², Polyimide (PI)^{13, 14}, Polycarbonate (PC) and Cyclic olefin copolymer (COC)^{15, 16}, Poly(etherimide) (PEI)¹⁵, (epoxy-based negative photoresist) SU-8¹³, Poly(glycolic acid) (PGA)^{18, 19}.

Ref:

1. Javadi M, *et al.* Conductive Tough Hydrogel for Bioapplications. *Macromolecular Bioscience* **18**, 1700270 (2018).
2. Yang C, Suo Z. Hydrogel iontronics. *Nature Reviews Materials* **3**, 125-142 (2018).
3. Guimarães CF, Gasperini L, Marques AP, Reis RL. The stiffness of living tissues and its implications for tissue engineering. *Nature Reviews Materials* **5**, 351-370 (2020).
4. Yang S, *et al.* Stretchable surface electromyography electrode array patch for tendon location and muscle injury prevention. *Nature Communications* **14**, 6494 (2023).
5. Li Y, *et al.* Achieving tissue-level softness on stretchable electronics through a generalizable soft interlayer design. *Nature Communications* **14**, 4488 (2023).
6. Liu S, Rao Y, Jang H, Tan P, Lu N. Strategies for body-conformable electronics. *Matter* **5**, 1104-1136 (2022).
7. Liu Y, *et al.* Soft and elastic hydrogel-based microelectronics for localized low-voltage neuromodulation. *Nature Biomedical Engineering* **3**, 58-68 (2019).
8. Yoon Y, *et al.* Neural probe system for behavioral neuropharmacology by bi-directional wireless drug delivery and electrophysiology in socially interacting mice. *Nature Communications* **13**, 5521 (2022).
9. Zhou T, *et al.* 3D printable high-performance conducting polymer hydrogel for all-hydrogel bioelectronic interfaces. *Nature Materials* **22**, 895-902 (2023).
10. Sahasrabudhe A, *et al.* Multifunctional microelectronic fibers enable wireless modulation of gut and brain neural circuits. *Nature Biotechnology*, (2023).
11. Xing Y, *et al.* Integrated opposite charge grafting induced ionic-junction fiber. *Nature Communications* **14**, 2355 (2023).
12. Yang J, Webb AR, Ameer GA. Novel Citric Acid-Based Biodegradable Elastomers for Tissue Engineering. *Advanced Materials* **16**, 511-516 (2004).
13. Shen K, Chen O, Edmunds JL, Piech DK, Maharbiz MM. Translational opportunities and challenges of invasive electrodes for neural interfaces. *Nature Biomedical Engineering* **7**, 424-442 (2023).
14. Song E, Li J, Won SM, Bai W, Rogers JA. Materials for flexible bioelectronic systems as chronic neural interfaces. *Nature Materials* **19**, 590-603 (2020).

15. Canales A, *et al.* Multifunctional fibers for simultaneous optical, electrical and chemical interrogation of neural circuits in vivo. *Nature Biotechnology* **33**, 277-284 (2015).
16. Lu C, *et al.* Polymer Fiber Probes Enable Optical Control of Spinal Cord and Muscle Function In Vivo. *Advanced Functional Materials* **24**, 6594-6600 (2014).
17. Lu C, *et al.* Flexible and stretchable nanowire-coated fibers for optoelectronic probing of spinal cord circuits. *Science Advances* **3**, e1600955.
18. Djordjevic I, Choudhury NR, Dutta NK, Kumar S. Poly[octanediol-co-(citric acid)-co-(sebacic acid)] elastomers: novel bio-elastomers for tissue engineering. *Polymer International* **60**, 333-343 (2011).
19. Sundback CA, *et al.* Biocompatibility analysis of poly(glycerol sebacate) as a nerve guide material. *Biomaterials* **26**, 5454-5464 (2005).

We appreciate the opportunity to elaborate on an additional critical aspect of our COMPACT strategy for fiber-shaped neural probes: the axial bending stiffness and its role in controlling the neural probe footprint.

Axial Bending Stiffness Consideration: Beyond Young’s modulus and stretchability, axial bending stiffness is a crucial parameter for fiber-shaped neural probes, especially in vivo applications involving freely moving subjects. Typically, one end of the neural probe is anchored to the skull for external stimulation and recording, while the other end is inserted into brain tissue. In such scenarios, the brain’s movement during the subject’s activities positions the neural probe in a cantilever beam configuration (**Fig. R19**). The axial bending stiffness, k , of the neural probe, is determined by the formula as $k = \frac{A_{cs}E}{L}$ (A_{cs} : area of the probe cross-section; E : Young’s modulus; L : length). The calculations indicate that reducing the hydrogel fiber footprint (A_{cs}) significantly lowers axial bending stiffness.

We conducted 3-point beam bending tests to measure the axial bending stiffness of COMPACT neural probes, comparing them with commercially available silica-based optical probes (**Fig. R20**), which are widely used in optogenetics and fiber photometry. These findings are now included in the revised **Fig. 2e** of our manuscript. Additionally, we have compared the axial bending stiffness of our COMPACT neural probes with other advanced neural probes (**Fig. R21**). This comparison underscores the benefit of the COMPACT strategy in reducing the diameter of neural probes, thereby minimizing tissue damage and the buckling force required for probe implantation. The buckling force, ($F_{buckling} = \frac{\pi^2 IE}{(KL)^2}$, $F_{buckling}$ is the critical buckling force, I is the geometry-dependent area moment of inertia, E is the Young’s modulus, L is the length of the probe, and K is the effective length factor).

Considering all these factors—Young’s modulus, device footprint, bending stiffness under dynamic conditions, and ease of implantation—we believe that the COMPACT strategy offers an effective and balanced approach for the miniaturization and integration of hydrogel bioelectronics. This strategy not only aligns with the mechanical requirements of in vivo applications but also minimizes potential damage to neural tissues, enhancing the overall functionality and safety of the neural probes.

Figure R19. a, An illustration of a hydrogel fiber inserted into a brain region, anchored at one end to the skull. This design allows the fiber to flexibly adapt to brain movements, facilitated by its low bending stiffness. **b**, A schematic of the cantilever beam bending model, which provides a framework for assessing the bending stiffness of the hydrogel fiber.

Figure R20. a, bending stiffness of COMPACT hydrogel fibers, CNTs-PVA hydrogel fibers, and silica fibers (200 μm). **b**, statistical comparison of bending stiffness (One-way ANOVA, **** $p < 0.0001$). Each dot represents one independent fiber sample, mean \pm s.d..

Figure R21 Axial bending stiffness of various soft material-based neural probes. Probes with $k < 100$ N/m are considered compliant, while probes with $k > 1000$ N/m are considered stiff^{1, 2, 3, 4, 5, 6, 7}. Ref:

1. Nguyen JK, *et al.* Mechanically-compliant intracortical implants reduce the neuroinflammatory response. *Journal of Neural Engineering* **11**, 056014 (2014).
2. Serruya MD, *et al.* Engineered Axonal Tracts as “Living Electrodes” for Synaptic-Based Modulation of Neural Circuitry. *Advanced Functional Materials* **28**, 1701183 (2018).
3. Park S, *et al.* One-step optogenetics with multifunctional flexible polymer fibers. *Nature Neuroscience* **20**, 612-619 (2017).
4. Park S, *et al.* Adaptive and multifunctional hydrogel hybrid probes for long-term sensing and modulation of neural activity. *Nature Communications* **12**, 3435 (2021).
5. Sahasrabudhe A, *et al.* Multifunctional microelectronic fibers enable wireless modulation of gut and brain neural circuits. *Nature Biotechnology*, (2023).
6. Garwood IC, *et al.* Multifunctional fibers enable modulation of cortical and deep brain activity during cognitive behavior in macaques. *Science Advances* **9**, eadh0974.
7. David-Pur M, Bareket-Keren L, Beit-Yaakov G, Raz-Prag D, Hanein Y. All-carbon-nanotube flexible multi-electrode array for neuronal recording and stimulation. *Biomedical Microdevices* **16**, 43-53 (2014).

To better explain the importance of the bending properties of our materials, we added description in **Page 11 line 14 to page 11 line 20**:

“When fiber-shaped neural probes are inserted into brain tissues, their axial bending stiffness serves as an important mechanical parameter under brain micromotions (Supplementary Fig. 15). Compared to silica fibers (~20 GPa elastic modulus) and polymer fibers (~1 GPa elastic modulus), COMPACT hydrogel fibers offer enhanced mechanical matching to the nervous tissues (1-4 kPa)^{47, 48} and much lower axial bending stiffness (Fig.2e) to achieve less neural tissue damage from micro-motion involved in vivo studies.”

Figure 2. Controllable hydrogel fiber fabrication and its properties. **a**, A shrinking diagram of COMPACT (+) hydrogel fibers. Each dot (mean \pm s.d.) represents an independent hydrogel fiber sample. The samples shaded in red areas are treated with acidification. **b**, Shrinking behaviors of COMPACT hydrogel fibers (4wt. % TEOS) prepared in different sizes of molds. Each dot (mean \pm s.d.) represents one independent fiber (One-way ANOVA and Tukey's multiple comparisons test, $F_{3,12}=0.9543$, n.s.: not significant, $p=0.4455$). **c**, COMPACT hydrogel fibers' optical properties of refractive index (blue) and normalized light transmittance (red) (mean \pm s.d.). Inset: representative photographs of 0 wt. % TEOS and 4 wt. % TEOS hydrogel membranes. Grid size: 1 mm. **d**, COMPACT hydrogel fibers' mechanical properties of elastic modulus (blue) and stretchability percentage (red). Each dot represents one independent fiber sample. One-way ANOVA and Tukey's multiple comparisons test were used to determine the statistical significance of elastic modulus: ($F_{4,15}=20.51$, **** $p<0.0001$;) and stretchability: ($F_{4,15}=1.492$, n.s. $p=0.2543$), respectively. **e**, Bending stiffness of COMPACT (3 wt. % TEOS) hydrogel fiber ($n=4$ independent samples) with identical cross-sections in comparison with a 200- μm silica waveguide ($n=3$ independent samples). **f**, Stability assessment of diameter reduction of COMPACT hydrogel

fibers (3wt. % TEOS). Each dot (mean \pm s.d.) represents one independent fiber (two-way ANOVA and Tukey's multiple comparisons tests). **g**, Cytotoxicity assessment of COMPACT (+) hydrogels. Hydrogel incubated media was used to culture with Human Embryonic Kidney (HEK) 293 cell cultures. Calcein-AM (green) was used to stain living cells and ethidium homodimer-1 (red) was used to stain dead cells. Cell death rates are presented as mean \pm standard error (s.e.m., unpaired student's t-test).

Comment 5:

The manuscript will be more readable if the discussion section at the end can be integrated with previous sections.

Response 5:

Thank you for the suggestion. In response to the reviewer's comment, we have revised the manuscript to integrate the discussion with the experimental results. This reorganization aims to enhance readability and provide a more seamless connection between our findings and their interpretation.

Responses to Reviewer 2's comments

General comments:

Huang et al. proposed hydrogel-based fibers for optically stimulating and electrically recording neural signals. The authors presented a method to control the properties of hydrogels, including their optical and mechanical characteristics. While previous hydrogel-based optical fibers expanded when swelling, the proposed process allows for maintaining the volume of hydrogels. Nevertheless, the reviewer has major concerns about the hydrogel fiber developed in the manuscript.

Response to general comment:

We appreciate the reviewer's comments and suggestions. We have significantly expanded our experimental investigations, incorporating a range of new studies and analyses to enhance the depth and scope of our manuscript. The major additions include.

- Mechanical Testing of Control of Metamorphic Polymers' Amorphous-Crystalline Transition (COMPACT) Hydrogel Fibers: We carried out additional 3-point beam bending tests to measure the axial bending stiffness of COMPACT hydrogel fibers. We compared these fibers with other neural probes available in the literature, detailing their mechanical properties such as Young's modulus, stretchability, and bending stiffness. We have updated these new results in the Supplementary Information (**Fig. 2e and Supplementary Fig. S15**).
- Immunohistology Assessments: We conducted immunohistology analyses of mouse brain tissues implanted with both COMPACT and silica fibers of similar dimensions (~400 μm). These assessments focused on astrocytes (GFAP) and activated microglia (Iba1) staining. Fluorescence microscopy and ImageJ-assisted analysis were employed to quantitatively compare the immune responses elicited by COMPACT hydrogel fibers and silica fibers. These new quantitative analyses have been updated in the Supplementary Information (**Supplementary Fig. S16**).
- In vivo Extracellular Electrophysiological Recording of Spontaneous Neural Activity: We performed electrophysiological recordings using CNTs-PVA hydrogel microelectrodes from the mouse ventral tegmental area (VTA) and successfully captured the spontaneous phasic bursting of VTA neurons. These additional results are now included in **Fig. 4h-i**.
- Stability and Efficacy Tests of CNTs-PVA Hydrogel Microelectrodes: The impedance of CNTs-PVA hydrogel microelectrodes was monitored over thousands of cycles of stretching with 0%-10% strains. In vivo tests were extended to 10 weeks, during which we recorded optically evoked neural electrical signals from *Thy1::ChR2::EYFP* mice. We have incorporated the latest results into the main text (**Fig. 4n-o**).
- To address the question about the COMPACT hydrogel fiber dimensions, we detailed the molding and extrusion procedures used to regulate hydrogel fiber diameters. The current hydrogel optical fibers with diameters of 400-600 μm are designed to cover sufficient brain tissues for mouse optogenetics and fiber photometry recordings in vivo. COMPACT strategy does not limit but rather facilitates the miniaturization of hydrogel fibers.
- Optimization of Behavioral Video Analysis: To enhance the quality of fiber photometry results during mouse social behaviors, we optimized our behavioral video analysis algorithm and increased the sample size of recording and behavioral tests. These additional results are now updated and reflected in **Fig. 3h**.
- The introduction of our revised manuscript has been improved to provide a more comprehensive overview of soft material bioelectronics.

Comment 1:

The hydrogel structure proposed does not appear to offer advantages over previous implantable devices based on hydrogels. Its mechanical properties, such as a higher elastic modulus than normal hydrogels, present a severe disadvantage for an implantable device. Therefore, the immune response shown in Supplementary Figure 14 does not exhibit significant differences between the hydrogel fiber and silica fiber. Additionally, the immune response should be quantitatively compared.

Response 1:

We thank the reviewer's comments and suggestions. In response to the concerns raised, we have further elaborated on the advantages and suitability of COMPACT hydrogel neural probes for in vivo applications, especially in scenarios involving freely moving experimental subjects.

Understanding the importance of mechanical properties for in vivo applications, we have conducted analysis of the COMPACT hydrogel neural probes, particularly focusing on their behavior under movement conditions (Fig. R22a). The cantilever beam model (Fig. R22b) represents the neural probe's positioning in the brain, highlighting the significance of axial bending stiffness in assessing potential brain tissue damage during movement. Despite a relatively higher Young's modulus yet still in the range to match biological tissues, COMPACT hydrogel neural probes exhibit satisfactory stretchability (Fig. R23 and Fig. 2d) and significantly lower bending stiffness compared to conventional silica fibers used in optogenetics and fiber photometry (Fig. R24 and Fig. 2e). Compared to the state-of-the-art neural probes (Fig. R25), COMPACT hydrogel neural probes exhibit sufficient low bending stiffness suitable for motion-involved working conditions. We have added new figures in Fig. 2e and Supplementary Fig. S15 to better explain the importance of the bending stiffness of our COMPACT hydrogel fibers for in vivo applications.

Figure R22. Analysis of hydrogel fiber mechanics. **a**, An illustration of a hydrogel fiber inserted into a brain region, anchored at one end to the skull. This design allows the fiber to flexibly adapt to brain movements, facilitated by its low bending stiffness. **b**, A schematic of the cantilever beam bending model, which provides a framework for assessing the bending stiffness of the hydrogel fiber.

Figure R23. Elastic modulus versus water contents for the all-hydrogel bioelectronic interface poly(3,4-ethylenedioxythiophene) and polystyrene sulfonate (PEDOT:PSS)¹, Ionic hydrogel², Peripheral nerve³, Heart³, Skin³, Tendon^{3, 4}, Polydimethylsiloxane (PDMS)^{5, 6, 7, 8, 9}, Styrene-Ethylene-Butylene-Styrene (SEBS)^{10, 11}, Poly(1,8-octanediol-co-citric acid) (POC)¹², Polyimide (PI)^{13, 14}, Polycarbonate (PC) and Cyclic olefin copolymer (COC)^{15, 16}, Poly(etherimide) (PEI)¹⁵, (epoxy-based negative photoresist) SU-8¹³, Poly(glycolic acid) (PGA)^{18, 19}.

Ref:

- Javadi M, *et al.* Conductive Tough Hydrogel for Bioapplications. *Macromolecular Bioscience* **18**, 1700270 (2018).
- Yang C, Suo Z. Hydrogel iontronics. *Nature Reviews Materials* **3**, 125-142 (2018).
- Guimarães CF, Gasperini L, Marques AP, Reis RL. The stiffness of living tissues and its implications for tissue engineering. *Nature Reviews Materials* **5**, 351-370 (2020).
- Yang S, *et al.* Stretchable surface electromyography electrode array patch for tendon location and muscle injury prevention. *Nature Communications* **14**, 6494 (2023).
- Li Y, *et al.* Achieving tissue-level softness on stretchable electronics through a generalizable soft interlayer design. *Nature Communications* **14**, 4488 (2023).
- Liu S, Rao Y, Jang H, Tan P, Lu N. Strategies for body-conformable electronics. *Matter* **5**, 1104-1136 (2022).
- Liu Y, *et al.* Soft and elastic hydrogel-based microelectronics for localized low-voltage neuromodulation. *Nature Biomedical Engineering* **3**, 58-68 (2019).
- Yoon Y, *et al.* Neural probe system for behavioral neuropharmacology by bi-directional wireless drug delivery and electrophysiology in socially interacting mice. *Nature Communications* **13**, 5521 (2022).
- Zhou T, *et al.* 3D printable high-performance conducting polymer hydrogel for all-hydrogel bioelectronic interfaces. *Nature Materials* **22**, 895-902 (2023).
- Sahasrabudhe A, *et al.* Multifunctional microelectronic fibers enable wireless modulation of gut and brain neural circuits. *Nature Biotechnology*, (2023).
- Xing Y, *et al.* Integrated opposite charge grafting induced ionic-junction fiber. *Nature Communications* **14**, 2355 (2023).
- Yang J, Webb AR, Ameer GA. Novel Citric Acid-Based Biodegradable Elastomers for Tissue Engineering. *Advanced Materials* **16**, 511-516 (2004).
- Shen K, Chen O, Edmunds JL, Piech DK, Maharbiz MM. Translational opportunities and challenges of invasive electrodes for neural interfaces. *Nature Biomedical Engineering* **7**, 424-442 (2023).
- Song E, Li J, Won SM, Bai W, Rogers JA. Materials for flexible bioelectronic systems as chronic neural interfaces. *Nature Materials* **19**, 590-603 (2020).
- Canales A, *et al.* Multifunctional fibers for simultaneous optical, electrical and chemical interrogation of neural circuits in vivo. *Nature Biotechnology* **33**, 277-284 (2015).
- Lu C, *et al.* Polymer Fiber Probes Enable Optical Control of Spinal Cord and Muscle Function In Vivo. *Advanced Functional Materials* **24**, 6594-6600 (2014).
- Lu C, *et al.* Flexible and stretchable nanowire-coated fibers for optoelectronic probing of spinal cord circuits. *Science Advances* **3**, e1600955.

18. Djordjevic I, Choudhury NR, Dutta NK, Kumar S. Poly[octanediol-co-(citric acid)-co-(sebacic acid)] elastomers: novel bio-elastomers for tissue engineering. *Polymer International* **60**, 333-343 (2011).
19. Sundback CA, *et al.* Biocompatibility analysis of poly(glycerol sebacate) as a nerve guide material. *Biomaterials* **26**, 5454-5464 (2005).

Figure R24. **a**, bending stiffness of COMPACT hydrogel fibers, CNTs-PVA hydrogel fibers, and silica fibers (200 μm). **b**, statistical comparison of bending stiffness (One-way ANOVA, **** $p < 0.0001$). Each dot represents one independent fiber sample, mean ± s.d..

Figure R25 Axial bending stiffness of various soft material-based neural probes. Probes with $k < 100$ N/m are considered compliant, while probes with $k > 1000$ N/m are considered stiff. 1, 2, 3, 4, 5, 6, 7

Ref:

1. Nguyen JK, *et al.* Mechanically-compliant intracortical implants reduce the neuroinflammatory response. *Journal of Neural Engineering* **11**, 056014 (2014).
2. Serruya MD, *et al.* Engineered Axonal Tracts as “Living Electrodes” for Synaptic-Based Modulation of Neural Circuitry. *Advanced Functional Materials* **28**, 1701183 (2018).
3. Park S, *et al.* One-step optogenetics with multifunctional flexible polymer fibers. *Nature Neuroscience* **20**, 612-619 (2017).
4. Park S, *et al.* Adaptive and multifunctional hydrogel hybrid probes for long-term sensing and modulation of neural activity. *Nature Communications* **12**, 3435 (2021).
5. Sahasrabudhe A, *et al.* Multifunctional microelectronic fibers enable wireless modulation of gut and brain neural circuits. *Nature Biotechnology*, (2023).
6. Garwood IC, *et al.* Multifunctional fibers enable modulation of cortical and deep brain activity during cognitive behavior in macaques. *Science Advances* **9**, eadh0974.
7. David-Pur M, Bareket-Keren L, Beit-Yaakov G, Raz-Prag D, Hanein Y. All-carbon-nanotube flexible multi-electrode array for neuronal recording and stimulation. *Biomedical Microdevices* **16**, 43-53 (2014).

A key feature of the COMPACT strategy is its ability to control the miniaturization of hydrogel bioelectronics. Smaller device footprints are crucial for minimizing tissue damage. For instance, reducing the diameter of COMPACT hydrogel fibers substantially lowers their axial bending stiffness, as described by the equation $k = \frac{A_{cs}E}{L}$ (A : area of the probe cross-section; E : Young's modulus; L : length). When we apply the COMPACT strategy to a hydrogel fiber originally measuring 5 mm in length and 800 μm in diameter and achieve an 80% reduction in diameter down to 160 μm , the bending stiffness of the hydrogel fibers is significantly reduced to 4%. Specifically, in applications targeting deep brain structures, such as the ventral tegmental area (DV -5 mm) in mice, this reduction in diameter results in a remarkable decrease in bending stiffness, which is an advantage for minimally invasive applications in sensitive neural regions. This reduction in diameter also directly decreases the buckling force required for neural probe implantation (a cylinder-shaped hydrogel neural probe: $I_{cyl} = \frac{\pi r^4}{4}$), calculated as $F_{buckling} = \frac{\pi^2 IE}{(kL)^2}$ ($F_{buckling}$ is the critical buckling force, I is the geometry-dependent area moment of inertia, E is the Young's modulus, L is the length of the probe, and K is the effective length factor).

In the revised manuscript, we updated the mechanical properties of our COMPACT hydrogel fibers in Page 11 line 14 to page 11 line 20:

“When fiber-shaped neural probes are inserted into brain tissues, their axial bending stiffness serves as an important mechanical parameter under brain micromotions (Supplementary Fig. 15). Compared to silica fibers (~20 GPa elastic modulus) and polymer fibers (~1 GPa elastic modulus), COMPACT hydrogel fibers offer enhanced mechanical matching to the nervous tissues (1-4 kPa)^{47, 48} and much lower axial bending stiffness (Fig.2e) to achieve less neural tissue damage from micro-motion involved in vivo studies.”

Figure 2. Controllable hydrogel fiber fabrication and its properties. **a**, A shrinking diagram of COMPACT (+) hydrogel fibers. Each dot (mean \pm s.d.) represents an independent hydrogel fiber sample. The samples shaded in red areas are treated with acidification. **b**, Shrinking behaviors of COMPACT hydrogel fibers (4wt. % TEOS) prepared in different sizes of molds. Each dot (mean \pm s.d.) represents one independent fiber (One-way ANOVA and Tukey's multiple comparisons test, $F_{3,12}=0.9543$, n.s.: not significant, $p=0.4455$). **c**, COMPACT hydrogel fibers' optical properties of refractive index (blue) and normalized light transmittance (red) (mean \pm s.d.). Inset: representative photographs of 0 wt. % TEOS and 4 wt. % TEOS hydrogel membranes. Grid size: 1 mm. **d**, COMPACT hydrogel fibers' mechanical properties of elastic modulus (blue) and stretchability percentage (red). Each dot represents one independent fiber sample. One-way ANOVA and Tukey's multiple comparisons test were used to determine the statistical significance of elastic modulus: ($F_{4,15}=20.51$, **** $p<0.0001$;) and stretchability: ($F_{4,15}=1.492$, n.s. $p=0.2543$), respectively. **e**, Bending stiffness of COMPACT (3 wt. % TEOS) hydrogel fiber ($n=4$ independent samples) with identical cross-sections in comparison with a 200- μm silica waveguide ($n=3$ independent samples). **f**, Stability assessment of diameter reduction of COMPACT hydrogel

fibers (3wt. % TEOS). Each dot (mean \pm s.d.) represents one independent fiber (two-way ANOVA and Tukey's multiple comparisons tests). **g**, Cytotoxicity assessment of COMPACT (+) hydrogels. Hydrogel incubated media was used to culture with Human Embryonic Kidney (HEK) 293 cell cultures. Calcein-AM (green) was used to stain living cells and ethidium homodimer-1 (red) was used to stain dead cells. Cell death rates are presented as mean \pm standard error (s.e.m., unpaired student's t-test).

Following the reviewer's suggestion, we performed additional tissue immunohistology staining to quantify the immune responses. Astrocytes and activated microglial cells were stained using glial fibrillary acidic protein (GFAP) and ionized calcium-binding adaptor molecule 1 (Iba1) antibodies, respectively. Fluorescence microscopy was employed to visualize the immune response in brain tissues around the implantation sites. Through ImageJ-assisted image processing, we compared fluorescence intensity and area, observing a significant reduction in the accumulation of astrocytes and activated microglia in the tissues implanted with COMPACT hydrogel fibers compared to those with silica-neural probes (**Fig. R22**). These findings have been incorporated into the Supplementary Information (**Supplementary Fig. S16**) of the revised manuscript. We have updated the latest results in the revised manuscript **Page 11 line 20 to page 11 line 22**:

“We compared the immune response of the brain tissues after implantations, less astrocytes and microglial accumulations were observed around the hydrogel fiber implantation sites than stiff silica fibers (Supplementary Fig. 16).”

It is noteworthy that the fabrication of COMPACT hydrogel fibers does not necessitate costly, specialized facilities such as photolithography or thermal pulling. This aspect of COMPACT technology allows for easy scalability and control over the dimensions of the hydrogel fibers.

In conclusion, the mechanical properties, miniaturization capabilities, and the quantitatively assessed immune response strongly support the suitability of COMPACT hydrogel neural probes for in vivo applications, particularly in scenarios involving dynamic movement. These results underscore the advantages of COMPACT hydrogels compared to traditional implantable devices.

Figure R26. **a**, Representative confocal images of astrocytes (GFAP, green) and microglia/macrophages (Iba1, red) surrounding a 400 μm hydrogel fiber probe (n=6) in mouse brain 1 month after implantation. **b**, Representative optical images of astrocytes (GFAP, green) and microglia/macrophages (Iba1, red) surrounding a 400 μm silica fiber probe (n=6) in mouse brain 1 month after implantation. Scale: 100 μm. **c**, Left: counted GFAP fluorescence intensity from hydrogel fibers implantation sites; Right: Left: counted Iba1 fluorescence intensity from hydrogel fibers implantation sites. (Unpaired student's t-test, GFAP: *p=0.0167; Iba1: *p=0.0108). **d**, Left: counted GFAP fluorescence intensity from silica fibers implantation sites; Right: Left: counted Iba1 fluorescence intensity from silica fibers implantation sites. (Unpaired student's t-test, GFAP: *p=0.0251; Iba1: **p=0.0056). Each dot represents measurement from one confocal image around the implantation site, mean ± sem.

Comment 2:

The hydrogel fiber used for neural signal recording does not demonstrate convincing results. The recorded signals lack spontaneous activity, and the noise level appears to be very high. Moreover, there are fluctuations in the noise level and recorded signals over the implanted time, indicating a potential decrease in electrode performance over time.

Response 2:

In response to the reviewer's concerns, we have conducted comprehensive extracellular electrical recordings using CNTs-PVA hydrogel microelectrodes in the ventral tegmental area (VTA) of wild-type mice. These efforts were aimed at addressing the issues related to signal quality and long-term performance of the neural probes.

Our recordings captured electrical signals from VTA neurons exhibiting spontaneous bursting activity. The signal-to-noise ratio (SNR) of these recordings was approximately 3.73. The spike sorting results indicated a repeatable spiking waveform (**Fig. R27**). This result has been updated in the revised main text **Fig. 4h-i**.

Figure R27. a, Representative endogenous activities recorded with CNTs-PVA hydrogel electrodes from mouse VTA. **i**, Overlaid action potentials sorted from the recording in **a**.

We have thoroughly investigated the long-term stability and performance of the CNTs-PVA hydrogel microelectrodes. These hydrogels, known for their fatigue resistance, have been further validated in our study. We subjected the CNTs-PVA hydrogel fibers to cyclic mechanical stretching underwater, simulating *in vivo* conditions, and measured their impedance before and after stretching. The fibers were subjected to strains of 5%, 10%, and 20% over 3000 cycles. Despite these rigorous conditions, the CNTs-PVA hydrogel fibers maintained an impedance of approximately 4 k Ω ·mm (**Fig. R28**), demonstrating their reliability for long-term use.

Figure R28. a, Plot of stress vs. cycle number of the CNTs-PVA fiber under cyclic tensile test. The applied strain amplitude in the cyclic tensile test is 10% and the total cycle number is 3000. **b**, Plot of stress vs. cycle number of the CNTs-PVA fiber under cyclic tensile test from cycle 3500 to 3510. Strain amplitude: 10%. **c**, Normalized impedance of CNTs-PVA fibers pre and post cyclic tensile tests under different strain amplitudes (5%, 10%, and 20%). Paired t-tests were used to determine statistical significance (5%: * $p=0.0338$, 10%: ** $p=0.0051$, and 20%: ** $p=0.0026$).

On the basis of the reliable conductivity of CNT-PVA hydrogel microelectrodes, to specifically address the question of recording stability over time, we extended our recording period to 10 weeks post-implantation (**Fig. R29**). During this period, we conducted optogenetics stimulation and simultaneous electrical recordings using the same device. The results showed that the SNR remained consistently high, ranging from 8.13 to 21.53, with the SNR at week 10 still maintaining at 14.94. These findings, indicative of sustained performance over a prolonged period, have been included in our revised manuscript **Figure 4n-o**.

Figure R29. a, Representative in vivo electrophysiological signals recorded with optrodes upon optical stimulation (blue bars, $\lambda=473$ nm, 0.5 Hz, pulse width 50 ms, 10 mW/mm²). b, Amplitudes of electrophysiological signals (n=3 mice) recorded with optical stimulation in week 0, week 1, week 2, week 4, week 6, and week 10 post-implantation ($\lambda=473$ nm, 0.5 Hz, pulse width 50 ms, 10 mW/mm², mean \pm s.e.m.).

We included these new set of results of in vivo electrophysiological recordings in updated **Fig. 4** and main text **Page 15 line 12 to page 15 line 18**:

“Instead of recording collective electrical response from muscles, we then implanted CNTs-PVA hydrogel electrodes in mouse VTA to record neuron spontaneous spiking activity in anesthetized wild-type mice under continuous isoflurane (Fig. 4g-i). A bandpass filter of 300–3000 Hz was applied to detect spiking activity, and we found one distinct cluster of spikes of principal component analysis (PCA). The signal-to-noise ratio (SNR) of these spiking activities was approximately 3.73 with repeatable waveforms.”

And page 16 line 4 to page 16 line 6:

“The optical evoked potentials were repeatedly captured with correlation with the onset of light stimulation over 10 weeks post-implantation (Fig. 4n and o).”

Figure 4. Integrated multifunctional hydrogel neural probes. **a**, A representative photograph of a carbon nanotubes (CNTs)-PVA hydrogel electrode as compared with a piece of human hair. Scale: 300 μm . **b**, A transmission electron microscopy (TEM) image of CNTs. Scale: 200 nm. **c**, Impedance at 1 kHz (red dots) and diameters of the electrodes (blue dots) fabricated different stretching percentages (mean \pm s.d.). Each dot represents one independent hydrogel electrode. **d**, Impedance at 1 kHz of electrodes fabricated with different CNTs concentrations (mean \pm s.d.). Each dot represents one independent hydrogel electrode. **e**, Impedance at 1 kHz of electrodes (red dots) and diameters of the electrode (blue dots) fabricated with different sizes of molds (mean \pm s.d.). Each dot represents one independent hydrogel electrode. **f**, Stability assessment on impedance (red dots) and diameters (blue dots) of hydrogel electrodes incubated in PBS at 37 $^{\circ}\text{C}$ (mean \pm s.d.). Each dot represents one independent hydrogel electrode. **g**, A schematic illustration of electrical recordings from mouse VTA with a CNTs-PVA electrode bundle (5 electrodes). **h**, Representative electrophysiology recording signals from mouse VTA with CNTs-PVA hydrogel electrodes. **i**, A representative sorted spiking signal. **j**, A representative scanning electron microscopy (SEM) image at the cross-section of an integrated multifunctional neural probe containing a hydrogel optical core and two CNTs-PVA hydrogel electrodes. Scale: 100 μm . **k-l**, Photographs of a hydrogel optoelectronic device (optrode) before implantation and after implantation in a *Thy1::ChR2-EYFP* mouse brain. Scale: 2 mm. **m**, Confocal images of the expression of ChR2-EYFP in the VTA region of the mouse. Scale: 50 μm . **n**, Representative in vivo electrical signals recorded with optrodes upon optical stimulation (blue bars, $\lambda=473$ nm, 0.5 Hz, pulse width 50 ms, 10 mW/mm²). **o**, Amplitudes of electrical signals (n=3 mice) recorded with optical stimulation over 10 weeks post-implantation ($\lambda=473$ nm, 0.5 Hz, pulse width 50 ms, 10 mW/mm², mean \pm s.e.m.).

Comment 3:

The dimensions of the implanted fibers seem considerably larger compared to previously reported hydrogel fiber electrodes. Although the fiber can maintain its volume and be manufactured in a small form factor, its size and elastic modulus are quite large, which pose critical disadvantages for an implanted device.

Response 3:

To address the reviewer's concerns regarding the size of the implanted hydrogel fibers, we would like to emphasize the versatility and adaptability of the COMPACT strategy in producing hydrogel fibers of various sizes. The dimensions of the fibers are not inherently limited by the COMPACT strategy but are determined by the molding methods used in their fabrication.

In our experiments, we have successfully prepared a range of hydrogel fibers with varying sizes using different silicon tubing molds (**Fig. R30**). This demonstrates the capability of the COMPACT strategy to produce hydrogel fibers in a variety of dimensions, catering to different application requirements. The 400 μm optical waveguides used in our study, were specifically designed to ensure adequate coverage of GCaMP-expressing neurons in the VTA and to effectively collect photometry optical signals in vivo. This dimension is consistent with what has been utilized in previous studies for in vivo mouse fiber photometry recording experiments. It is important to note that the size of these fibers was deliberately chosen to fulfill the functional requirements of our specific in vivo applications, particularly for optogenetics and fiber photometry recordings.

Figure R30. The pristine fiber and ultimate fiber diameters fabricated through different mold sizes via the COMPACT strategy (4% TEOS, acidification and 200% Stretching).

For different applications, such as extracellular electrophysiological recordings, we designed CNTs-PVA hydrogel fibers with dimensions of $118 \pm 27 \mu\text{m}$ (**Fig. R31**). This further illustrates the flexibility of the COMPACT strategy in creating hydrogel fibers suited to various experimental needs.

Figure R31. a, A representative photograph of a carbon nanotube (CNTs)-PVA hydrogel electrode as compared with a piece of human hair. Scale: $300 \mu\text{m}$. b, Impedance at 1 kHz (red dots) and diameters of the electrodes (blue dots) fabricated different stretching percentages (mean \pm s.d.). Each dot represents one independent hydrogel electrode. c, Impedance at 1 kHz of electrodes (red dots) and diameters of the electrode (blue dots) fabricated with different sizes of molds (mean \pm s.d.). Each dot represents one independent hydrogel electrode.

Additionally, the COMPACT approach, which includes the use of additional crosslinkers, acidification, and mechanical stretching, allows for a significant reduction in fiber diameters. Our experiments have shown that we can achieve over an 80% reduction in the diameter of hydrogel fibers (**Table R3**).

TEOS content (wt. %)	0% stretch	100% stretch	200% stretch
0	48.83 ± 1.43	60.58 ± 2.43	74.22 ± 1.84

1	49.36 ± 2.04	65.53 ± 1.66	75.02 ± 2.33
2	49.53 ± 2.30	66.67 ± 2.49	76.93 ± 1.19
3	50.29 ± 1.96	69.83 ± 1.46	80.78 ± 1.84
4	50.37 ± 1.00	67.95 ± 2.87	79.74 ± 2.30

The aforementioned detailed information has been included in **Fig. 2b**, **Fig. 4a, c** and **f**, the discussion about our fiber diameter further miniaturization has been included page 9 line 20 to page 10 line 2:

“Controllable hydrogel shrinking provides an effective methodology for miniaturization. The molding and extrusion approaches offer a series of precisely controlled hydrogel fiber diameters with structural homogeneity and low surface asperity to avoid diffuse reflection at the hydrogel interfaces. Although the mold sizes are commercially limited, COMPACT procedures, including regulating polymer and crosslinker constituent content and fiber extensions can expand the range of available fiber sizes.”

Comment 4:

The authors measured the GCaMP signal from socially interacting mice. However, no significant correlation was found between the GCaMP signal and the duration of social interaction.

Response 4:

In response to the reviewer's comment, we have made substantial improvements to our approach for capturing and analyzing GCaMP signals during mouse social interactions. This includes enhancements to both our data acquisition and analytical methods. We have refined the DeepLabCut labeling process and updated our MATLAB codes for more accurate analysis of social behaviors. We also increased the number of social behavior experiments. A larger cohort of mice were subjected to repeated sessions of photometry recordings and social interaction tests. The refined analysis and expanded data set revealed a general correlation between social bouts and increased GCaMP fluorescence in the VTA (**Fig. R32**). These observations suggest a link between social interactions and neural activity in this brain region. These results have been updated in our revised manuscript **Fig. 3h**.

We acknowledge that there were instances where the GCaMP signal fluctuations did not align precisely with social interactions. We attribute this to the complex functions of the VTA, which is involved in a variety of behaviors including reward and motivation. Given the multifaceted nature of mouse social behavior, which is influenced by multiple brain circuits, it is conceivable that not all increases in calcium influx in VTA neurons are directly related to social interactions. When we compared our fiber photometry recordings using COMPACT hydrogel fibers with those obtained using traditional silica fibers in moving mice (**Fig. R33**), we found the quality of our recordings to be comparable. This indicates that despite the complexities and challenges inherent in recording neural activity during dynamic social interactions, our COMPACT hydrogel fibers are effective in capturing relevant photometric signals.

Figure R32. Up: A schematic illustration of fiber photometry recording setup with concurrent mouse social behavior tests and a representative image in mouse social interaction tests. Down: Normalized fluorescence intensity change ($\Delta F/F_0$) of GCaMP6s in the VTA from mice social interactions. Blue bars indicate social interaction time analyzed by DeepLabCut (DLC). Fiber photometric recordings were performed with 5 mice implanted with COMPACT hydrogel optical probes.

[REDACTED]

Figure R33. Fiber photometric recordings with GCaMP fluorescence results during mouse behaviors in other published works showing correlations of animal behavioral results and GCaMP signals.

Comment 5:

In the past, several hydrogel-based optical fibers and electrodes have been developed. However, the reviewer was unable to identify any advantages of the proposed device as an implantable device.

Response 5:

We appreciate the reviewer's attention to previous work in the field of hydrogel-based optical fibers and electrodes. In response, we have expanded our review of the accomplishments in this field, providing context for our advancements with the COMPACT strategy.

Hydrogel-based optical fibers have found diverse applications in light delivery and biosensing, each leveraging the unique properties of hydrogels. For instance, alginate-based hydrogel fibers have been celebrated for their exceptional stretchability^{1, 2}, finding use in biosensing applications due to their flexibility and durability. Similarly, polyacrylamide-based hydrogel fibers have been developed with a no-core fiber sensor design, exhibiting a linear response in pH³ and glucose sensing⁴, demonstrating the adaptability of hydrogels to various sensing needs. Polyethylene glycol diacrylate hydrogels, crafted into flexible fibers with tip functionalization⁵, have also shown potential in glucose sensing applications. Agarose-based hydrogel fibers have been utilized as biopolymeric planar waveguides for light guiding^{6, 7}, showcasing their optical properties. Additionally, the synthetic engineering of spider silk fibers for implantable optical waveguides represents a fusion of natural and synthetic materials for biomedical applications^{8, 9}. By further incorporating hydrogels as ionic conductors and hydrophobic elastomers as dielectrics, stretchable hydrogel fibers have been developed for luminescent fabrics¹⁰.

Some hydrogel optical fibers have been advanced for in vivo application with behavioral results. Biodegradable PLLA optical fibers have been applied in vivo optogenetics with locomotor behaviors¹¹. Our recent work has demonstrated the effective use of chemically crosslinked PVA hydrogels in peripheral nerve stimulation via optogenetics, achieving a significant pain inhibition effect in freely moving mice¹².

Hydrogel-based electrodes represent significant advancement in neuromodulation technology¹³. These conductive hydrogels are typically created by printing conductive polymers like poly(3,4-ethylenedioxythiophene): poly(styrene sulfonate) (PEDOT: PSS) onto the hydrogel substrate^{14, 15} or by incorporating them directly into the hydrogel matrix^{16, 17, 18}. However, these methods often require specialized equipment such as 3D printers, electrodeposition units, or photolithography systems.

In contrast, the production of CNTs-PVA hydrogel microelectrodes via the COMPACT method simplifies the process significantly. This technique allows for scalable production of microelectrodes using a one-step crosslinking process. Utilizing the COMPACT strategy for creating integrated optrodes ensures that the CNTs-PVA microelectrodes are embedded within the same PVA matrix as the optical waveguides. This integration is facilitated by an intrinsic polymer filtration process during fabrication, which enhances the interfacial contact between the microelectrodes and the waveguides, thereby improving the overall performance of the optrodes.

In addition to COMPACT hydrogel fibers' softness and stretchability, which makes them particularly suited for dynamic in vivo applications, especially with freely moving subjects, one of the key features of the COMPACT hydrogels is the controllable miniaturization and integration.

Rather than directly using crosslinked hydrogel fibers as a waveguide¹², the COMPACT strategy offers a comprehensive approach to hydrogel-based device fabrication and integration. It focuses on minimizing device footprint and integrating multiple functionalities into a single device. This advantage makes it a notable advancement in the field of neurobiological interfaces.

Ref:

1. Choi M, Humar M, Kim S, Yun S-H. Step-Index Optical Fiber Made of Biocompatible Hydrogels. *Advanced Materials* 27, 4081-4086 (2015).
2. Guo J, et al. Highly Stretchable, Strain Sensing Hydrogel Optical Fibers. *Advanced Materials* 28, 10244-10249 (2016).
3. Pathak AK, Singh VK. A wide range and highly sensitive optical fiber pH sensor using polyacrylamide hydrogel. *Optical Fiber Technology* 39, 43-48 (2017).
4. Yetisen AK, et al. Glucose-Sensitive Hydrogel Optical Fibers Functionalized with Phenylboronic Acid. *Advanced Materials* 29, 1606380 (2017).
5. Elsherif M, Hassan MU, Yetisen AK, Butt H. Hydrogel optical fibers for continuous glucose monitoring. *Biosensors and Bioelectronics* 137, 25-32 (2019).
6. Manocchi AK, Domachuk P, Omenetto FG, Yi H. Facile fabrication of gelatin-based biopolymeric optical waveguides. *Biotechnology and Bioengineering* 103, 725-732 (2009).
7. Jain A, Yang AHJ, Erickson D. Gel-based optical waveguides with live cell encapsulation and integrated microfluidics. *Opt Lett* 37, 1472-1474 (2012).
8. Qiao X, et al. Synthetic Engineering of Spider Silk Fiber as Implantable Optical Waveguides for Low-Loss Light Guiding. *ACS Applied Materials & Interfaces* 9, 14665-14676 (2017).
9. Applegate MB, Perotto G, Kaplan DL, Omenetto FG. Biocompatible silk step-index optical waveguides. *Biomed Opt Express* 6, 4221-4227 (2015).
10. Yang C, Cheng S, Yao X, Nian G, Liu Q, Suo Z. Ionotronic Luminescent Fibers, Fabrics, and Other Configurations. *Advanced Materials* 32, 2005545 (2020).
11. Fu R, et al. Implantable and Biodegradable Poly(l-lactic acid) Fibers for Optical Neural Interfaces. *Advanced Optical Materials* 6, 1700941 (2018).
12. Liu X, et al. Fatigue-resistant hydrogel optical fibers enable peripheral nerve optogenetics during locomotion. *Nature Methods* 20, 1802-1809 (2023).
13. Liu Y, et al. Soft and elastic hydrogel-based microelectronics for localized low-voltage neuromodulation. *Nature Biomedical Engineering* 3, 58-68 (2019).
14. Sekine S, Ido Y, Miyake T, Nagamine K, Nishizawa M. Conducting Polymer Electrodes Printed on Hydrogel. *Journal of the American Chemical Society* 132, 13174-13175 (2010).
15. Sasaki M, Karikkineth BC, Nagamine K, Kaji H, Torimitsu K, Nishizawa M. Highly Conductive Stretchable and Biocompatible Electrode-Hydrogel Hybrids for Advanced Tissue Engineering. *Advanced Healthcare Materials* 3, 1919-1927 (2014).
16. Wang Y, et al. A highly stretchable, transparent, and conductive polymer. *Science Advances* 3, e1602076.
17. Feig VR, Tran H, Lee M, Bao Z. Mechanically tunable conductive interpenetrating network hydrogels that mimic the elastic moduli of biological tissue. *Nature Communications* 9, 2740 (2018).
18. Green RA, et al. Conductive Hydrogels: Mechanically Robust Hybrids for Use as Biomaterials. *Macromolecular Bioscience* 12, 494-501 (2012).

We have enriched the introduction and discussion about soft-material bioelectronics in neuromodulation applications and recording our revised manuscript in **page 3 line 2 to page 3 line 4**:

“Multifunctional COMPACT hydrogel bioelectronics grants bi-directional access to cellular-level neural activity and links behavioral assessment, providing effective tools to comprehensively understand neural mechanisms.”

Page 4 line 5 to page 4 line 10:

“For the sophisticated yet delicate nervous system interfaces, elastic polymer materials, including polydimethylsiloxane (PDMS), cyclic olefin copolymer elastomer (COCE), polyurethane (PU), and hydrogels have been deployed as the suitably elastic substrate for multifunctional devices that enable neural optogenetics stimulation, electrophysiological recording, drug infusion and neurotransmitter detection.”

And page 14 line 9 to page 14 line 11:

“Linking the neural activities at the cellular level to system neuroscience behavioral assessment provides important tools to discover causal relationship of neural circuits and behaviors for neuroscience studies.”

Responses to Reviewer 3's comments

General comments:

The manuscript presents an innovative fabrication strategy for hydrogel bioelectronics using metamorphic polymers' amorphous-crystalline transition (COMPACT). The approach enables the miniaturization of hydrogel fibers with multifunctional interrogation of neural circuits. The work also focuses on the fabrication of step-index hydrogel optical probes and CNTs-PVA hydrogel microelectrodes for optogenetics and photometric recordings in the mouse brain region of the ventral tegmental area (VTA) and hindlimb muscle electromyographic and brain electrophysiological recordings of light-triggered neural activities in transgenic mice expressing Channelrhodopsin-2 (ChR2). Overall, the manuscript was well prepared. The results were presented in a logical and easy-to-follow manner, with appropriate figures and tables to support the findings. The study is of high significance, as it presents a novel approach for constructing hydrogel bioelectronics with miniaturized fiber shape and multifunctional interrogation of neural circuits, which has potential applications in brain-machine interfaces and investigating complex neural circuits. Therefore, the manuscript is recommended for publication in Nature Communications, subject to minor revisions detailed as follows:

Response to general comment:

We are grateful for the reviewer's positive assessment of our manuscript, acknowledging the innovation and significance of our COMPACT (Control of Metamorphic Polymers' Amorphous-Crystalline Transition) strategy in the fabrication of hydrogel bioelectronics. In response to your suggestion for minor revisions, we have undertaken additional steps to enhance our manuscript. These improvements were carefully considered to not only address the specific feedback but also to further clarify and underscore the potential applications of our research.

Comment 1:

The manuscript could benefit from a more detailed discussion of the potential limitations and challenges of the proposed approach. For instance, the study only demonstrates the fabrication of hydrogel fibers with a diameter decrease of about 80%. It would be helpful to discuss the potential challenges of achieving further miniaturization and the impact of the hydrogel properties on the device's performance (review the advantages and current approaches).

Response 1:

We appreciate the reviewer's suggestions to delve deeper into the potential limitations and challenges of the COMPACT strategy.

The COMPACT approach offers a controlled method for minimizing hydrogel fiber dimensions using additional crosslinkers (TEOS), acidification, and mechanical stretching. The current capability of achieving approximately an 80% diameter decrease reflects a deliberate balance, designed to maintain suitable optical properties like light transmission (~90%) and refractive index (RI 1.37-1.40 at 480 nm), along with favorable mechanical properties such as stretchability (95% - 111%), Young's modulus (10-63 MPa), and axial bending stiffness (4.6 ± 1.4 N/m).

Despite the success of COMPACT in reducing fiber diameters, there are inherent limitations primarily due to the nature of polymer crosslinking and changes in mechanical properties resulting from polymeric nanocrystalline growth. The interplay between optical, mechanical, and

biocompatibility characteristics imposes constraints on further dimensional reduction. For instance, not all hydrogels can undergo cross-linking and polymeric crystallization processes, and increasing crosslinking or crystallization density can decrease the hydrogel's water content, subsequently increasing Young's modulus and limiting stretchability. These tradeoffs of mechanical and optical properties have also been recognized in other anti-swelling hydrogel development through regulating the crosslinking, hydrophilicity/hydrophobicity balance, and nanocomposite^{1, 2, 3, 4}. Most of the anti-swelling hydrogel development has been focused on water hydrated state for tissue engineering applications^{5, 6}. The transition of hydrogel swelling, tuning by the polymeric nanostructures has also granted hydrogels other behaviors, such as dramatic volume swelling and humidity-responsive swelling. Recent advances in 3D polymer structure miniaturization involve collapsing polymer chains, often achieved with laser-patterning techniques. It's important to note that this miniaturization is typically achieved in a dehydrated state, which differs from the physiologically hydrated conditions where COMPACT operates. COMPACT focuses on constraining polymer chain swelling in water to maintain the designed dimensions of hydrogel bioelectronics.

Ref:

1. Zhan Y, Fu W, Xing Y, Ma X, Chen C. Advances in versatile anti-swelling polymer hydrogels. *Materials Science and Engineering: C* 127, 112208 (2021).
2. Xu B, et al. High-Strength Nanocomposite Hydrogels with Swelling-Resistant and Anti-Dehydration Properties (2018).
3. Zhu T, et al. Recent advances in conductive hydrogels: classifications, properties, and applications. *Chemical Society Reviews* 52, 473-509 (2023).
4. Richbourg NR, Peppas NA. The swollen polymer network hypothesis: Quantitative models of hydrogel swelling, stiffness, and solute transport. *Progress in Polymer Science* 105, 101243 (2020).
5. Wang S, et al. Underwater Adhesion and Anti-Swelling Hydrogels. *Advanced Materials Technologies* 8, 2201477 (2023).
6. Feng W, Wang Z. Tailoring the Swelling-Shrinkable Behavior of Hydrogels for Biomedical Applications. *Advanced Science* 10, 2303326 (2023).

In the revised manuscript, we have enriched the introduction and discussion about soft-material bioelectronics and hydrogel shrinking behaviors in **Page 4 line 21 to page 5 line 8**:

“Hydrogel swelling behaviors in response to external stimuli have enabled drug release, ingestible devices, and expansion microscopy to enhance microimaging resolution. Anti-swelling hydrogels are usually constructed by regulating the cross-linking, hydrophilicity/hydrophobicity balance and nanocomposite for tissue engineering applications. Meanwhile, hydrogel shrinking behaviors in a desiccated state have been applied to densify patterned materials in volumetric scaffold deposition and obtain nanoscale feature sizes in three dimensions. However, the hydrogel swelling and shrinking behaviors in these techniques are based on reversible polymer chains collapse in the desiccated state and expansion upon hydration. When applied to an aqueous in vivo environment, the shrunk hydrogels will expand and lose the miniaturized structures from the original manufacturing.”

Page 9 line 15 to page 10 line 2:

“Axial mechanical deformation re-orientated nanocrystalline and created anisotropic nanostructures (Fig. 1k and Supplementary Fig. 4 and 5), which enabled hydrogel fibers' desired decrease in diameter while causing a minimal effect on crystallinity degree (Supplementary Fig. 1a and b) or nanocrystalline size (Fig. 1k and Supplementary Fig. 3c and d).

Controllable hydrogel shrinking provides an effective methodology for miniaturization. The molding and extrusion approaches offer a series of precisely controlled hydrogel fiber diameters with structural homogeneity and low surface asperity to avoid diffuse reflection at the hydrogel interfaces. Although the mold sizes are commercially limited, COMPACT procedures, including regulating polymer and crosslinker constituent content and fiber extensions can expand the range of available fiber sizes.”

Comment 2:

The explanation of the social behavioral assessment and the results obtained from the optogenetics and photometric recordings in the mouse brain region of the ventral tegmental area (VTA) should be further elaborated, along with more detailed discussion of the significance of the findings and their potential implications for future research.

Response 2:

Thank you for the insightful comments. We appreciate the opportunity to elaborate on the behavioral assessment and the results obtained from the optogenetics and photometric recordings in behaving mice. In our study, the social behavioral assessment was conducted using a well-established social interaction test. This involved observing and recording the interactions of test mice with a novel mouse. We monitored the time when social interaction happened and correlated the behavioral observation with the fiber photometry recordings using genetically encoded calcium indicators, GCaMP, with the COMPACT hydrogel optical fibers. Fiber photometry was used to record calcium signals as proxies for neuronal activity. Such simultaneous photometric recording and behavioral assessment will provide insights into the neural activity of VTA neurons associated with social behaviors. Although it is a proof-of-concept demonstration of the functionality of COMPACT hydrogel fibers for photometry recording in behaving mice, we found the correlation of VTA neurons with social interaction can be an example to link cellular neural activity to behaviors, which is beneficial for neuroscience study of the brain function. One thing we would like to highlight is that, although VTA has been chosen to demonstrate our COMPACT fiber can probe deep brain structures, the application of COMPACT neural probes is not limited to specific brain regions. Due to the soft, stretchability and stability in complex in vivo working conditions, COMPACT hydrogel fibers can easily access deep brain structure involved with motions. For instance, recording VTA activity during mouse social behaviors can open new avenues for research into the neural mechanisms underlying social behaviors and their dysregulation in disorders such as autism spectrum disorder and schizophrenia. Future studies could explore the modulation of VTA activity as a potential therapeutic approach for these conditions. When use COMPACT neural probes for optogenetics, fiber photometry and electrophysiological recordings of neural circuits in behaving experimental subjects, it presents a powerful tool to dissect brain function and comprehensively understand neurological disorders.

In the revised manuscript, we further demonstrate the functionality of the multifunctional COMPACT neural probes with optogenetics stimulation and electrophysiological recording functions over long-term observations (**Fig. 4**). Integrating neural stimulation and recording functionalities into a single miniaturized device enhances neuroscience research, offering a comprehensive tool for understanding complex brain functions. This integration enables simultaneous manipulation and observation of neural circuits, providing insights into the causal relationships within the brain. Such precision and control are essential for studies focusing on specific neural pathways or brain regions, facilitating more accurate research.

We have updated the manuscript, emphasizing the importance of linking behavior assessments with neuromodulation and recording as reflected in the main text Page 2 line 19 to page 3 line 4:

“To extend COMPACT hydrogel multifunctional scaffolds to assimilate conductive nanomaterials, we developed carbon nanotubes (CNTs)-PVA hydrogel microelectrodes for in vivo electrophysiological recordings of spontaneous neural activities. When integrating multiple components of optical waveguide and electrodes into a miniaturized device, we simultaneously conducted optogenetic stimulation and electrophysiological recordings of light-triggered neural activities in transgenic mice expressing Channelrhodopsin-2 (ChR2). Multifunctional COMPACT hydrogel bioelectronics grants bi-directional access to cellular-level neural activity and links behavioral assessment, providing effective tools to comprehensively understand neural mechanisms.”

Page 6 line 11 to page 6 line 17:

“Taking advantage of these tunable hydrogel matrix scaffolds, we loaded conductive nanomaterials, carbon nanotubes (CNTs), into COMPACT hydrogels for soft microelectrodes, and tested its functionalities for electrophysiological recordings of spontaneous neural activity in the mouse brain. Integrating a hydrogel optical core and CNTs-PVA microelectrodes, we developed multifunctional hydrogel optoelectronic devices and demonstrated the simultaneous optogenetic stimulation and electrophysiological recording of optically triggered neural activities in transgenic *Thy1::ChR2-EYFP* mice.”

Page 14 line 6 to page 14 line 11:

“Mouse social interactions were analyzed with DeepLabCut (DLC)⁵⁸ markless pose estimation and a custom developed MatLab algorithm (Fig. 3f). We observed the increased fluorescent intensity of GCaMP was correlated with mouse social interaction epochs (Fig. 3h). Linking the neural activities at the cellular level to system neuroscience behavioral assessment provides important tools to discover causal relationship of neural circuits and behaviors for neuroscience studies.”

Page 16 line 7 to page 16 line 14:

“Simultaneous bi-directional stimulation and recording of neurons with optical and electrical modalities offer comprehensive approaches to studying brain function. The COMPACT strategy offers the convenience of integrating multiple functional components into a single miniaturized device. Successive rounds of molding with strong polymer chain infiltration at the interfaces enable the design of multimodal microstructures. Currently, the number of integrated components, such as electrodes and microfluidic channels, is limited by the coaxial alignment in the secondary molding step; the accessibility and throughput of multimodal fabrication can be further improved with guiding devices to facilitate integration and alignment, or alternative coating approaches.”

Comment 3:

This work is related to the hydrophilic materials, polyvinyl alcohol (PVA) in particular. In fact, PVA is a versatile polymer that has been studied extensively for various applications. There might

be a need to address the structure and properties of PVA when it is combined with other materials like biopolymer in terms of different applications.

(see <https://doi.org/10.1016/j.jobab.2022.01.002>; <https://doi.org/10.1016/j.jobab.2021.11.004>; and <https://www.nature.com/articles/s41467-022-32987-6>).

A concise review is suggested to be added to the introduction section. More importantly, PVA hydrogels are amorphous polymers that can undergo an amorphous-crystalline transition under certain conditions. The as-proposed COMPACT strategy for controlling this transition involves the introduction of multiple cross-linkers and acidification treatment to facilitate oriented polymeric crystalline growth under mechanical stretching.

Response 3:

Thanks for the reviewer's suggestion to enhance our manuscript with a more detailed exploration of polyvinyl alcohol (PVA) and its applications. We agree that incorporating a concise review of PVA, especially in the context of our COMPACT strategy, would provide valuable context and depth to our study.

PVA is a versatile polymer, widely recognized for its biocompatibility, non-toxic nature, and excellent film-forming capabilities^{1, 2}. These attributes have made PVA a popular choice in various biomedical and pharmaceutical applications. In drug delivery systems, PVA serves as a hydrophilic polymer matrix for controlled-release formulations. Its ability to finely tune water solubility through adjustments in polymerization and hydrolysis levels allows for precise control over drug release rates. This adaptability is crucial in developing systems that can consistently deliver therapeutic agents over extended periods.

PVA's moisture retention properties make it an ideal component in hydrogel-based systems for wound dressings. These hydrogels promote healing and reduce scar formation, offering an effective solution for wound care. PVA can also be combined with other components, such as chitosan³, to create composite membranes for wound dressing applications, or with cellulose nanocrystals for applications in food packaging⁴.

The crosslinking of PVA, achievable through physical, chemical, and UV-assisted methods, leads to the formation of hydrogels characterized by their hydrophilicity, biocompatibility, mechanical strength, and elasticity^{5, 6, 7, 8}. These crosslinked PVA hydrogels can effectively mimic the extracellular matrix, fostering cell adhesion and proliferation. As a result, they are suitable as scaffolds in tissue repair and regeneration. The mechanical properties of these hydrogels can be adjusted to provide the necessary support to wound sites while maintaining the flexibility to conform to tissue surfaces.

In addition to biomedical applications, the nanostructuring of PVA hydrogels has been explored to design hydrogel optical and mechanical properties^{9, 10, 11}. For instance, nanoimprinting techniques have been applied to PVA hydrogels to create tunable and erasable optical security metasurfaces¹², demonstrating the potential of PVA in nanophotonic applications. Dual crosslinked PVA with other polymer or introducing nanocomposite^{13, 14} into the hydrogel matrix offer advantageous mechanical properties^{15, 16}.

In our COMPACT strategy, we leverage the amorphous-crystalline transition properties of PVA hydrogels, controlled by introducing multiple cross-linkers and acidification treatment, to facilitate oriented polymeric crystalline growth under mechanical stretching. This approach exemplifies the manipulation of PVA's inherent properties to create hydrogel bioelectronics with desired characteristics, including miniaturization, mechanical resilience, and functional adaptability for biomedical applications.

Ref:

1. Paradossi G, Cavalieri F, Chiessi E, Spagnoli C, Cowman MK. Poly(vinyl alcohol) as versatile biomaterial for potential biomedical applications. *Journal of Materials Science: Materials in Medicine* 14, 687-691 (2003).
2. Marin Cardona ES, Rojas Camargo J, Ciro Monsalve YA. A review of polyvinyl alcohol derivatives: promising materials for pharmaceutical & biomedical applications. (2014).
3. Huq T, Khan A, Brown D, Dhayagude N, He Z, Ni Y. Sources, production and commercial applications of fungal chitosan: A review. *Journal of Bioresources and Bioproducts* 7, 85-98 (2022).
4. Wang J, Euring M, Ostendorf K, Zhang K. Biobased materials for food packaging. *Journal of Bioresources and Bioproducts* 7, 1-13 (2022).
5. Jiang S, Liu S, Feng W. PVA hydrogel properties for biomedical application. *Journal of the Mechanical Behavior of Biomedical Materials* 4, 1228-1233 (2011).
6. Kumar A, Han SS. PVA-based hydrogels for tissue engineering: A review. *International Journal of Polymeric Materials and Polymeric Biomaterials* 66, 159-182 (2017).
7. Adelnia H, Ensandoost R, Shebbrin Moonshi S, Gavvani JN, Vasafi EI, Ta HT. Freeze/thawed polyvinyl alcohol hydrogels: Present, past and future. *European Polymer Journal* 164, 110974 (2022).
8. Yuk H, Wu J, Zhao X. Hydrogel interfaces for merging humans and machines. *Nature Reviews Materials* 7, 935-952 (2022).
9. Guo J, et al. Highly Stretchable, Strain Sensing Hydrogel Optical Fibers. *Advanced Materials* 28, 10244-10249 (2016).
10. Liu X, et al. Fatigue-resistant hydrogel optical fibers enable peripheral nerve optogenetics during locomotion. *Nature Methods* 20, 1802-1809 (2023).
11. Lin S, Liu J, Liu X, Zhao X. Muscle-like fatigue-resistant hydrogels by mechanical training. *Proceedings of the National Academy of Sciences* 116, 10244-10249 (2019).
12. Ko B, et al. Tunable metasurfaces via the humidity responsive swelling of single-step imprinted polyvinyl alcohol nanostructures. *Nature Communications* 13, 6256 (2022).
13. Karimzadeh Z, Mahmoudpour M, Rahimpour E, Jouyban A. Nanomaterial based PVA nanocomposite hydrogels for biomedical sensing: Advances toward designing the ideal flexible/wearable nanoprobe. *Advances in Colloid and Interface Science* 305, 102705 (2022).
14. Millon LE, Guhadós G, Wan W. Anisotropic polyvinyl alcohol—Bacterial cellulose nanocomposite for biomedical applications. *Journal of Biomedical Materials Research Part B: Applied Biomaterials* 86B, 444-452 (2008).
15. Chang C, Lue A, Zhang L. Effects of Crosslinking Methods on Structure and Properties of Cellulose/PVA Hydrogels. *Macromolecular Chemistry and Physics* 209, 1266-1273 (2008).
16. Ma R, Xiong D, Miao F, Zhang J, Peng Y. Novel PVP/PVA hydrogels for articular cartilage replacement. *Materials Science and Engineering: C* 29, 1979-1983 (2009).

In the revised manuscript, we have enriched the introduction and discussion about **Page 4 line 21 to page 5 line 2:**

“Hydrogel swelling behaviors in response to external stimuli have enabled drug release, ingestible devices, and expansion microscopy to enhance microimaging resolution. Anti-swelling hydrogels are usually constructed by regulating the cross-linking, hydrophilicity/hydrophobicity balance and nanocomposite for tissue engineering applications.”

Page 5 line 13 to page 5 line 18:

“Utilizing the nanoscale structure change to regulate soft materials properties has been proven as an effective approach to biomedical applications. As one of the typical semi-crystalline polymers, polyvinyl alcohol (PVA) hydrogels, have been widely used in drug release, food packaging and

wound healing. For hydrogel bioelectronics, the ability to control nanostructures through polymeric crystallization approaches can stably maintain their designed architectures in vivo.”
Page 6 line 20 to page 6 line 21:

“Chemically cross-linked PVA hydrogels have been widely employed with superior optical properties, fatigue-resistance and biocompatibility for bioelectronics applications.”

Comment 4:

The mechanisms involved in oriented polymeric crystalline growth under mechanical stretching may need to be further depicted, and videos or cartoons may be added to facilitate the readers’ understanding. (Fig. 1k) When a polymer is mechanically stretched, the degree of crystallinity and the orientation of polymer chains depend on the stretching conditions such as the rate and magnitude of the applied force, temperature, and the molecular structure of the polymer. The oriented crystalline structures contribute to the mechanical properties of the polymer, such as strength and stiffness, and can be utilized in various applications such as fibers, films, and composites.

Response 4:

We appreciate the reviewer’s thoughtful suggestions. In response, we have revised manuscript with additional visual aids and explanatory content. To visually demonstrate the process, we have included photographs of the controllable mechanical stretching devices used in the preparation of COMPACT hydrogel fibers (**Fig. R34**). These images provide a tangible representation of the equipment and setup involved in our experimental procedures. To further aid understanding, we have incorporated schematic drawings that detail how mechanical stretching influences the anisotropic spacing between nanocrystalline structures in hydrogels (**Fig. R35**).

By providing these additional resources and explanations, we aim to offer readers a comprehensive understanding of the scientific principles underpinning our COMPACT strategy. This approach not only serves to enhance the mechanical properties of hydrogel fibers but also opens new possibilities for their application in various fields. We have updated the discussion section of our manuscript to include a more detailed explanation of how tuning hydrogel nanostructures can regulate hydrogel mechanical, optical and miniaturization properties^{1, 2, 3}.

Figure R34. Stretching demonstration of hydrogel fibers. A customized stretching device for hydrogel fibers stretching (200% stretching).

Figure R35. Small-angle X-ray (SAXS) and wide-angle X-ray (WAXS) results of hydrogel materials in the desiccated state (mean \pm s.d.). Upper inset: schematic of nanocrystalline domains before and after stretching, middle inset: SAXS 2D patterns., lower inset: WAXS 2D patterns.

Ref:

1. Kuang X, Arican MO, Zhou T, Zhao X, Zhang YS. Functional Tough Hydrogels: Design, Processing, and Biomedical Applications. *Accounts of Materials Research* 4, 101-114 (2023).
2. Ko B, et al. Tunable metasurfaces via the humidity responsive swelling of single-step imprinted polyvinyl alcohol nanostructures. *Nature Communications* 13, 6256 (2022).
3. Lin S, Liu J, Liu X, Zhao X. Muscle-like fatigue-resistant hydrogels by mechanical training. *Proceedings of the National Academy of Sciences* 116, 10244-10249 (2019).

In the revised manuscript, this information has been added in **Supplementary Note 2b** and **Fig. 1k**, and the descriptions are reflected in **Page 4 line 23** to **page 5 line 2**:

“Anti-swelling hydrogels are usually constructed by regulating the cross-linking, hydrophilicity/hydrophobicity balance and nanocomposite for tissue engineering applications.”

Page 5 line 13 to page 5 line 16:

“Utilizing the nanoscale structure change to regulate soft materials properties has been proven as an effective approach to biomedical applications. As one of the typical semi-crystalline polymers, polyvinyl alcohol (PVA) hydrogels, have been widely used in drug release, food packaging and wound healing.”

Comment 5:

The abstract is well-written and effectively communicates the research findings. In the second sentence (Line 4), "microfabrication" should be hyphenated as "micro-fabrication." The "polymetric" in line 12 should be corrected to "polymeric."

Response 5:

In the revised manuscript, we have corrected “microfabrication” as “micro-fabrication”, and “polymetric” as “polymeric”.

Comment 6:

The novelty of this work lies in the development of a new fabrication strategy, called COMPACT, for hydrogel bioelectronics with miniaturized fiber shape and multifunctional interrogation of neural circuits. To highlight the novelty, the authors could emphasize the use of metamorphic polymers (e.g., in the abstract), the achievement of miniaturized fiber shapes, and the multifunctional interrogation of neural circuits using the newly developed hydrogel bioelectronics. Additionally, the use of these materials for optogenetics and photometric recordings in the mouse brain and hindlimb muscle electromyographic and brain electrophysiological recordings of light-triggered neural activities in transgenic mice expressing Channelrhodopsin-2 (ChR2) could be emphasized.

Response 6:

We appreciate the reviewer's suggestions. We have accordingly revised the manuscript to succinctly highlight the novelty of our COMPACT strategy in developing miniaturized and multifunctional hydrogel bioelectronics. We have emphasized the use of metamorphic polymers in our abstract, underscoring how their structural transitions enable significant advancements in bioelectronics miniaturization. Additionally, we have concisely illustrated the multifunctional applications of these hydrogel bioelectronics, particularly in optogenetics and various neural recording techniques in transgenic mice expressing Channelrhodopsin-2 (ChR2). This focus accentuates the innovative aspects of our work and its potential impact on neurobiology and brain-machine interface studies.

The updates have been made as:

Page 2 line 19 to page 3 line 4 in the abstract:

“To extend COMPACT hydrogel multifunctional scaffolds to assimilate conductive nanomaterials, we developed carbon nanotubes (CNTs)-PVA hydrogel microelectrodes for in vivo electrophysiological recordings of spontaneous neural activities. When integrating multiple components of optical waveguide and electrodes into a miniaturized device, we simultaneously conducted optogenetic stimulation and electrophysiological recordings of light-triggered neural activities in transgenic mice expressing Channelrhodopsin-2 (ChR2). Multifunctional COMPACT hydrogel bioelectronics grants bi-directional access to cellular-level neural activity and links behavioral assessment, providing effective tools to comprehensively understand neural mechanisms.”

Page 6 line 11 to page 6 line 17:

“Taking advantage of these tunable hydrogel matrix scaffolds, we loaded conductive nanomaterials, carbon nanotubes (CNTs), into COMPACT hydrogels for soft microelectrodes, and tested its functionalities for electrophysiological recordings of spontaneous neural activity in the mouse brain. Integrating a hydrogel optical core and CNTs-PVA microelectrodes, we developed multifunctional hydrogel optoelectronic devices and demonstrated the simultaneous optogenetic stimulation and electrophysiological recording of optically triggered neural activities in transgenic Thy1:: ChR2-EYFP mice.”

Page 14 line 6 to page 14 line 11:

“Mouse social interactions were analyzed with DeepLabCut (DLC) markless pose estimation and a custom developed MatLab algorithm (Fig. 3f). We observed the increased fluorescent intensity of GCaMP was correlated with mouse social interaction epochs (Fig. 3h). Linking the neural activities at the cellular level to system neuroscience behavioral assessment provides important tools to discover causal relationship of neural circuits and behaviors for neuroscience studies.”

Page 16 line 7 to page 16 line 14:

“Simultaneous bi-directional stimulation and recording of neurons with optical and electrical modalities offer comprehensive approaches to studying brain function. The COMPACT strategy offers the convenience of integrating multiple functional components into a single miniaturized device. Successive rounds of molding with strong polymer chain infiltration at the interfaces enable the design of multimodal microstructures. Currently, the number of integrated components, such as electrodes and microfluidic channels, is limited by the coaxial alignment in the secondary molding step; the accessibility and throughput of multimodal fabrication can be further improved with guiding devices to facilitate integration and alignment, or alternative coating approaches.”

REVIEWER COMMENTS

Reviewer #1 (Remarks to the Author):

The manuscript by Huang et al. has been much improved after revision. The authors provide an extensive set of materials characterizations to support their strategy of controlling amorphous-crystalline transition and hydrogel miniaturization. However, the authors still need to address several issues listed below before considering acceptance.

1. The reviewer is still not convinced of the description of tissue-like elasticity as neural probes. It is true that modulus is not the only factor for consideration, stretchability and bending stiffness should also be included for a comprehensive evaluation. But a neural probe with high modulus implies a high risk of damage to tissues, especially considering long-term applications. Can the authors investigate the local tissue property surrounding the probe after 14-day implantation?
2. In Fig. R5, the authors provide long term stability results in PBS, but how about the in vivo case? Will the contents such as proteins in real body fluids affect the stability of the hydrogel microelectrodes?
3. It is recommended to add more explanation about the results of materials characterization. For example, why does the impedance improve after cyclic stretching in Fig. R6? Why does the polymeric crystallinity decrease with the addition of CNTs in Fig. R2?

Reviewer #2 (Remarks to the Author):

Thank you for considering my comments and suggestions. The revised manuscript accurately incorporates my input and shows significant improvement. I have no further objections regarding the publications of your paper. Congratulations to your interesting results.

Reviewer #3 (Remarks to the Author):

In this revised manuscript, authors addressed most concerns raised previously. The manuscript indeed also showcases the fabrication of step-index hydrogel optical probes and CNTs-PVA hydrogel microelectrodes for various medical-related applications. The manuscript was well-written and effectively demonstrated an innovative method for constructing hydrogel bioelectronics, achieving miniaturization and multifunctional integration through the control of polymers' amorphous-crystalline transition. However, some very minor issues are needed to be addressed by authors prior to the acceptance of the manuscript for publishing *Natura Commu.*, a very top journal in relevant areas. Following are some specific comments:

1. The current title is already clear and accurate, but a slight modification would make it even better. For example, changing the title to "Controlling Polymers' Amorphous-Crystalline Transition Enables Miniaturization and Multifunctional Integration for Hydrogel Bioelectronics" may be worth considering.
2. Polyvinyl alcohol (PVA) plays a crucial role in this work as a versatile polymer used in the construction of various hydrogels and other materials. It would be beneficial to systematically summarize relevant studies related to PVA. This work is focused on materials innovation based on PVA, which has been extensively studied for various applications. It may be necessary to address the structure and properties of PVA, particularly when combined with other materials for different applications (see <https://doi.org/10.1016/j.jobab.2022.01.002>; <https://doi.org/10.1016/j.jobab.2021.11.004>; and <https://www.nature.com/articles/s41467-022-32987-6>). A concise review could be added to the introduction section. Notably, PVA hydrogels are widely utilized in various applications due to their biocompatibility, transparency, and high water content. These amorphous polymers can undergo an amorphous-crystalline transition under specific conditions. The COMPACT strategy for controlling this transition involves the introduction of multiple cross-linkers and acidification treatment to facilitate oriented polymeric crystalline growth under mechanical stretching.
3. Although some figures illustrate the mechanisms involved in the construction of hydrogel fibers, it would be advantageous to elaborate the explanations of the variations in crystallinity and the chemical bond formation processes.

Response to Referees

We appreciate the feedback from our peer reviewers. In response to the questions raised this round, we have conducted additional experiments, analyses, and discussions. These improvements are thoroughly integrated into the revised manuscript.

To facilitate a clear understanding of our revisions, we have organized our responses as follows:

- **Reviewer Comments and Our Responses:** Each comment from the reviewers (presented in blue text) is followed by our corresponding response (in black text).
- **Referencing Figures:** To support our responses, we have used figures which are referenced as follows:
 - Response-Related Figures: Figures accompanying our responses to the reviewers are labeled as Fig. R#.
 - Revised Manuscript Figures: Figures mentioned within the main text of the revised manuscript are numbered as Fig. #.
 - Supplementary Information Figures: Figures included in the revised Supplementary Information are denoted as Supplementary Fig. S#.
- **Highlighted Text in the Manuscript:** For ease of identification, all text modifications in the revised manuscript are highlighted in red.

Responses to Reviewer #1 comments

General comments:

The manuscript by Huang et al. has been much improved after revision. The authors provide an extensive set of materials characterizations to support their strategy of controlling amorphous-crystalline transition and hydrogel miniaturization. However, the authors still need to address several issues listed below before considering acceptance.

Response to general comment:

We sincerely thank the reviewer for the comment and feedback. In response to the comments, we have dedicated considerable efforts to comprehensively addressing the remaining concerns. In the latest revision of our manuscript, we have conducted additional experiments and analysis in the summary below:

- Mechanical Characterization of COMPACT Hydrogel Fibers under Water: to mimic the in vivo working condition, we characterized the COMPACT hydrogel fibers in a custom-designed tensile test assay with a water bath. We conducted a set of COMPACT hydrogel fibers (from 0% to 4% TEOS) by using the single-cycle tensile test underwater to thoroughly assess the elastic modulus and stretchability in hydrated status. These additional results are now included in the main text (**Fig. 2d**).
- Tissue Response Analysis to COMPACT Hydrogel Fibers: We have examined the tissue reactions to COMPACT hydrogel fibers at 14 days post-implantation. Utilizing

immunostaining techniques, we conducted a thorough evaluation of the tissue responses, specifically focusing on microglia activation, astrocyte migration, and macrophage formation. The detailed findings of these investigations have been added to the Supplementary Information (**Supplementary Fig. S16**).

- Evaluating the Performance of COMPACT Hydrogel Microelectrodes in vivo: We have conducted additional experiments to evaluate the stability of the electrical performance of our hydrogel microelectrodes over 14 days in vivo. These findings are now added to the Supplementary Information (**Supplementary Fig. S31**).

Comment 1:

The reviewer is still not convinced of the description of tissue-like elasticity as neural probes. It is true that modulus is not the only factor for consideration, stretchability and bending stiffness should also be included for a comprehensive evaluation. But a neural probe with high modulus implies a high risk of damage to tissues, especially considering long-term applications. Can the authors investigate the local tissue property surrounding the probe after 14-day implantation?

Response 1:

We are grateful for the reviewer's insightful feedback and apologize for any confusion caused by our previous terminology. To clarify, we have replaced the term “tissue-like elasticity” with “soft and stretchable” in the main text to describe the mechanical properties of our COMPACT hydrogel fibers more accurately. To keep the consistency of the characterization of hydrogel fibers' mechanical properties, we specified the measuring conditions in our revised manuscript. Previously, the wet hydrogel fibers were tested in the air during the tensile testing, during which the fast evaporation from hydrogel fibers and the consequent variation of water content will affect the mechanical properties. During the revision this round, we employed a horizontal tensile testing machine equipped with a water bath, which mimics the hydrated conditions in vivo. A similar characterization setup has been reported in our recent publication (*Liu, Rao, et. al., Nat. Methods, 2023*). We updated the mechanical properties characterization results, including elastic modulus and stretchability underwater in our revised manuscript. These results indicate that the COMPACT hydrogel fibers exhibit an elastic modulus ranging from 2.8 to 9.3 MPa and a stretchability between 139% to 169%. Our optimally formulated COMPACT hydrogel fiber (3% TEOS) exhibited an elastic modulus of 4.8 ± 1.7 MPa and a stretchability of $139 \pm 36\%$, showing no significant deviation from the 0% TEOS hydrogel fibers (**Fig. R1**). These findings underscore the effectiveness of our formulation in achieving desirable mechanical properties akin to in vivo tissue environments. The key aspects of our revised characterization are as follows:

Figure R1. COMPACT hydrogel fibers’ mechanical properties of elastic modulus (blue) and stretchability percentage (red) calculated from tensile tests in water. Each dot represents one independent fiber sample. One-way ANOVA and Tukey’s multiple comparisons test were used to determine the statistical significance of elastic modulus: ($F_{4,34}=30.07$, **** $p<0.0001$) and stretchability: ($F_{4,34}=1.040$), respectively.

Ref:

Liu, X., Rao, S., Chen, W. et al. Fatigue-resistant hydrogel optical fibers enable peripheral nerve optogenetics during locomotion. *Nat Methods* 20, 1802–1809 (2023).

In the revised manuscript, the aforementioned information has been updated on Page 12, Line 1 to Page 12, Line 6, as well as in Figure 2:

“To mimic the *in vivo* working condition, we examined COMPACT hydrogels’ mechanical properties in the hydrated state. The COMPACT hydrogel fibers exhibited relatively low elastic moduli while maintaining high stretchability (Fig.2d and Supplementary Fig. 14). The optimized COMPACT hydrogel fiber (3 wt. % TEOS, 12 mM HCl acidification treatment and 200% stretching, diameter: $227 \pm 18 \mu\text{m}$) exhibited an elastic modulus of $4.8 \pm 1.7 \text{ MPa}$ and stretchability of $139.4 \pm 26.0\%$.

Figure 2. Controllable hydrogel fiber fabrication and its properties. **a**, A shrinking diagram of COMPACT (+) hydrogel fibers. Each dot (mean \pm s.d.) represents an independent hydrogel fiber sample. The samples shaded in red areas are treated with acidification. **b**, Shrinking behaviors of COMPACT hydrogel fibers (4wt. % TEOS) prepared in different sizes of molds. Each dot (mean \pm s.d.) represents one independent fiber (One-way ANOVA and Tukey's multiple comparisons test, $F_{3,12}=0.9543$, n.s.: not significant, $p=0.4455$). **c**, COMPACT hydrogel fibers' optical properties of refractive index (blue) and normalized light transmittance (red) (mean \pm s.d.). Inset: representative photographs of 0 wt. % TEOS and 4 wt. % TEOS hydrogel membranes. Grid size: 1 mm. **d**, COMPACT hydrogel fibers' mechanical properties of elastic modulus (blue) and stretchability percentage (red) calculated from tensile tests in water. Each dot represents one independent fiber sample. One-way ANOVA and Tukey's multiple comparisons test were used to determine the statistical significance of elastic modulus: ($F_{4,34}=30.07$, **** $p<0.0001$) and stretchability: ($F_{4,34}=1.040$), respectively. **e**, Bending stiffness of COMPACT (3 wt. % TEOS) hydrogel fiber ($n=4$ independent samples) with identical cross-sections in comparison with a 200- μm silica waveguide ($n=3$ independent samples). **f**, Stability assessment of diameter reduction of COMPACT hydrogel fibers (3wt. % TEOS). Each dot (mean \pm s.d.) represents one independent fiber (two-way ANOVA and Tukey's multiple comparisons tests). **g**, Cytotoxicity assessment of COMPACT (+) hydrogels. Hydrogel incubated media was used to culture with Human Embryonic Kidney (HEK) 293 cell cultures. Calcein-AM (green) was used to stain living cells and ethidium homodimer-1 (red) was

used to stain dead cells. Cell death rates are presented as mean \pm standard error (s.e.m., unpaired student's t-test).

We further characterized and evaluated the tissue responses around the implantation site 14 days post-implantation. We bilaterally implanted a COMPACT hydrogel fiber (400 μ m) and a silica optical fiber (400 μ m) into the brains of the same mouse (n=3 mice). Following a 14-day incubation period, we harvested the brain tissues and conducted immunohistology staining to assess the tissue responses to both hydrogel and silica fibers. This involved characterizing the expression of markers associated with glial scarring: glial fibrillary acidic protein (GFAP) for astrocytes, ionized calcium-binding adapter molecule 1 (Iba1) for microglia, the presence of immunoglobulin G (CD16/32, an indicative of blood-brain barrier integrity), and cluster of differentiation 68 (CD68) for activated macrophages. Similar immunohistology analysis has been conducted in previous publications reporting neuroengineering technologies (*Nat Biotechnol.*, 2023; *Nat. Nanotechnol.* 2023; *Nat Commun.*, 2021; *Nat Neurosci.*, 2017; *Nat Biotechnol.*, 2015).

With fluorescent microscope imaging, we compared the total fluorescence area of labeled cells. The results exhibited that the total fluorescent area of hydrogel fibers was notably smaller than that around silica fibers for all markers assessed (GFAP, IBA1, CD16/32, and CD68). Furthermore, NeuN staining was employed to label neuronal cells near the implantation sites, revealing a significantly higher neuronal cell density surrounding hydrogel fibers compared to silica fibers (**Fig. R2**). We have added these results to the Supplementary Information (**Supplementary Fig. S16**).

We compared the immune response between COMPACT hydrogel and silica fibers after different periods of in vivo implantation (14 days and 30 days). In the in vivo application, bending stiffness is the dominant factor during micromotion. After 14 days of implantation, brain tissue surrounding hydrogel fibers exhibited significantly lower immune responses than that around silica fibers (**Fig. R2b**). Furthermore, when examining extended post-implantation periods (1 month, **Fig. R3**), the COMPACT hydrogel fibers consistently demonstrated a significantly decreased foreign body response relative to silica fibers (**Fig. R3d**). This sustained reduction in tissue reactivity is attributed to the significantly lower bending stiffness of hydrogel fibers as compared to silica fibers. The new finding has now been included in the Supplementary Information (**Supplementary Fig. S16**).

Figure R2. a, Representative confocal images of astrocytes (GFAP, red), microglia (Iba1, green), presence of immunoglobulin G (CD16/32, cyan), activated macrophages (CD68, yellow), and neuronal cells (NeuN, magenta) surrounding a 400 μm hydrogel fiber probe ($n=3$) and a 400 μm silica fiber ($n=3$) in mouse brain 14 days after implantation. **b**, measured total area of antibody-labeled cells of astrocytes (GFAP, red), microglia/macrophages (Iba1, green), macrophages (CD16/32, cyan), macrophages (CD68, yellow), and neuronal cells (NeuN, magenta) surrounding a 400 μm hydrogel fiber probe ($n=3$) and a 400 μm silica fiber in mouse brain 14 days after implantation. Each dot represents measurements from one confocal image around the implantation site, mean \pm sem.

Figure R3. **a**, Representative confocal images of astrocytes (GFAP, green) and microglia (Iba1, red) surrounding a 400 μm hydrogel fiber probe (n=6) in mouse brain 1 month after implantation. **b**, Representative optical images of astrocytes (GFAP, green) and microglia/macrophages (Iba1, red) surrounding a 400 μm silica fiber probe (n=6) in mouse brain 1 month after implantation. Scale: 100 μm. **c**, Left: counted GFAP fluorescence intensity from fibers implantation sites; Right: counted Iba1 fluorescence intensity from fibers implantation sites. (Unpaired student's t-test, GFAP: *p=0.0167; Iba1: *p=0.0108). **d**, Left: counted total area of astrocytes around fibers implantation sites; Right: Left: counted total area of microglia around fibers implantation sites. (Unpaired student's t-test, GFAP: *p=0.0251; Iba1: **p=0.0056). Each dot represents measurement from one confocal image around the implantation site, mean ± sem.

The description and discussion about brain tissue properties on immune response towards hydrogel fiber implantation has been updated in Page 12, Line 11 to Page 12, Line 17:

“Our initial evaluation focused on the immune response of brain tissues 14 days post-implantation. We observed a reduced presence of astrocytes, microglial accumulations, activated macrophages, and immunoglobulin G around the sites of hydrogel fiber implantation compared to the stiffer silica fibers (Supplementary Fig. S16). Subsequently, we assessed the chronic immune response after 30 days and noted a lower incidence of astrocyte and microglial formation around the hydrogel fiber sites relative to those with silica fibers (Supplementary Fig. S17).”

Ref:

1. Sahasrabudhe, A., Rupprecht, L.E., Orguc, S. et al. Multifunctional microelectronic fibers enable wireless modulation of gut and brain neural circuits. *Nat Biotechnol* (2023).

2. Le Floch, P., Zhao, S., Liu, R. et al. 3D spatiotemporally scalable in vivo neural probes based on fluorinated elastomers. *Nat. Nanotechnol.* (2023).
3. Park, S., Yuk, H., Zhao, R. et al. Adaptive and multifunctional hydrogel hybrid probes for long-term sensing and modulation of neural activity. *Nat Commun* 12, 3435 (2021).
4. Park, S., Guo, Y., Jia, X. et al. One-step optogenetics with multifunctional flexible polymer fibers. *Nat Neurosci* 20, 612–619 (2017).
5. Canales, A., Jia, X., Froriep, U. et al. Multifunctional fibers for simultaneous optical, electrical and chemical interrogation of neural circuits in vivo. *Nat Biotechnol* 33, 277–284 (2015).

Comment2:

In Fig. R5, the authors provide long term stability results in PBS, but how about the in vivo case? Will the contents such as proteins in real body fluids affect the stability of the hydrogel microelectrodes?

Response 2:

To address the question of hydrogel fibers' stability in vivo, we investigated the impedance stability of CNTs-PVA hydrogel electrodes in vivo and protein-rich solutions. We implanted COMPACT hydrogel microelectrodes (3% TEOS with 0.16% CNTs) in mouse brains (n=14 fibers) for 14 days and compared the impedance of hydrogel microelectrodes before and after the 14-day in vivo incubation period. The results indicated that there were no significant changes in electrical performance post-implantation (**Fig. R4a**). We also incubated the hydrogel microelectrodes in artificial cerebrospinal fluid (aCSF) (*Exton, Higgins, & Chen, Sci. Rep., 2023*), both with and without the addition of 1% bovine serum albumin, at 37 °C for 14 days. This experimental condition was designed to mimic the protein interactions that microelectrodes will typically encounter in vivo physiological conditions. We found that the microelectrodes' impedance remained stable, showing no significant differences pre- and post-incubation in both protein-enriched and protein-free aCSF environments (**Fig. R4b-c**). We have updated these new results in the Supplementary Information (**Supplementary Fig. S31**).

Figure R4. a, Impedance at 1 kHz of COMPACT hydrogel microelectrodes (3% TEOS with 0.16% CNT) before and after 14 days incubation in brains of mice (n=2). Paired t-test, n.s. $p=0.1544$. **b,** Impedance at 1 kHz of COMPACT hydrogel microelectrodes (3% TEOS with 0.16% CNT) before and after 14 days incubation (37 °C) artificial cerebrospinal fluid (aCSF). Paired t-test, n.s. $p=0.6321$. **c,** Impedance at 1 kHz of COMPACT hydrogel microelectrodes (3% TEOS with 0.16% CNT) before and after 14 days incubation (37 °C) aCSF with 1% bovine serum albumin (BSA). Paired t-test, n.s. $p=0.3796$. Each dot represent one individual microelectrode, mean \pm s.d..

The related description has been added on Page 16, Line 5 to Page 16, Line 8:

“The stability of CNT-PVA electrodes was evaluated through over 6 weeks of incubation in PBS (37 °C, Fig.4f), over 2 weeks of incubation in artificial cerebrospinal fluid (aCSF) and implantation in mouse brain tissues (Supplementary Fig. 31). We observed stable impedance performance under all these conditions.”

Our recent publication (Liu, Rao, et al., *Nat. Methods*, 2023) on PVA hydrogel fibers also presented their long-term stability in vivo. The PVA hydrogel optical fibers were implanted into the sciatic nerves of *Thy1-ChR2-EYFP* mice, a model chosen for its relevance in light delivery applications (Fig. R5b and d). After 8 weeks post-implantation, the PVA hydrogel fibers were still able to maintain stable light delivery efficiency to excite sciatic nerves and trigger muscle contraction. Even in moving animals, the hydrogel fibers after 8 weeks of implantation were able to achieve optogenetics-assisted pain inhibition. These in vivo results indicate that the hydrogel fibers maintained their integrity and operational efficacy in vivo, thus underscoring their suitability for long-term studies (Fig. R5g-i).

[REDACTED]

Figure R5 (*Liu, Rao, et. al., Nat. Methods, 2023*) [REDACTED]

Ref:

1. Exton, J., Higgins, J.M.G. & Chen, J. *Acute brain slice elastic modulus decreases over time. Sci Rep* 13, 12826 (2023).

2. Liu, X., Rao, S., Chen, W. et al. *Fatigue-resistant hydrogel optical fibers enable peripheral nerve optogenetics during locomotion. Nat Methods* 20, 1802–1809 (2023).

Comment3:

It is recommended to add more explanation about the results of materials characterization. For example, why does the impedance improve after cyclic stretching in Fig. R6? Why does the polymeric crystallinity decrease with the addition of CNTs in Fig. R2?

Response 3:

We thank the reviewer for the suggestions. We have thoroughly enhanced the discussion and the introduction parts of the manuscript.

In our revised manuscript, the discussion of crosslinking and acidification contribution to the hydrogel volumetric decrease has been updated on Page 9, Line 18 to Page 10, Line 5:

“An increased number of chemical cross-linkers, TEOS (0-4 wt. %, Fig. 1g), enhanced the anchoring of amorphous PVA chains through covalent cross-linking and prevented swelling in the hydrated state. Under the same cross-linking degree, acidification treatment granted polymer chains enhanced interactions and suppressed the folding of polymer chains to form crystalline (Fig. 1j and Supplementary Fig. 1c, 8, 9a-b, 10, and 11). Nanocrystalline domains maintained the nanoscale size (~3.5 nm) without compromising the light transmittance in the visible range. Axial mechanical deformation introduced tensile stress to re-orientate polymer chain alignment and created anisotropic nanostructures (Fig. 1k and Supplementary Fig. 4 and 5), which enabled hydrogel fibers’ desired decrease in diameter while causing a minimal effect on crystallinity degree (Supplementary Fig. 1a and b) or nanocrystalline size (Fig. 1k and Supplementary Fig. 3c and d).”

The discussion of the impedance change and the crystallinity of CNT-PVA hydrogel fibers has been enriched on Page 15, Line 11 to Page 15, Line 22:

“Through acidification to promote polymer chain interactions and mechanical stretching facilitated CNTs plaiting into polymer matrices and ensured entanglement with PVA chains and consequently augmented electrical conductivity as a percolated network. After introducing CNTs (0.08-0.24 wt. %), we observed insignificantly changed nanocrystallinity (Supplementary Fig. 24 and Table. S3) but decreased nanocrystalline sizes (Supplementary Fig. 25 and 26). These results could be interpreted as the limited polymer chain folding under additional interactions between nanomaterials and polymer chains. When CNT-PVA hydrogel fibers underwent mechanical stretching, they preserved anisotropic structures with similar sizes (Supplementary Fig. 9c, 27 and 28). With rigid carbon materials incorporated into the hydrogel matrix, the CNT-PVA composite

exhibited an increase in elastic modulus (0.16 wt. % CNTs-PVA hydrogels, 39.4 ± 13.7 MPa, and decreased stretchability ($47.9 \pm 12.2\%$, Supplementary Fig. 29).”

To provide a comprehensive introduction of semi-crystalline polymer properties, we added more background to elucidate the materials science mechanism of control hydrogel volume on Page 5, Line 9 to Page 5, Line 13:

“Inspired by the volumetric change resulting from polymer chains' folding and expansion, we propose a hypothesis centered on controlling the amorphous-crystalline transition in semi-crystalline hydrogels. By intervening in the polymer chain folding and crystallization process, we aim to limit the expansion of polymer chains from their nanocrystalline structure and consequently enable hydrogels to preserve their designed volumes under a solvated state.”

Page 5, Line 17 to Page 6, Line 2:

“In semi-crystalline polymer matrices, the swelling behavior involves water molecule diffusion, amorphous polymeric chain relaxation via hydration, and expansion of the cross-linked polymer network. To finely tune polymeric crystallization processes, we can apply engineering approaches that impact polymer chain interactions, solvent evaporation, and external stretching to facilitate molecular chain arrangement. Moreover, PVA polymer matrix can be incorporated with nanomaterials to enhance mechanical strength, conductivity, and biocompatibility. For PVA hydrogel bioelectronics, controlling nanostructures through polymeric crystallization approaches can enhance the stability to maintain their designed architectures in biological environments.”

Responses to Reviewer #2 comments

General comments:

Thank you for considering my comments and suggestions. The revised manuscript accurately incorporates my input and shows significant improvement. I have no further objections regarding the publications of your paper. Congratulations to your interesting results.

Response to general comment:

We are grateful for the reviewer's comment and deeply appreciate the insightful and constructive feedback provided during the review process.

Responses to Reviewer #3 comments

General comments:

In this revised manuscript, authors addressed most concerns raised previously. The manuscript indeed also showcases the fabrication of step-index hydrogel optical probes and CNTs-PVA hydrogel microelectrodes for various medical-related applications. The manuscript was well-written and effectively demonstrated an innovative method for constructing hydrogel bioelectronics, achieving miniaturization and multifunctional integration through the control of polymers' amorphous-crystalline transition. However, some very minor issues are needed to be addressed by authors prior to the acceptance of the manuscript for publishing *Natura Commun.*, a very top journal in relevant areas. Following are some specific comments:

Response to general comment:

We are grateful to the reviewer for their constructive recommendations. In response, we have incorporated the suggested material into the main text of the manuscript. This includes a revision of the title for greater clarity, an expanded review of PVA in the introduction section, and a more detailed discussion on the crystallinity and chemical bonding characteristics of PVA.

Comment 1:

The current title is already clear and accurate, but a slight modification would make it even better. For example, changing the title to "Controlling Polymers' Amorphous-Crystalline Transition Enables Miniaturization and Multifunctional Integration for Hydrogel Bioelectronics" may be worth considering.

Response 1:

We appreciate the reviewer's advice. We have now changed our title to "**Controlling Polymers' Amorphous-Crystalline Transition Enables Miniaturization and Multifunctional Integration for Hydrogel Bioelectronics**".

Comment2:

Polyvinyl alcohol (PVA) plays a crucial role in this work as a versatile polymer used in the construction of various hydrogels and other materials. It would be beneficial to systematically summarize relevant studies related to PVA. This work is focused on materials innovation based on PVA, which has been extensively studied for various applications. It may be necessary to address the structure and properties of PVA, particularly when combined with other materials for different applications (see <https://doi.org/10.1016/j.jobab.2022.01.002>; <https://doi.org/10.1016/j.jobab.2021.11.004>; and <https://www.nature.com/articles/s41467-022-32987-6>). A concise review could be added to the introduction section. Notably, PVA hydrogels are widely utilized in various applications due to their biocompatibility, transparency, and high water content. These amorphous polymers can undergo an amorphous-crystalline transition under

specific conditions. The COMPACT strategy for controlling this transition involves the introduction of multiple cross-linkers and acidification treatment to facilitate oriented polymeric crystalline growth under mechanical stretching.

Response 2:

We appreciate the reviewer's recognition of the importance of our work. We have extensively revised the introduction to offer a concise overview of PVA hydrogels and the materials science underlying their volumetric reduction. We also updated our literature citation to reflect the relevant references.

The changes have been updated on Page 5, Line 9 to Page 6, Line 2:

“Inspired by the volumetric change resulting from polymer chains' folding and expansion, we propose a hypothesis centered on controlling the amorphous-crystalline transition in semi-crystalline hydrogels. By intervening in the polymer chain folding and crystallization process, we aim to limit the expansion of polymer chains from their nanocrystalline structure and consequently enable hydrogels to preserve their designed volumes under a solvated state. Utilizing the nanoscale structure change to regulate soft materials properties has been proven as an effective approach to biomedical applications. Polyvinyl alcohol (PVA) hydrogels, one of typical semi-crystalline polymers, have been extensively used in drug release food packaging and wound healing owing to their high water content, transparency, and biocompatibility. In semi-crystalline polymer matrices, the swelling behavior involves water molecule diffusion, amorphous polymeric chain relaxation via hydration, and expansion of the cross-linked polymer network. To finely tune polymeric crystallization processes, we can apply engineering approaches that impact polymer chain interactions, solvent evaporation, and external stretching to facilitate molecular chain arrangement. Moreover, PVA polymer matrix can be incorporated with nanomaterials to enhance mechanical strength, conductivity, and biocompatibility. For PVA hydrogel bioelectronics, controlling nanostructures through polymeric crystallization approaches can enhance the stability to maintain their designed architectures in biological environments.”

Page 7, Line 5 to Page 7, Line 9:

“Chemically cross-linked PVA hydrogels have been widely employed with superior optical properties, fatigue-resistance and biocompatibility for bioelectronics applications. To further explore PVA hydrogels' controllable miniaturization properties while preserving these advantageous features, we designed new fabrication approaches by control of metamorphic polymers' amorphous-crystalline transition with the following aspects.”

Comment 3:

Although some figures illustrate the mechanisms involved in the construction of hydrogel fibers, it would be advantageous to elaborate the explanations of the variations in crystallinity and the chemical bond formation processes.

Response 3:

We thank the reviewer for the suggestions. In the revised manuscript, we have extensively improved the discussion on the polymer chain interactions and the crystallinity, as well as the corresponding property change.

We improved the discussion of crosslinking and acidification contribution to the hydrogel volumetric decrease on Page 9, Line 18 to Page 10, Line 5:

“An increased number of chemical cross-linkers, TEOS (0-4 wt. %, Fig. 1g), enhanced the anchoring of amorphous PVA chains through covalent cross-linking and prevented swelling in the hydrated state. Under the same cross-linking degree, acidification treatment granted polymer chains enhanced interactions and suppressed the folding of polymer chains to form crystalline (Fig. 1j and Supplementary Fig. 1c, 8, 9a-b, 10, and 11). Nanocrystalline domains maintained the nanoscale size (~3.5 nm) without compromising the light transmittance in the visible range. Axial mechanical deformation introduced tensile stress to re-orientate polymer chain alignment and created anisotropic nanostructures (Fig. 1k and Supplementary Fig. 4 and 5), which enabled hydrogel fibers' desired decrease in diameter while causing a minimal effect on crystallinity degree (Supplementary Fig. 1a and b) or nanocrystalline size (Fig. 1k and Supplementary Fig. 3c and d).”

The discussion of the impedance change and the crystallinity of CNT-PVA hydrogel fibers has been enriched on Page 15, Line 11 to Page 15, Line 22:

“Through acidification to promote polymer chain interactions and mechanical stretching facilitated CNTs plaiting into polymer matrices and ensured entanglement with PVA chains and consequently augmented electrical conductivity as a percolated network. After introducing CNTs (0.08-0.24 wt. %), we observed insignificantly changed nanocrystallinity (Supplementary Fig. 24 and Table. S3) but decreased nanocrystalline sizes (Supplementary Fig. 25 and 26). These results could be interpreted as the limited polymer chain folding under additional interactions between nanomaterials and polymer chains. When CNT-PVA hydrogel fibers underwent mechanical stretching, they preserved anisotropic structures with similar sizes (Supplementary Fig. 9c, 27 and 28). With rigid carbon materials incorporated into the hydrogel matrix, the CNT-PVA composite exhibited an increase in elastic modulus (0.16 wt. % CNTs-PVA hydrogels, 39.4 ± 13.7 MPa, and decreased stretchability ($47.9 \pm 12.2\%$, Supplementary Fig. 29).”

To provide a comprehensive introduction of semi-crystalline polymer properties, we added more background to elucidate the materials science mechanism of control hydrogel volume on Page 5, Line 9 to Page 5, Line 13, and Page 5, Line 17 to Page 6, Line 2:

“Inspired by the volumetric change resulting from polymer chains' folding and expansion, we propose a hypothesis centered on controlling the amorphous-crystalline transition in semi-crystalline hydrogels. By intervening in the polymer chain folding and crystallization process, we aim to limit the expansion of polymer chains from their nanocrystalline structure and consequently enable hydrogels to preserve their designed volumes under a solvated state.”

“In semi-crystalline polymer matrices, the swelling behavior involves water molecule diffusion, amorphous polymeric chain relaxation via hydration, and expansion of the cross-linked polymer network. To finely tune polymeric crystallization processes, we can apply engineering approaches

that impact polymer chain interactions, solvent evaporation, and external stretching to facilitate molecular chain arrangement. Moreover, PVA polymer matrix can be incorporated with nanomaterials to enhance mechanical strength, conductivity, and biocompatibility. For PVA hydrogel bioelectronics, controlling nanostructures through polymeric crystallization approaches can enhance the stability to maintain their designed architectures in biological environments.”

REVIEWERS' COMMENTS

Reviewer #1 (Remarks to the Author):

The authors addressed all of the comments from the referees. I feel that the revised version is suitable for publication in its current form.

Reviewer #3 (Remarks to the Author):

Authors have well addressed the concerns raised previously. The revised manuscript is acceptable for publication.